# Rainbow Generator: A Generative Approach for Name Only Continual Learning

## Abstract

Requiring extensive human supervision is often impractical for continual learning due to its cost, leading to the emergence of 'name-only continual learning' that only provides the name of new concepts (*e.g.*, classes) without providing supervised samples. To address the task, recent approach uses web-scraped data but results in issues such as data imbalance, copyright, and privacy concerns. To overcome the limitations of both human supervision and webly supervision, we propose *Generative name only Continual Learning* (**GenCL**) using generative models for the name only continual learning. But naïve application of generative models results in limited diversity of generated data. So, we specifically propose a diverse prompt generation method, HIerarchical Recurrent Prompt Generation (**HIRPG**) as well as COmplexity-NAvigating eNsembler (**CONAN**) that selects samples with minimal overlap from multiple generative models. We empirically validate that the proposed GenCL outperforms prior arts, even a model trained with fully supervised data, in various tasks including image recognition and multi-modal visual reasoning. Data generated by GenCL is available at https://anonymous.4open.science/r/name-only-continual-E079

## 1 Introduction

Continual learning (CL) has been addressing various domains across different modalities including computer vision (Seo et al., 2024b), natural language processing (Wu et al., 2024b), and multimodal learning (He et al., 2023a). But even the most existing methods (Wang et al., 2022; Kim et al., 2024a) often rely on abundant well-curated human supervision. For example, obtaining 423.5k clean images in DomainNet (Neyshabur et al., 2020) required 50,000 working hours to manually filter out outliers.

In standard learning, where all training data are provided at once, the time allocated for data collection and preprocessing does not affect performance, since these steps are completed before model training begins. In contrast, continual learning involves the continuous encounter of new 'concepts', which can refer to classes, adjectives, and verbs in multi-modal tasks, necessitating ongoing data preparation throughout training. Therefore, delays in preparing data for encountered concepts hinder the model's ability to quickly adapt to new concepts (Koh et al., 2021; Caccia et al., 2022). Consequently, delays in data preparation, such as human annotation, could limit the applicability of CL method in deployment, such as e-commerce recommendation systems and autonomous driving.

In recent literature, several alternatives to human annotation have been proposed. Madaan et al. (2021) propose an unsupervised CL setup that eliminates the need for annotation. However, they assume that the unlabeled data stream contains only data related to the target concepts, while in real-world scenarios, unlabeled data often include irrelevant data, which potentially hinder the performance of the target concept (Halevy et al., 2016; Yang et al., 2023). As another alternative to human-annotated data, Sato (2023); Prabhu et al. (2024) proposes the use of web-scraped data for online learning. Although web-scrapped data presents advantages such as abundance (Xu et al., 2024), diversity (Agun, 2023), and easy accessibility (Sun et al., 2018), challenges arise from privacy and copyright concerns (Zhang et al., 2023a), as well as inherent noise (Neyshabur et al., 2020), which significantly hinders the performance of continual learner (Kim et al., 2021; Bang et al., 2022).

To address those issues, we propose to leverage text-to-image (T2I) generative models for CL. Specifically, we propose *Generative name only Continual Learning* (**GenCL**), which takes only *concepts* as input and trains on images generated by text-to-image (T2I) generative models based on the

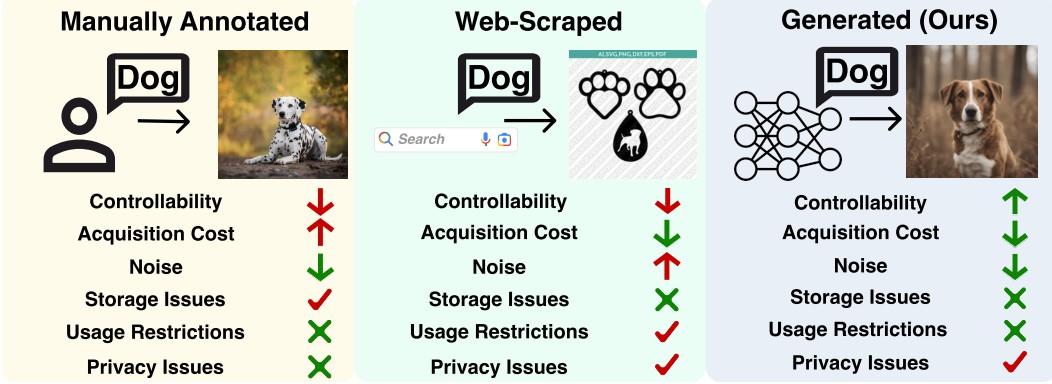

Figure 1: **Comparison of Manually Annotated (MA) data, Web-Scraped Data, and Generated data.** Generated data addresses constraints associated with Web-scraped or MA data, mitigating privacy concerns and usage restrictions (*i.e.*, whether images can be used for learning). Also, it maintains controllability (the ability to generate images with various contexts, *e.g.*, background, color) as desired. Generated data are less noisy (*i.e.*, containing fewer undesired images) than web-scrapped data and proves to be a more cost-effective than MA data which requires human annotation. For more details on the terminology employed in this figure, see Sec. A.19

given *concepts*. It takes advantage of generative models, such as controllability (Nie et al., 2021) (*i.e.*, generating desired data), and unlimited image generation (Liang et al., 2022), as illustrated in Fig. 1. Additionally, it significantly accelerates the data collection process; for example, generating DomainNet using SDXL (Podell et al., 2023) with 8 NVIDIA RTX 4090 GPUs take only 80 hours, compared to the 50,000 hours required for manual annotation.

However, generated images often suffer from limited diversity (Tian et al., 2024a; Bianchi et al., 2023; Fraser et al., 2023). To address this issue, we define *intra-diversity* and *inter-diversity*, which refers to the diversity of data generated by a single T2I model and the diversity of data generated by multiple T2I models, respectively. Specifically, to improve intra-diversity, we propose HIerarchical Recurrent Prompt Generation (**HIRPG**), a diverse prompt generation method that utilizes the in-context learning capabilities of Large Language Models (LLMs) to generate a diverse set of text prompts. To improve the inter-diversity, we propose a complexity-guided data ensemble method, named COmplexity-NAvigating eNsembler (**CONAN**). CONAN not only ensembles data from multiple generative models, but also selects a coreset, and trains a model exclusively on this coreset to improve training efficiency for real-time adaptation to new concepts. We empirically demonstrate that our framework significantly outperforms baselines in both class-incremental and multi-modal visual concept-incremental setups.

In sum, we aim to address the following research questions, thus summarizing our core contributions as follows:

RQ1. *Can generated data substitute manually annotated (MA) in CL setups?* **GenCL** improves $A_{\mathrm{AUC}}$ on the PACS (Zhou et al., 2020) OOD domain by 9% and 13% over the model trained with web-scrapped and MA data, respectively.

RQ2. *How to ensure diversity of images generated from generative models?* We propose **HIRPG**, a prompt generation method that takes advantage of LLM to create diverse text prompts for a given concept, which are subsequently used by T2I models to generate images.

RQ3. *How to ensemble generated data from a set of generators?* We propose **CONAN**, a data selection method that accounts for the complexity of generated samples.

## 2 RELATED WORK

### 2.1 CONTINUAL LEARNING

**Setups for Continual Learning.** Many recent works propose realistic CL scenarios, such as blurry (Prabhu et al., 2020; Bang et al., 2021), i-blurry (Koh et al., 2021), continuous (Shanahan et al., 2021; Koh et al., 2023), and noisy (Bang et al., 2022) setups. However, they only focus on the realistic data distribution of the stream data, rather than the acquisition of data for a new category, for which the model needs to be learned. Recently, C2C (Prabhu et al., 2024) used web-scraped data for

continual learning, to address the high cost of manual data annotation and the difficulty in acquiring real-time data for the target concepts the model needs to learn. However, web-scraped data present several limitations, including privacy concerns, usage restrictions, and inherent noise, as highlighted in Fig. 1. Please refer to Sec. A.19 for more comparison between web data and generated data.

## 2.2 Data Selection

For ensembling generated images from multiple T2I generative models, we consider data selection methods that extract most essential samples to build a coreset from a larger candidate set. Formally, from the candidate set $T$, these methods select a coreset $V$ ($|V| \ll |T|$), aiming to preserve as much task-relevant information from $T$ as possible (Shin et al., 2023). To estimate the informativeness of the candidates, several metrics have been proposed, such as gradient (Paul et al., 2021; Pooladzandi et al., 2022; Shin et al., 2023), uncertainty (Coleman et al., 2020), influential score (Yang et al., 2022a; Pooladzandi et al., 2022), and distance (Xia et al., 2023).

Although these methods are effective at selecting coresets, many come with substantial computational costs. Gradient-based methods (Paul et al., 2021; Pooladzandi et al., 2022; Shin et al., 2023), which aim to minimize the difference between the gradients of the training dataset $T$ and the selected set $V$, require a well-trained model on $T$, which significantly increases computational overhead. Similarly, the influence score-based method (Yang et al., 2022a) also requires significant computation due to the necessity of calculating the Hessian in the influence function, along with its iterative data selection process (Xia et al., 2023). In contrast, distance-based methods, such as Xia et al. (2023) and our proposed CONAN, can directly leverage a well-trained feature extractor, *i.e.*, requiring only model forward passes for feature extraction, leading to faster data preparation time.

## 2.3 Training with Text-to-Image (T2I) Generative Models

With the availability of robust generative models (Gu et al., 2023; Tang et al., 2023; Podell et al., 2023), several recent studies have utilized synthetic data for training (Azizi et al., 2023; Tian et al., 2024a; Zhang et al., 2024c; Tian et al., 2024b). Notably, (Tian et al., 2024a) demonstrate the positive impact of using diffusion model-generated datasets at ImageNet (Deng et al., 2009) scale for training.

To train a model with data generated by T2I generative models for a given concept $c$, concept-specific prompts $p_c$ are needed. Ramesh et al. (2022); Jones et al. (2024) use the template "A photo of a $c$", as proposed in CLIP (Radford et al., 2021), to construct prompt for concept $c$. However, Sarıyıldız et al. (2023) claim that using only the class name as a prompt ($p_c$ = "$c$") yields better image generation than $p_c$ = "A photo of a $c$". To add more concept-specific context to $p_c$, Sarıyıldız et al. (2023) combine the concept name $c$ with its definition $d_c$ from WordNet (Miller, 1995), resulting in $p_c$ = "$c, d_c$". Nonetheless, all these approaches rely on a single type of $p_c$ per concept, limiting the diversity of generated images despite the ability of T2I models to generate an unlimited number of images (Vardanyan et al., 2024; Tian et al., 2024a).

To address the limited diversity in generated images, several prompt diversification methods have been proposed. LE (He et al., 2023b) leverages a pre-trained word-to-sentence T5 model (Raffel et al., 2020) to generate diverse sentences that incorporate class names. Furthermore, Sarıyıldız et al. (2023) integrates the concept's hypernym $h_c$ from WordNet, along with a background scene $b$ from the 365 scenes in Places365 (López-Cifuentes et al., 2020), resulting in $p_c$ = "$c, h_c$ inside $b$". In contrast to random background selection, Tian et al. (2024a) utilize LLM to generate a list of contextually appropriate backgrounds for the given concept $c$ to create more plausible prompts. Similarly, Hammoud et al. (2024) and our proposed diverse prompt generation method, HIRPG, also employ an LLM-based prompt generator. However, while previous LLM-based prompt generation methods do not account for the relationship between generated prompts, HIRPG minimizes overlap between them by providing previously generated prompts as negative examples to the LLM.

We review more relevant literature and provide extended related work in Sec. A.18 for space's sake.

## 3 Problem Statement of Name only Continual Learning

In the name-only CL setup (Prabhu et al., 2024), only new concepts to be learned, denoted $\mathcal{Y} = \{y_1, y_2, ...\}$, are provided in a streaming manner, while the prevalent online continual learning setups

assume well-curated annotated data $(\mathcal{X}, \mathcal{Y})$ are given. The objective of this setup is to train a model $f_\theta$, parameterized by $\theta$, to classify the data into the concepts seen up to the given time step $t$, *i.e.*, $\{y_i\}_{i=1}^t$. To train $f_\theta$ for the given concepts, the learner can access either public data, such as data scraped on the Web (Prabhu et al., 2024), or generated data. To evaluate whether the model $f_\theta$ well-learn concepts $\{y_i\}_{i=1}^t$ at the time step $t$, curated data $\{\mathcal{X}_i, y_i\}_{i=1}^k$ are used, where $\mathcal{X}_i$ refers to the set of data that corresponds to the category $y_i$.

## 4 APPROACH

We propose a Generative name only Continual Learning (**GenCL**) framework to address the absence of data in the name-only CL setup. The GenCL framework is composed of four integral components: (i) a Prompt Generation Module $\psi$ (Sec. 4.1), (ii) a set of Generators $\mathcal{G}$ (Sec. 4.2), (iii) an Ensembler $\Delta$ (Sec. 4.3), and (iv) a learner $f_\theta$. We illustrate an overview of GenCL in Fig. 2.

When a new concept is introduced, for which $f_\theta$ needs to be learned, a generator $g \in \mathcal{G}$ generates images related to the concept. However, despite generative models being capable of producing an unlimited number of images, output diversity is often limited (Liu et al., 2023a; Sadat et al., 2023). To address this, we employ a prompt generation module $\psi$, which generates diverse text prompts and forwards them to the T2I generative models. Additionally, we further enhance the diversity of the generated images by utilizing a proposed ensemble approach that combines the outputs of a set of generators $\mathcal{G}$ through ensembler $\Delta$. Generated images are streamed to the learner $f_\theta$ in real-time, while a finite episodic memory is maintained to replay previously encountered data. Note that while it is possible to generate data for past concepts in real-time without using episodic memory, we use it for efficiency, as it helps reduce computational costs.

### 4.1 PROMPT GENERATION MODULE ($\psi$)

Our pipeline $\psi$ begins with the new concept as input and constructs a base prompt $P_B$ using the template: '*A photo of [concept]*' following (Shtedritski et al., 2023; Shi et al., 2023). While this base prompt can be used directly with the T2I generators $\mathcal{G}$, generating images using a single prompt may lead to limited diversity in style, texture, and backgrounds across the generated images (Fan et al., 2024). To enhance diversity, we generate additional diverse prompts using LLMs.

A straightforward approach for diverse prompt generation using LLMs is to generate $N$ different prompts at once or to generate a single prompt $N$ times, as in previous work (He et al., 2023b; Hammoud et al., 2024). However, multiple inferences to LLM with the same input can produce similar outputs (Zhang et al., 2024a; Skapars et al., 2024), despite the non-deterministic nature of LLM (Song et al., 2024). Empirically, as shown in Sec. A.32, our observations indicate that using these approaches often leads to many generated prompts with similar meanings, which may reduce the diversity of the generated images from T2I generative models.

To reduce overlap between generated prompts, we iteratively create new prompts that are distinct from those produced in previous steps. Inspired by previous work that has shown improved performance in solving complex problems by providing negative examples with positive examples in in-context learning (Zhang et al., 2024b) and contrastive Chain-of-Thought (Chia et al., 2023), we incorporate previously generated prompts into the LLM input to serve as negative examples. By presenting these previously generated prompts and requesting a new prompt that is distinct from them, we impose a hard constraint that effectively prevents overlap between the newly generated prompts and the previous ones. Formally, this process can be described as follows:

$$P_i = \begin{cases} \text{LLM}(P_S, P_B) & i = 1 \\ \text{LLM}(P_S, \{P_B\} \cup \{P_m\}_{m=1}^{i-1}) & i \geq 2, \end{cases} \tag{1}$$

which takes the system prompt $P_S$ and all previously generated prompts $\{P_m\}_{m=1}^{i-1}$ as input. Since there are no negative examples in the initial step (*i.e.*, $i = 1$), we use the base prompt $P_B$ as the initial negative example, where $P_B = $ '*A photo of [concept]*'. To generate $N$ different prompts, we repeat the process $N$ times. As previously generated prompts are iteratively used as negative examples, we name this process as Recurrent Prompt Generation (RPG). The system prompt $P_S$ we use is as follows:

Figure 2: **Illustration of the proposed GenCL framework.** When a new concept that needs to be learned is encountered, it is passed through a prompt generation module, $\psi$, to produce diverse prompts. These prompts are then used to generate data from a set of generators, $\mathcal{G}$. The data generated by each generator are combined through the ensembler, $\Delta$, and subsequently used to train the model, $f_\theta$.

> To generate images using a text-to-image generative model, I need to create a prompt ~~~
> **Here is a list of prompts that I have previously generated. Please create a new prompt that does not overlap with {base prompt & previously generated prompts}** ~~~.

However, generating a large number of prompts using RPG poses a challenge. As the iterative steps are repeated, the length of the LLM input for in-context learning (ICL) increases, which can lead to difficulties in fully utilizing information within the long context, a problem known as *lost-in-the-middle challenge* (An et al., 2024; Liu et al., 2023c), as well as substantial computational overhead in long-context ICL (Li et al., 2024).

To address this challenge, we divide the RPG into multiple subtasks using a hierarchical tree structure. Specifically, we construct a complete $K$-ary tree (Gross et al., 2018), where every internal node has exactly $K$ child nodes. Each node represents a prompt, with the root node (*i.e.*, the node at depth 0) defined as $P_B = $ '*A photo of [concept]*'. To generate the $K$ child nodes at depth 1, we first perform RPG. To generate more diverse prompts, we extend the tree to depth 2, again using RPG, with each parent node at depth 1 serving as the base prompt $P_B$ in Eq. 1, and this process continues for subsequent depths. Formally, focusing on the $k^{\text{th}}$ child node at depth $d$, denoted as $P_{d,k}$ ($d \geq 0,\ 1 \leq k \leq K$), its child nodes $P_{d+1,k'}$ ($1 \leq k' \leq K$) are generated through the RPG as follows:

$$P_{d+1,k'} = \begin{cases} \text{LLM}(P_S, P_{d,k}) & k' = 1 \\ \text{LLM}(P_S, \{P_{d,k}\} \cup \{P_{d+1,m}\}_{m=1}^{k'-1}) & 2 \leq k' \leq K, \end{cases} \quad (2)$$

where $\{P_{d+1,m}\}_{m=1}^{k'-1}$ refers to the previously generated nodes that share the same parent node $P_{d,k}$. By constructing complete $K$-ary Tree with a depth of $D$, we can generate $\frac{K^{d+1}-1}{K-1}$ nodes (*i.e.*, prompts), which includes all internal and leaf nodes. This hierarchical generation enables us to generate diverse prompts while bounding the number of negative examples by $K$ ($\ll N$), thereby addressing both the *lost-in-the-middle challenge* and the computational overhead. We name this proposed diverse prompt generation process as HIerarchical Recurrent Prompt Generation (**HIRPG**).

Since we divide RPG into subtasks using a hierarchical tree structure, we cannot consider nodes generated from different branches as negative examples during the RPG in each node. Nonetheless, overlap between generated prompts from different nodes is rare. This is because RPG in each node begins with a distinct $P_{d,k}$ in Eq. 2, serving as a negative example in the first step ($k' = 1$), and different examples in in-context learning lead to varied outputs (Su et al., 2022; Agarwal et al., 2024). We empirically demonstrate the effectiveness of HIRPG by comparing it quantitatively and qualitatively with existing prompt generation methods in Sec.5.2 and Sec.A.32, respectively. We provide a pseudocode for the prompt diversification module $\psi$ in Algorithm 3 in the appendix.

## 4.2 GENERATORS ($\mathcal{G}$)

In addition to enhancing intra-diversity, we amplify the inter-diversity, the diversity between images generated by multiple T2I generative models, by ensembling the images generated by these models. Specifically, using a T2I generator $g_i(\cdot) \in \mathcal{G}$ and a prompt set $\mathbf{P}$ generated by $\psi$, we generate a

set of images $U_i = g_i(\mathbf{P})$. At the end of generation, we have $\mathbf{U} = \bigcup_{i=1}^{|\mathcal{G}|} U_i$, the union of images generated by $|\mathcal{G}|$ generative models, with the same number of images generated for each model, *i.e.*, $|U_1| = |U_2| = \cdots = |U_{|\mathcal{G}|}|$. We provide detailed information about the generators we employ, including examples of generated images from each generator in Sec. A.17.

## 4.3 ENSEMBLER ($\Delta$)

When ensembling images generated by different T2I models, a key question arises: *Do we need to use all of them?* While large-scale training datasets have become standard for achieving state-of-the-art deep learning models (Zhao et al., 2021; Yang et al., 2022b), training with massive data imposes not only computational burden and time complexity (Sharir et al., 2020; Kim et al., 2022), but also significant energy and carbon footprint (Schwartz et al., 2020; Patterson et al., 2021). In addition, in CL setups, prolonged training periods can hinder fast adaptation to new concepts (Seo et al., 2024a).

Therefore, we aim to select a coreset $\mathbf{V}$ from the entire generated data $\mathbf{U}$, and train a learner only using $\mathbf{V}$. Specifically, we select $|U_i|$ ($U_i \in \mathbf{U}$) samples from $\mathbf{U}$ for $\mathbf{V}$ to maintain the same training cost as using data generated by a single generative model while increasing the diversity of the ensemble set. A straightforward selection method for constructing $\mathbf{V}$ is to sample images from each generator with equal weights. However, surprisingly, this method degrades the performance of models trained with ensembled images, even compared to those trained on images from a single generative model (*i.e.*, no ensembling), as shown in Tab. 3. This degradation occurs because the equal-weight selection method does not account for the overlap between images, *i.e.*, diversity.

To enhance diversity in the ensembled set, we select samples positioned far from the class prototype in the feature space, *i.e.*, *difficult samples*, since these images are less likely to overlap with common images compared to those that are closer to the prototype. To achieve this, we consider the class-wise Mahalanobis distance (Mahalanobis, 2018), where a higher distance indicates that a sample is farther from the class prototype. However, it only accounts for the class-specific difficulty, while the distance from other classes can also affect the difficulty of samples. For example, consider two samples, $x_1$ and $x_2$, both belonging to class $c$ and having the same class-wise Mahalanobis distance. If $x_1$ is closer to the global prototype (*i.e.*, the class-agnostic prototype) than $x_2$, then $x_1$ may be more challenging to classify as class $c$, since $x_1$ is more likely to be confused with other classes. Therefore, to select difficult samples in the ensemble set while considering for both class-wise difficulty and their relationship to other classes, we employ the relative Mahalanobis distance (RMD) score (Ren et al., 2021). It measures the *difficulty* in classifying a sample into its corresponding class by comparing the distance from the class prototype with the distance from the global prototype (Cui et al., 2023). The RMD score for a sample $(x_i, y_i) \in \mathbf{U}$ is given by the following:

$$\mathcal{RMD}(x_i, y_i) = \mathcal{M}(x_i, y_i) - \mathcal{M}_{\text{agn}}(x_i),$$

$$\mathcal{M}(x,y) = D_M\left(g(x), \frac{1}{|\mathbf{U}_y|} \sum_{j \in \mathbf{U}_y} f(x_j)\right), \quad \mathcal{M}_{agn}(x) = D_M\left(g(x), \frac{1}{|\mathbf{U}|} \sum_{j \in \mathbf{U}} f(x_j)\right), \quad (3)$$

where $g(x)$ refers to the penultimate feature of the feature extractor $g$, $D_M$ refers to the Mahalanobis distance (MD), $\mathbf{U}_y$ denotes the set of samples belonging to class $y$, $\mathcal{M}(x_i, y_i)$ and $\mathcal{M}_{\text{agn}}(x_i)$ represents class-wise MD and class-agnostic MD (*i.e.*, global MD), respectively. If a sample is close to the class prototype but far from the global prototype (*i.e.*, low RMD score), it is easy to classify correctly. Conversely, if it is far from the class prototype but close to the global prototype (*i.e.*, high RMD score), the sample is hard to classify correctly and may belong to other near-classes. We show samples with low RMD scores and samples with high RMD scores in Fig. 3.

Measuring the difficulty using the RMD score, we select images with high RMD scores, which are expected to exhibit a widespread dispersion from the class prototype. However, in the coreset, which is a representative subset of an entire dataset (Anonymous, 2023), it is necessary to include not only samples near the decision boundary, but also class-representative samples (Bang et al., 2021; Harun et al., 2023). Therefore, we adopt a probabilistic approach to ensemble selection, rather than simply choosing images with the $k$-highest RMD scores, to incorporate class-representative samples into the ensemble set. Specifically, we calculate $p_{u|c}$, the selection probability for sample $u$ to be included in the coreset of class $c$, with details provided below.

First, we truncate the samples with RMD scores in the upper and lower $L\%$ to minimize the impact of outliers on the probability distribution. Next, we normalize the scores using Z-score normalization and apply a softmax function to obtain the selection probability as:

$$p_{u|c} = \frac{e^{R\bar{M}D_{u|c}/\tau}}{\sum_{u' \in \mathbf{U}_c} e^{R\bar{M}D_{u'|c}/\tau}}, \qquad (4)$$

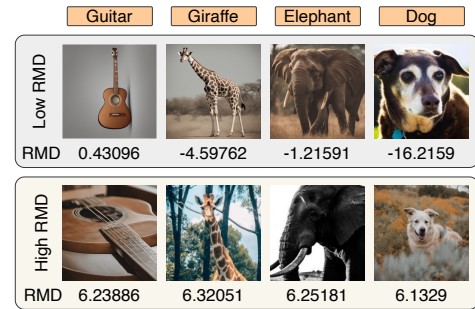

Figure 3: **Samples with high RMD scores and low RMD scores**

where $\mathbf{U}_c$ refers to the set of samples for class $c$, $R\bar{M}D_{u|c}$ represents the normalized RMD score for sample $u \in \mathbf{U}_c$, and $\tau$ denotes the temperature parameter. Using the selection probability, we not only sample complex samples, but also incorporate a small portion of class-representative samples into the ensemble set. We name our proposed RMD-based probabilistic ensemble method as COmplexity-NAvigating eNsembler (**CONAN**).

We compare CONAN with various RMD-based ensemble methods and existing coreset selection methods in Sec.A.14 and Sec.5.2, respectively. Furthermore, we provide additional justification for using the RMD score in Sec. A.13.

## 5 EXPERIMENTS

### 5.1 EXPERIMENTAL SETUP

**Continual Learning Setups.**    We first empirically validate the efficacy of our GenCL by comparing it with state-of-the-art methods in class-incremental learning (CIL) task setups. Beyond CIL setups, we also assess GenCL in multi-modal visual-concept incremental learning (MVCIL). In MVCIL, concepts to be learned (*e.g.*, 'ride a bike', 'kick a ball') are encountered incrementally. To learn a concept, both positive and negative support sets are required, where the positive set contains images representing the concept, while the negative set contains images that do not. We consider two types of tasks that address the following queries: (1) *What is the concept exclusively depicted by the positive support set?* and (2) *Give a query image, does the query image belong to the positive or negative support set?*. We refer to these tasks as CA (Concept Answering) and P/N, respectively. We provide a detailed explanation of the MVCIL setup in Sec. A.1.

**Models.**    We use ResNet-18 and ImageNet-1K pretrained ViT-base as the network architecture for the CIL setup. For the MVCIL setup, we fine-tune the LLaVA-1.5-7B model (Liu et al., 2023b). Specifically, following Ye et al. (2023); Dong et al. (2024), we fine-tune only the pretrained projection MLP layers and LoRA adapters (Hu et al., 2021), keeping the LLM frozen for training efficiency. In all experiments, we train a model with ER (Rolnick et al., 2019), which is a simple but strong CL method (Prabhu et al., 2023; Seo et al., 2024a).

**Datasets.**    We evaluate the domain generalization performance of GenCL in the CIL setup using widely adopted domain generalization (DG) benchmarks: PACS (Zhou et al., 2020), NICO (Zhang et al., 2023c) and DomainNet (Neyshabur et al., 2020), dividing them into multiple discrete tasks. Each DG benchmark consists of multiple domain datasets, *e.g.*, PACS includes four domains: Photo, Art, Cartoon, and Sketch. We selected data from one domain (*i.e.*, photo domain) as MA data for each benchmark, to compare GenCL with the oracle scenario, which assumes that manually annotated (MA) data are available for training. During the evaluation, we considered the selected domain as the in-distribution (ID) domain, while the other domains as out-of-distribution (OOD) domains. For details on the task splits for each benchmark, please refer to Sec.A.2 due to space's sake. For the MVCIL setup, we used Bongard-HOI (Jiang et al., 2022) and Bongard-OpenWorld (Wu et al., 2024a).

**Metrics.**    We report $A_{\text{AUC}}$ (Koh et al., 2021; Caccia et al., 2022; Koh et al., 2023) and $A_{last}$, which measure inference performance at any time and at the end of training, respectively. In MVCIL-CA task setups, to compare model-predicted sentences with ground-truth sentences, we use CiDER (Vedantam et al., 2015), which measures the similarity between generated and ground truth sentences, while also

capturing aspects such as grammaticality, saliency, importance, and both precision and recall. Note that for evaluation, we use the test set for seen categories up to that point in time. Please refer to Sec. A.16 for a more detailed explanation of the metrics we used.

**Baselines.** We compare a model trained using GenCL with models trained using web-scraped data (C2C (Prabhu et al., 2024), IE (Li et al., 2023b)), other synthetic data (Glide-Syn(He et al., 2023b), CHB (Sarıyıldız et al., 2023), SC (Tian et al., 2024a), LE (He et al., 2023b), CCG (Hammoud et al., 2024)), and Real-Fake (Yuan et al., 2024), and manually annotated (MA) data. Specifically, CHB, SC, LE, and CCG generate diverse prompts to enhance the variety of generated images. We compare these methods with our proposed diverse prompt generation method, *i.e.*, HIRPG. Furthermore, we integrate prompt generation baselines with our proposed ensemble method (CONAN) to showcase the effectiveness of CONAN, as well as to provide a fair comparison with GenCL, which leverages multiple generators for ensembling.

Next, we extend GenCL to a standard learning setup (*i.e.*, joint training), where all concepts to be learned are provided at once. In this setup, we compare a model trained with data generated by GenCL to models trained with web-scraped data, other synthetic data, and manually annotated (MA) data. We also compare with training-free baselines, including CLIP-ZS(Radford et al., 2021), SuS-X-SD(Udandarao et al., 2023), CuPL (Pratt et al., 2023), VisDesc (Menon & Vondrick, 2023), and CALIP (Guo et al., 2023), as well as SD-Clf (Li et al., 2023a), which utilizes SDXL as a classifier.

Finally, we compare our proposed ensemble selection method, *i.e.*, CONAN, with various baselines for coreset selection, including Uncertainty (Coleman et al., 2020), CRAIG (Mirzasoleiman et al., 2020), Glister (Killamsetty et al., 2021b), GradMatch (Killamsetty et al., 2021a), Adacore (Pooladzandi et al., 2022), LCMat (Shin et al., 2023), and Moderate (Xia et al., 2023).

For detailed description of training-free name-only classification baselines, prompt generation baselines and data ensemble baselines, see Sec. A.12, Sec. A.10 and Sec. A.11, respectively.

## 5.2 QUANTITATIVE ANALYSIS

**Effectiveness of GenCL in CIL.** To assess the effectiveness of our proposed GenCL in a setup, where only concept names are provided without data, we compare its performance against models trained on web-scraped data, other synthetic data, as well as manually annotated data, representing the ideal case. In the CIL setup, we assume that the category names of the PACS and DomainNet datasets are provided incrementally, and we summarize the results not only in the ID domain but also in the OOD domain in Tab. 1. Note that since DomainNet is a web-scraped dataset, sharing the same domain as C2C (Prabhu et al., 2024), which also uses web-scraping for data acquisition, we exclude C2C from the comparison in DomainNet.

In in-distribution (ID) domain, MA outperforms other baselines, as well as GenCL. This is because the ID test set we use is derived from the same test set as the MA data, giving it an advantage in this specific domain. However, in the out-of-distribution (OOD) domains of PACS, CONAN outperforms both MA and other baselines. We believe that we achieve better generalization performance by generating a more diverse set of images through a diversified set of prompts and an ensemble selection of generators. We provide additional comparisons with various combinations of diverse prompt generation baselines and data ensemble methods in Sec. A.33.

**Effectiveness of GenCL in MVCIL.** We also empirically validate the effectiveness of GenCL in the MVCIL setup, and summarize the result in Tab. 2. Since CHB, SC, and CCG focus solely on image classification tasks, we exclude them from the MVCIL setup. Additionally, Glide-Syn and LE utilize a word-to-sentence model to generate diverse prompts, making them inapplicable to Bongard-OpenWorld, which uses sentences as concepts. Note that while C2C and manually annotated (MA) data utilize human-annotated hard negative concepts alongside the concepts to be learned (*i.e.*, positive concepts), GenCL relies solely on positive concepts. Specifically, to acquire MA data, high-quality annotators from Amazon Mechanical Turk were employed to select hard negative examples and filter out noisy data (Jiang et al., 2022). In contrast, GenCL automatically selects relevant hard negative concepts based on the specified positive concept, leveraging commonsense priors from large language models (Zhao et al., 2023; Yang et al., 2024). For web-scraped data, since

| Method | PACS | | | | DomainNet | | | |
|---|---|---|---|---|---|---|---|---|
| | ID | | OOD | | ID | | OOD | |
| | $A_{\text{AUC}} \uparrow$ | $A_{last} \uparrow$ | $A_{\text{AUC}} \uparrow$ | $A_{last} \uparrow$ | $A_{\text{AUC}} \uparrow$ | $A_{last} \uparrow$ | $A_{\text{AUC}} \uparrow$ | $A_{last} \uparrow$ |
| C2C (CoLLAs 2024) | 47.29±2.75 | 39.23±3.78 | 28.33±1.93 | 20.77±1.51 | 35.06±0.41 | 27.81±0.15 | 11.89±0.22 | 8.82±0.08 |
| Glide-Syn (ICLR 2023) | 34.59±2.14 | 32.05±1.44 | 31.53±1.56 | 26.56±1.84 | 15.64±0.44 | 10.68±0.19 | 4.06±0.13 | 2.59±0.03 |
| Real-Fake (ICLR 2024) | 55.60±2.36 | 53.00±2.26 | 28.66±1.47 | 21.22±1.33 | 24.43±0.26 | 18.89±0.30 | 6.33±0.11 | 4.50±0.05 |
| IE (ICML 2023) | 47.29±3.29 | 38.99±2.94 | 25.74±2.11 | 18.23±1.87 | 34.76±0.52 | 27.55±0.24 | 11.92±0.26 | 8.50±0.14 |
| LE (ICLR 2023) | 46.47±2.00 | 45.76±2.33 | 32.42±1.35 | 27.56±0.66 | 20.01±0.27 | 15.38±0.31 | 6.40±0.13 | 4.59±0.09 |
| (+) CONAN | 49.37±3.77 | 50.45±1.56 | 33.88±1.79 | 30.29±0.81 | 30.80±0.63 | 25.33±0.20 | 9.54±0.25 | 7.59±0.17 |
| CHB (CVPR 2023) | 47.52±2.69 | 46.11±1.07 | 31.02±1.11 | 22.82±1.61 | 16.69±0.16 | 13.45±0.19 | 5.61±0.11 | 4.18±0.05 |
| (+) CONAN | 52.01±2.72 | 45.46±3.27 | 32.62±1.72 | 24.26±0.89 | 29.06±0.37 | 24.52±0.17 | 9.28±0.14 | 7.56±0.14 |
| SC (CVPR 2024) | 44.03±1.95 | 41.48±3.05 | 30.72±1.19 | 23.07±1.04 | 11.89±0.17 | 8.66±0.20 | 3.90±0.07 | 2.68±0.04 |
| (+) CONAN | 50.45±2.70 | 52.35±0.99 | 31.04±1.26 | 23.90±1.35 | 22.36±0.34 | 19.13±0.32 | 6.71±0.15 | 5.48±0.13 |
| CCG (arXiv 2024) | 45.49±2.81 | 45.29±1.69 | 30.20±1.91 | 23.44±0.71 | 12.55±0.22 | 10.21±0.26 | 4.03±0.10 | 2.91±0.10 |
| (+) CONAN | 46.65±3.36 | 45.75±1.92 | 31.14±1.88 | 25.77±1.18 | 18.32±0.42 | 15.83±0.34 | 5.78±0.17 | 4.70±0.14 |
| HIRPG | 51.36±2.59 | 51.63±2.49 | 34.12±1.27 | 28.18±1.32 | 27.72±0.30 | 23.71±0.39 | 10.70±0.19 | 8.75±0.13 |
| (+) CONAN (**Ours**) | 55.89±3.06 | 55.43±2.49 | **38.53±1.15** | **33.73±1.82** | 35.60±0.31 | 29.99±0.11 | **14.53±0.22** | **12.65±0.09** |
| MA | **67.10±4.07** | **61.95±0.92** | 27.75±1.44 | 20.90±0.95 | **51.13±0.28** | **42.95±0.15** | 13.48±0.09 | 10.69±0.07 |

Table 1: **Quantitative comparison between different name-only baselines on CIL setup.** We follow the ID and OOD domains as described in Tab. 6. MA refers to training a model with manually annotated data.

| Method | Bongard-HOI | | | | Bongard-OpenWorld | | | |
|---|---|---|---|---|---|---|---|---|
| | Positive / Negative | | Concept Answering | | Positive / Negative | | Concept Answering | |
| | $A_{\text{AUC}} \uparrow$ | $A_{last} \uparrow$ | $A_{\text{AUC}} \uparrow$ | $A_{last} \uparrow$ | $A_{\text{AUC}} \uparrow$ | $A_{last} \uparrow$ | $A_{\text{AUC}} \uparrow$ | $A_{last} \uparrow$ |
| C2C (CoLLAs 2024) | 61.53±3.13 | 59.58±2.49 | 73.88±3.21 | 67.40±3.15 | 49.75±0.49 | 50.39±0.89 | 69.56±3.58 | 67.56±1.47 |
| Glide-Syn (ICLR 2023) | 54.83±2.07 | 55.77±3.54 | 67.87±3.30 | 59.38±3.62 | - | - | - | - |
| LE (ICLR 2023) | 64.03±3.10 | 62.40±2.58 | 73.65±3.60 | 70.68±3.80 | - | - | - | - |
| (+) CONAN | 65.90±2.59 | 65.63±2.59 | 74.99±3.07 | 72.38±2.76 | - | - | - | - |
| HIRPG | 67.25±2.61 | 71.49±0.42 | 75.52±3.17 | 73.97±3.11 | 48.37±1.17 | 47.48±3.47 | 70.09±1.92 | 74.59±3.11 |
| (+) CONAN (Ours) | **70.20±3.97** | **73.18±2.40** | **77.01±3.45** | **75.80±1.83** | **53.68±1.18** | **57.74±2.18** | **73.10±3.79** | **76.77±3.81** |
| MA | 69.50±1.84 | 73.04±2.71 | 76.02±3.85 | 70.37±3.87 | 53.44±1.91 | 53.06±3.45 | 70.84±3.44 | 72.21±3.75 |

Table 2: **Quantitative comparison between different name-only baselines on Multi-modal MVCIL setup.**

long-context queries (*e.g.*, *'hard negative images of riding a bike'*) often retrieve noisy images, they require negative concepts derived from manually annotated data instead.

Even in the absence of hard negative concepts, GenCL outperforms models trained with both manually annotated and web-scraped data by leveraging the ability of LLMs to generate prompts for hard negative examples and the controllability of T2I generative models through text prompts (Nie et al., 2021). For the prompts we use to select hard negative examples, see Sec. A.4.

**Comparison of HIRPG with Diverse Prompt Generation Methods.**    To evaluate the effectiveness of HIRPG in diverse prompt generation, we compare models trained on data generated from prompts derived by prompt generation baselines (LE, CHB, SC, and CCG). As shown in Tab. 1, HIRPG significantly outperforms the baselines, both with and without the combination of CONAN. We attribute this performance improvement to two key components of the prompts: recurrent prompt generation (RPG), which reduces the overlap between generated prompts, and hierarchical generation (HIG), which addresses the *lost-in-the-middle* challenge (Liu et al., 2023c) that arises from solely using RPG. We provide an ablation study of these two components of HIRPG in Sec. A.9. Furthermore, we analyze the Diversity (Naeem et al., 2020) and Recognizability (Fan et al., 2024) of images generated by each prompt generation baseline in Sec. A.5 in the Appendix for the sake of space.

**Comparison of CONAN with Data Ensemble Methods.**    To demonstrate the effectiveness of CONAN, we compare it with existing data ensemble methods (*i.e.*, Moderate, Uncertainty, Glister, GradMatch, and LCMat), as well as the equal-weight selection (EWS) and No ensembling (*i.e.*, using images generated from a single generative model). For a fair comparison, we use the same candidate sets and ensure an equal number of selected images in the ensemble set across all ensemble methods. After data selection, we evaluate the performance of continual learners trained with each ensemble set and summarize the results in Tab. 3. We use a CLIP-pretrained ResNet-50 as the feature extractor for data ensembling, following  Cui et al. (2023), while employing ResNet-18 as the backbone network for the continual learner across all baselines. Note that Uncertainty, Glister, GradMatch, and LCMat require fine-tuning on the full dataset for gradient calculations of the fine-tuned model, even

though they use a pre-trained feature extractor for initialization. Consequently, for these baselines, we fine-tune the CLIP pre-trained ResNet-50 model for 30 epochs using the full dataset for those baselines. In contrast, Moderate and CONAN do not require fine-tuning; they only need a feature extractor to calculate distances. Despite being training-free, as shown in Tab. 3, CONAN outperforms methods that require fine-tuning with a full dataset, as well as moderate.

We further compare CONAN with various RMD-based ensemble, such as k-highest RMD ensemble, and summarize the results in Sec. A.14 for the space's sake.

| Method | Full Dataset Training | PACS | | | | DomainNet | | | |
|---|---|---|---|---|---|---|---|---|---|
| | | ID | | OOD | | ID | | OOD | |
| | | $A_{\text{AUC}}$ ↑ | $A_{last}$ ↑ | $A_{\text{AUC}}$ ↑ | $A_{last}$ ↑ | $A_{\text{AUC}}$ ↑ | $A_{last}$ ↑ | $A_{\text{AUC}}$ ↑ | $A_{last}$ ↑ |
| No ensembling | ✗ | 51.36±2.59 | 51.63±2.49 | 34.12±1.27 | 28.18±1.32 | 27.72±0.30 | 23.71±0.39 | 10.70±0.19 | 8.75±0.13 |
| EWS | ✗ | 50.56±2.32 | 50.03±2.13 | 34.59±1.41 | 27.13±3.44 | 32.38±0.47 | 26.45±0.35 | 12.93±0.23 | 10.92±0.06 |
| Moderate (ICLR 2023) | ✗ | 47.03±3.52 | 45.34±1.11 | 35.06±2.03 | 27.91±2.17 | 25.57±0.42 | 20.38±0.16 | 10.53±0.29 | 8.17±0.13 |
| CONAN (**Ours**) | ✗ | **55.89±3.06** | **55.43±2.49** | **38.53±1.15** | **33.73±1.82** | **34.60±0.31** | **30.09±0.11** | **14.53±0.22** | **12.65±0.09** |
| Uncertainty (ICLR 2020) | ✓ | 39.75±2.10 | 33.17±3.69 | 32.99±1.42 | 25.17±3.01 | 21.90/±0.37 | 15.70±0.08 | 10.01±0.23 | 7.19±0.11 |
| CRAIG (ICML 2020) | ✓ | 53.57±2.43 | 54.24±2.04 | 35.54±0.90 | 32.29±0.96 | 32.53±0.20 | 28.44±0.23 | 13.25±0.15 | 11.53±0.06 |
| Glister (AAAI 2021) | ✓ | 40.55±2.43 | 37.75±3.81 | 34.30±1.66 | 27.56±1.31 | 23.16±0.37 | 16.98±0.35 | 10.56±0.26 | 7.60±0.18 |
| GradMatch (ICML 2022) | ✓ | 54.93±3.24 | 54.06±1.49 | 35.05±1.70 | 29.81±1.35 | 32.53±0.43 | 28.36±0.41 | 13.48±0.31 | 11.74±0.18 |
| Adacore (ICML 2022) | ✓ | 52.06±2.64 | 48.37±2.80 | 35.55±2.09 | 30.36±0.85 | 32.15±0.55 | 26.83±0.18 | 13.62±0.27 | 11.37±0.04 |
| LCMat (AISTATS 2023) | ✓ | 53.40±2.35 | 54.60±1.65 | 35.37±1.62 | 30.04±0.82 | 32.38±0.44 | 28.36±0.32 | 13.42±0.26 | 11.76±0.17 |

Table 3: **Quantitative comparison between data ensemble methods on CIL setup.** EWS refers to the method of selecting and combining generated data from different generative models in equal proportions. No ensembling refers to using a single generative model (*i.e.*, SDXL).

Additionally, we provide a comparison of HIRPG and baselines in the joint training setup in Sec.A.7.

### 5.3 ABLATION STUDY

We conduct an ablation study on two components of GenCL, *i.e.*, HIRPG and CONAN using the ResNet-18 and ImageNet-1k pretrained ViT-Base models, and summarize the results in Tab. 4 and Tab. 5, respectively. Our observations indicate that both components play a significant role in enhancing both the ID domain performance and the OOD domain performance. In the tables, Vanilla GenCL refers to generating 50 different prompts using an LLM without employing RPG or HIG, and using a single T2I generator, *i.e.*, SDXL. We provide the details about Vanilla GenCL in Sec A.9.

| Method | PACS | | | | DomainNet | | | |
|---|---|---|---|---|---|---|---|---|
| | ID | | OOD | | ID | | OOD | |
| | $A_{\text{AUC}}$ ↑ | $A_{last}$ ↑ | $A_{\text{AUC}}$ ↑ | $A_{last}$ ↑ | $A_{\text{AUC}}$ ↑ | $A_{last}$ ↑ | $A_{\text{AUC}}$ ↑ | $A_{last}$ ↑ |
| Vanilla GenCL | 47.74±1.52 | 47.30±2.38 | 31.66±1.45 | 25.41±0.66 | 20.82±0.39 | 17.19±0.34 | 7.09±0.21 | 5.55±0.11 |
| (+) HIRPG | 51.36±2.59 | 51.63±2.49 | 34.12±1.27 | 28.18±1.32 | 27.72±0.30 | 23.71±0.39 | 10.70±0.19 | 8.75±0.13 |
| (+) CONAN | 50.02±2.52 | 45.34±4.25 | 33.94±1.37 | 27.30±1.16 | 28.17±0.35 | 24.12±0.11 | 9.76±0.17 | 8.18±0.15 |
| (+) HIRPG & CONAN (**Ours**) | **55.89±3.06** | **55.43±2.49** | **38.53±1.15** | **33.73±1.82** | **34.60±0.31** | **29.99±0.11** | **14.53±0.22** | **12.65±0.09** |

Table 4: **Ablations for proposed components of GenCL.** We use ResNet-18 model.

| Method | PACS | | | | DomainNet | | | |
|---|---|---|---|---|---|---|---|---|
| | ID | | OOD | | ID | | OOD | |
| | $A_{\text{AUC}}$ ↑ | $A_{last}$ ↑ | $A_{\text{AUC}}$ ↑ | $A_{last}$ ↑ | $A_{\text{AUC}}$ ↑ | $A_{last}$ ↑ | $A_{\text{AUC}}$ ↑ | $A_{last}$ ↑ |
| Vanilla GenCL | 72.91±1.40 | 56.85±2.68 | 40.39±1.67 | 27.11±2.90 | 30.96±0.34 | 22.52±0.46 | 11.17±0.25 | 7.78±0.21 |
| (+) HIRPG | 78.52±1.90 | 72.40±2.40 | 45.46±1.59 | 36.76±2.35 | 37.90±0.31 | 30.37±0.64 | 15.30±0.19 | 11.31±0.29 |
| (+) CONAN | 77.31±1.58 | 64.39±2.40 | 48.01±2.05 | 35.22±2.64 | 37.81±0.47 | 30.15±0.25 | 14.61±0.29 | 10.83±0.20 |
| (+) HIRPG & CONAN (**Ours**) | **79.32±1.97** | **72.46±0.42** | **53.88±1.57** | **41.31±2.42** | **42.73±0.25** | **36.09±0.50** | **18.64±0.28** | **14.68±0.16** |

Table 5: **Ablations for proposed components of GenCL.** We use ImageNet-1k pretrained ViT-base model.

## 6 CONCLUSION

Online continual learning represents a practical, real-world-aligned learning paradigm. However, the assumption of having access to well-curated and annotated data in these scenarios hinders its real-world application. To address the challenges arisen from using manually annotated and web-crawled data, we introduce a unified name-only continual learning framework that integrates generators with the continual learner, termed 'Generative name only Continual Learning' (**GenCL**).

Within the GenCL framework, we propose an diverse prompt generation method (*i.e.*, HIRPG) and complexity-guided ensembling (*i.e.*, CONAN). Extensive experimental validations demonstrate the performance improvements achieved by both components within the GenCL framework, showcasing its effectiveness in both ID and OOD settings compared to webly-supervised and human supervision.

ETHICS STATEMENT

We propose a better learning scheme for continual learning for realistic learning scenarios. While the authors do not explicitly aim for this, the increasing adoption of deep learning models in real-world contexts with streaming data could potentially raise concerns such as inadvertently introducing biases or discrimination. We note that we are committed to implementing all feasible precautions to avert such consequences, as they are unequivocally contrary to our intentions.

REPRODUCIBILITY STATEMENT

We take reproducibility in deep learning very seriously and highlight some of the contents of the manuscript that might help to reproduce our work. We provide a link in the abstract to access the generated data. Additionally, we will definitely release our implementation of the proposed method in Sec. 4, the data splits and the baselines used in our experiments in Sec. 5

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

# A APPENDIX

## A.1 DETAILS ABOUT VISUAL-CONCEPT INCREMENTAL LEARNING SETUP

Beyond CIL setups, we also assess GenCL in multimodal tasks, such as context-dependent visual reasoning tasks, focusing on Bongard-HOI (Jiang et al., 2022) and Bongard-OpenWorld (Wu et al., 2024a). These benchmarks are based on two desirable characteristics of classical Bongard problems: (1) few-shot concept learning and (2) context-dependent reasoning. The former refers to the ability to induce visual concepts from a small number of examples, while the latter indicates that the label of a query image may vary depending on the given context (*i.e.*, positive and negative support set). Specifically, as shown in Fig. 4 and Fig. 5, given a positive support set and a negative support set for a particular concept (*e.g.*, "ride a bike"), we consider two types of tasks that address the following queries: (1) *What is the concept exclusively depicted by the positive support set?* and (2) *Given a query image, does the query image belong to the positive or negative support set?* We refer to these tasks as CA (Concept Answering) and P/N, respectively. In addition, we provide a detailed description of each visual concept reasoning benchmark.

**Bongard-HOI (Jiang et al., 2022).** Bongard-HOI denotes a concept $c = \langle a, o \rangle$ as a visual relationship tuple, where $a$, $o$ are the class labels of action and object, respectively. Following Bongard's characteristic, there are positive support set $\mathcal{I}_p$ and negative support set $\mathcal{I}_n$, where $\mathcal{I}_p$ and $\mathcal{I}_n$ have different concepts. Specifically, if the concept of $\mathcal{I}_p$ is $\langle a, c \rangle$, $\mathcal{I}_n$ is composed of data with concept $c' = \langle \bar{a}, o \rangle$, where $\bar{a} \neq a$. As a result, images from both $\mathcal{I}_n$ and $\mathcal{I}_p$ contain the same categories of objects, with the only difference being the action labels, making it impossible to trivially distinguish positive images from negative ones through visual recognition of object categories alone (*i.e.*, hard negative examples). We provide examples of Bongard-HOI-CA & Bongard-HOI-P/N (Jiang et al., 2022) in Fig. 4.

**Bongard-OpenWorld (Wu et al., 2024a).** In contrast to Bongard-HOI, which has a structured concept $c$ represented as (action, object), Bongard-OpenWorld utilizes a free-form sentence as $c$ to describe the content depicted by all images in the positive set $\mathcal{I}_p$ exclusively. Specifically, concepts are obtained by the annotators, who are instructed to write visual concepts by following a predefined set of categories. We provide examples of Bongard-OpenWorld-CA & Bongard-OpenWorld-P/N (Wu et al., 2024a) in Fig. 5.

Note that since the input consists of both text queries and images (*i.e.*, support sets and a query image) and outputs are sentences, we use multimodal large language models (MLLMs), such as LLaVA (Liu et al., 2023b), which connects a vision encoder with an LLM for general-purpose visual and language understanding. For further implementation details, such as the prompts we use, see Sec. A.3.

## A.2 DETAILS ABOUT EXPERIMENT SETUP

To set a domain generalization benchmarks (*i.e.*, PACS (Zhou et al., 2020), DomainNet (Neyshabur et al., 2020), and CIFAR-10-W (Sun et al., 2024)) for a class incremental learning (CIL) setup, we divide it into multiple disjoint tasks. We assume a disjoint setup (Parisi et al., 2019), where tasks do not share classes. We summarize the in-distribution (ID) domain, the out-of-distribution (OOD) domains, the total number of classes per dataset, the number of classes per task, and the number of tasks for each dataset in Tab. 6. Within each dataset, all tasks have the same size, except PACS, which has a total of 7 classes. For PACS, the first task includes 3 classes, while the subsequent tasks include data for 2 classes each. For CIFAR-10-W, even though CIFAR-10 Krizhevsky et al. (2009) can use MA data, the image resolution of CIFAR-10 is 32×32, while CIFAR-10-W has a resolution of 224×224, leading to performance degradation. Therefore, we exclude comparison with MA in the CIFAR-10-W experiments. For multi-modal visual-concept incremental learning (MVCIL) setup, we summarize the the total number of concepts, number of tasks, and number of concepts per task in Tab. 7.

Note that we run five different task splits using five different random seeds and report the average and standard error of the mean (SEM) for all experiments.

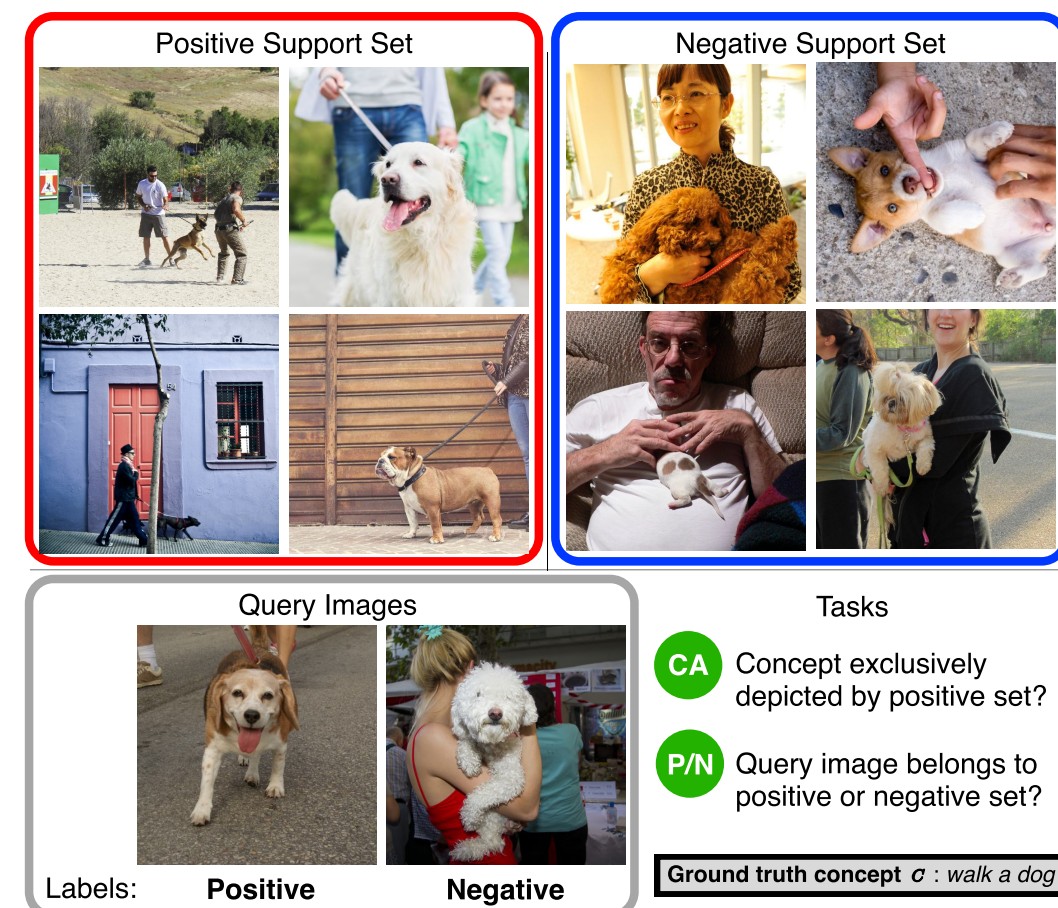

Figure 4: **An example of the Bongard-HOI task.** CA refers to the concept answering task, while P/N refers to the classifying whether a query image belongs to the positive or negative set.

| Dataset | ID domain | OOD domains | total # of classes | # of tasks | # of classes / task |
|---------|-----------|-------------|--------------------|-----------|---------------------|
| PACS | Photo | Art, Cartoon, Sketch | 7 | 3 | 2 (only initial task: 3) |
| DomainNet | Real | Clipart, Painting, Sketch | 345 | 5 | 69 |
| CIFAR-10-W | - | CIFAR-10-W | 10 | 5 | 2 |

Table 6: **Task configurations for the CIL setup on each domain generalization dataset.**

| Dataset | Form of Concepts | total # of concepts | # of tasks | # of concepts / task |
|---------|------------------|---------------------|-----------|----------------------|
| Bongard-OpenWorld | Free-form | 10 | 5 | 2 |
| Bongard-HOI | (action, object) | 50 | 5 | 10 |

Table 7: **Task configurations for the MVCI setup on each Bongard benchmark.**

### A.3 IMPLEMENTATION DETAILS

We used ResNet18 (He et al., 2016) and Vision Transformer (ViT) (Dosovitskiy & Brox, 2016) as network architectures for the class-incremental learning (CIL) setup. Due to the large number of parameters in ViT, training it from scratch in an online setup resulted in lower performance. Therefore, we used the weights of a model pre-trained on ImageNet-1K (Russakovsky et al., 2015) as initial weights for ViT. For data augmentation, we consistently applied RandAugment (Cubuk et al., 2020) in all experiments. For the optimizer and the learning rate (LR) scheduler in CIL setup, we employed the Adam optimizer with initial LR of 0.0003 and Constant LR scheduler, respectively, following prior works (Koh et al., 2023; Seo et al., 2024b). In MVCIL setup, we use Adam optimizer with LR $5 \times 10^{-5}$ and Constant LR scheduler. For task split, we adopt a disjoint setup, where tasks

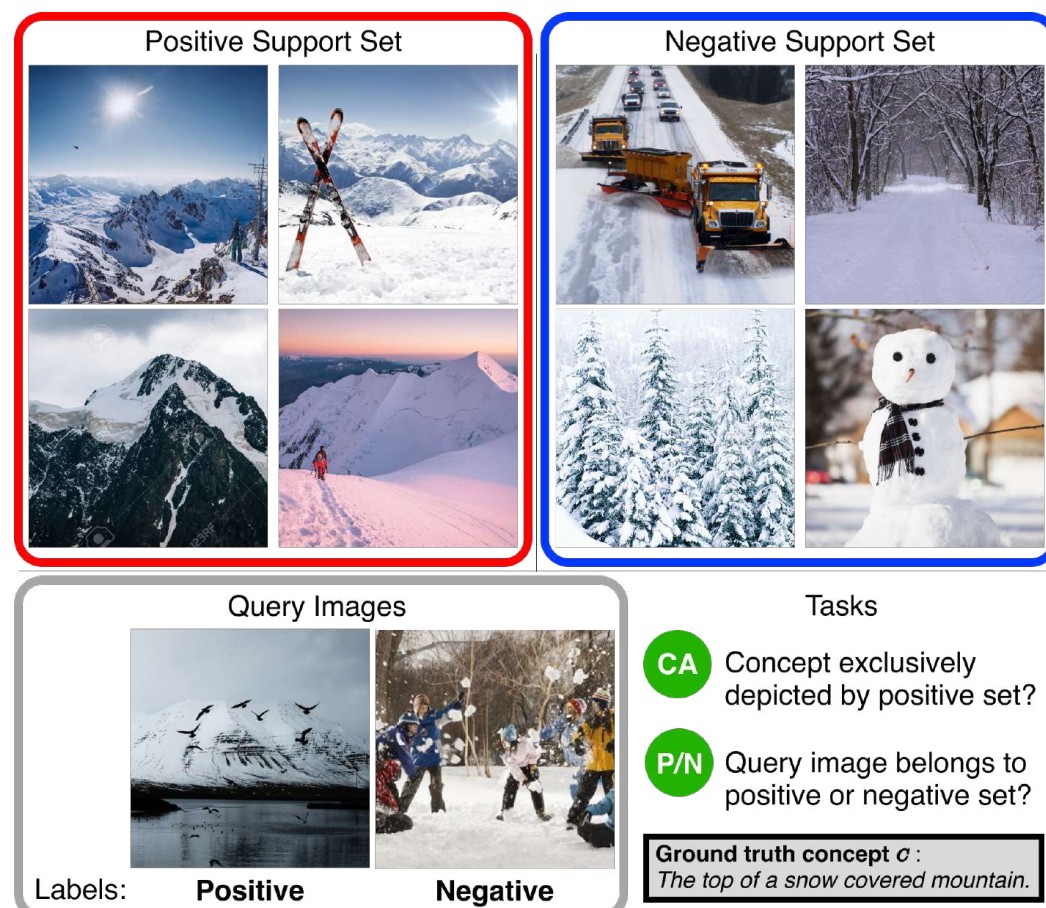

Figure 5: **An example of the Bongard-OpenWorld task.** CA refers to the concept answering task, while P/N refers to the classifying whether a query image belongs to the positive or negative set. The concept $c$ is free-form, such as sentences.

do not share classes (Parisi et al., 2019). We used the GPT-4 model (Achiam et al., 2023) for all LLM-based prompt generation baselines including HIRPG. To ensure a fair comparison among manually annotated data, generated data, and web-scraped data, we used an equal number of samples in all experiments. Regarding the web-scraped data, we obtained 20% more samples than necessary for batch training with the aim of filtering out noisy data. To achieve this, we utilized pre-trained CLIP (Radford et al., 2021) for filtering, which excludes the most noisy bottom samples, resulting in a cleaned subset used for training, following (Schuhmann et al., 2022). We used $8\times$RTX 4090 GPUs to generate images using text-to-image generative models.

**Hyperparameters.** For $T$, which refers to the temperature of the softmax function in CONAN, is uniformly set to 0.5 across all datasets. For $L$, the truncation ratio used in RMD score normalization, we set it to 5% for all experiments In all experiments, we run five different random seeds and report the average and standard error mean. For diverse prompt generation, we generate 50 different prompts for all baselines across all benchmarks, including HIRPG, to ensure a fair comparison. Specifically, to generate 50 prompts using HIRPG, we set depth $D = 2$, and $K = 7$ for all setups.

Following (Koh et al., 2021; 2023; Kim et al., 2024a), we conduct batch training for each incoming sample. Specifically, for PACS, CIFAR-10, and DomainNet, the number of batch iterations per incoming sample is set to 2, 2, and 3, respectively, with batch sizes of 16, 16, and 128. Episodic memory sizes are configured as 200, 2000, and 10000 for PACS, CIFAR-10-W, and DomainNet, respectively.

For MVCIL setups, the number of batch iterations per incoming sample is set to 0.5, with a batch size of 2, and a memory size of 500 in both Bongard-HOI and Bongard-OpenWorld. Unlike the CIL

setup, where data is composed solely of image and label pairs, in the MVCIL setup, each set contains both negative and positive examples corresponding to a given concept. We store 500 sets in episodic memory. In MVCIL benchmarks, *i.e.*, Bongard-HOI and Bongard-OpenWorld, we used 2 positive images and 2 negative images for a support set and 4 positive images and 4 negative images for a support set, respectively. For the MVCIL setup, we use the LLaVA-1.5-7B (Liu et al., 2023b).

**Prompts.** For the prompt diversification module $\psi$, we use the following system prompt to sequentially generate the prompts:

> To generate images using a text-to-image generation model, I need to create a prompt. Keep the domain photorealistic and use different visual scenes and visual styles or different color profiles/ palettes. Here is a list of prompts that I have previously generated <previous outputs>. Please create a new prompt that does not overlap with these.

In Bongard-HOI-P/N, we use the following prompt:

> 'positive' images:<|endofchunk|><image><image>
> 'negative' images:<|endofchunk|><image><image>
> 'query' image:<|endofchunk|><image>
> Given 2 'positive' images and 2 'negative' images, where both 'positive' and 'negative' images share a 'common' object, and only 'positive' images share a 'common' action whereas 'negative' images have different actions compared to the 'positive' images, the 'common' action is exclusively depicted by the 'positive' images. And then given 1 'query' image, please determine whether it belongs to 'positive' or 'negative' You must choose your answer from the Choice List. Choice list:[Positive, Negative].
> Your answer is:

In Bongard-HOI-CA, we use the following prompt:

> 'positive' images:<|endofchunk|><image><image>
> 'negative' images:<|endofchunk|><image><image>
> Given 2 'positive' images and 2 'negative' images, where both 'positive' and 'negative' images share a 'common' object, and only 'positive' images share a 'common' action whereas 'negative' images have different actions compared to the 'positive' images, the 'common' action is exclusively depicted by the 'positive' images. Your job is to find the 'common' action within the 'positive' images. You must choose your answer from the Choice List. Choice List: [choice lists].
> Your answer is:

In Bongard-OpenWorld-P/N, we use the following prompt:

> 'positive' images:<|endofchunk|><image><image><image><image>
> 'negative' images:<|endofchunk|><image><image><image><image>
> Given 4 'positive' images and 4 'negative' images, where 'positive' images share 'common' visual concepts and 'negative' images cannot, the 'common' visual concepts exclusively depicted by the 'positive' images. Here, 'common' sentence from 'positive' images is common concept. And then given 1 'query' image, please determine whether it belongs to 'positive' or 'negative'.

In Bongard-OpenWorld-CA, we use the following prompt:

> 'positive' images:<|endofchunk|><image><image><image><image>
> 'negative' images:<|endofchunk|><image><image><image><image>
> Given 4 'positive' images and 4 'negative' images, where 'positive' images can be summarized as 1 'common' sentence and 'negative' images cannot, the 'common' sentence describes a set of concepts that are common to 'positive' images. Please give the 'common' sentence from 'positive' images.

## A.4 SELECTING HARD NEGATIVE CONCEPTS IN GENCL ON MVCIL SETUPS

For Bongard-HOI benchmark, Given a (object, concept), *e.g.*, (ride, a bike), GenCL retrieves hard negative concept using an LLM and the following prompt:

> To train a model that distinguishes between positive and negative images, you need to choose $N$ negative actions from the following negative action list. When choosing negative actions, you should consider the available actions from the object. For example, if the object is 'bird', possible actions are 'chase', 'feed', 'no interaction', 'watch', etc. If the object is 'orange', possible actions are 'cut', 'hold', 'no interaction', 'peel', etc. You should choose hard negative actions that are clearly distinguishable from positive actions among the possible actions.
> object: <object class>
> positive action: <positive action>
> negative action list: <action set>
> Please select a total of $N$ negative actions. The response format must be strictly result: ['negative action1', 'negative action2', ... ], and all negative actions must be included in the negative action list.

For Bongard-OpenWorld benchmark, GenCL retrieves hard negative concept using an LLM and the following prompt:

> To create an image using a text-to-image generation model, I want to create a prompt. Below, a prompt for a positive image will be provided, and the goal is to generate a prompt for a negative image. It is important that the negative prompt partially overlaps with the positive prompt and has slight differences. For example, if the positive prompt is 'Dogs are running', then 'Dogs are drinking water' would be the negative prompt. Please create N 'negative' prompt sentences (under 5 words) that fits this description. Please ensure the response format is strictly 'prompt: answer'.
> Positive prompt: <positive prompt>.

## A.5 COMPARISON OF HIRPG WITH DIVERSE PROMPT GENERATION METHODS.

To evaluate the effectiveness of HIRPG in diverse prompt generation, we further analyze the generated images based on two attributes: Recognizability (Fan et al., 2024), which evaluates whether the images accurately represent the intended concepts, and Diversity (Naeem et al., 2020), which assesses the variation among the images. Although we aim to generate diverse images using varied prompts, the generated images should accurately represent the desired concepts. For a fair comparison, we generate 50 text prompts using each prompt diversification baseline and use SDXL to generate the same number of images for all baselines, including HIRPG. We summarize the results in Tab. 8.

| Method | PACS | | DomainNet | |
|---|---|---|---|---|
| | Recognizability ↑ | Diversity ↑ | Recognizability ↑ | Diversity ↑ |
| LE (He et al., 2023b) | 65.39 | 0.27 | 38.49 | 0.31 |
| CHB (Sarıyıldız et al., 2023) | 62.96 | 0.16 | 41.57 | 0.24 |
| SC (Tian et al., 2024a) | 71.50 | 0.19 | 33.19 | 0.20 |
| CCG (Hammoud et al., 2024) | 68.78 | 0.18 | 32.71 | 0.19 |
| HIRPG (Ours) | **90.77** | **0.31** | **52.83** | **0.35** |

Table 8: **Comparison of prompt diversification methods.** We compare the Recognizability and Diversity of images generated using text prompts derived from prompt generation methods in conjunction with a text-to-image generative model.

The model trained with data generated by HIRPG significantly outperforms those trained with data generated by the baselines in both the in-distribution (ID) and out-of-distribution (OOD) domains. Furthermore, as shown in the Recognizability and Diversity, HIRPG not only generates more diverse data, but also produces more recognizable data compared to baselines. Overall, DomainNet exhibits higher Diversity. This is because, despite having approximately 50 times more classes than PACS, it has fewer images per class, resulting in a smaller number of generated images per prompt. For detailed descriptions of the baselines and metrics (*i.e.*, Rec and Div), see Sec. A.10 and Sec. A.16, respectively.

### A.6 QUALITATIVE ANALYSIS

We qualitatively compare web-scraped images, manually annotated images, and GenCL-generated images, highlighting diversity and recognizability of GenCL-generated images.

**Multi-modal Visual-concept Incremental Learning Setup.** We compare samples acquired through different data acquisition methods for the given concept in the Bongard-HOI and Bongard-OpenWorld datasets, as shown in Fig.6 and Fig.7, respectively. In Fig. 6, although the desired positive images are related to 'ride a bike', the web-scraped positive set includes images of 'not ride a bike', such as 'sitting on a bike'. In addition, the positive set for 'riding a bike' contains even an image of a road with a bicycle symbol painted on it. These inherent noises in web-scraped data, *i.e.*, the inclusion of unwanted or irrelevant content, can significantly hinder model performance, as discussed in Sec. 1. In contrast, GenCL leverages the controllability (Nie et al., 2021) of the generative model, *i.e.*, the ability to generate the desired output through text descriptions, allowing it to produce the intended images.

Similarly, in Fig. 7, GenCL effectively generates both positive and negative support sets. In contrast, web-scraped data include images that do not match the given concept '*A leopard relaxing on a tree branch.*' This discrepancy arises from the lengthy and free-form concepts used in Bongard OpenWorld, such as descriptive sentences, compared to the simpler object-action combinations in Bongard-HOI. In web scraping, those detailed and lengthy search queries may yield unrelated results.

Note that GenCL relies solely on positive concepts, as mentioned in Sec. 5.2. Specifically, in manually annotated (MA) data, high-quality annotators not only select positive support sets but also curate hard negative examples for the negative sets. In contrast, GenCL utilizes only positive concepts (*i.e.*, concepts that the model needs to learn) and automatically generates hard negative examples using text prompts created by large language models (LLMs), as demonstrated in Sec. A.4. Nonetheless, as shown in Fig. 6 and Fig. 7, the negative samples generated by GenCL are not clearly distinct from the positive examples, which enhances the model's ability to differentiate between the concepts.

**Class Incremental Learning Setup.** We compare samples acquired through different baselines in the CIL setup, as illustrated in Fig. 8 and Fig. 9.

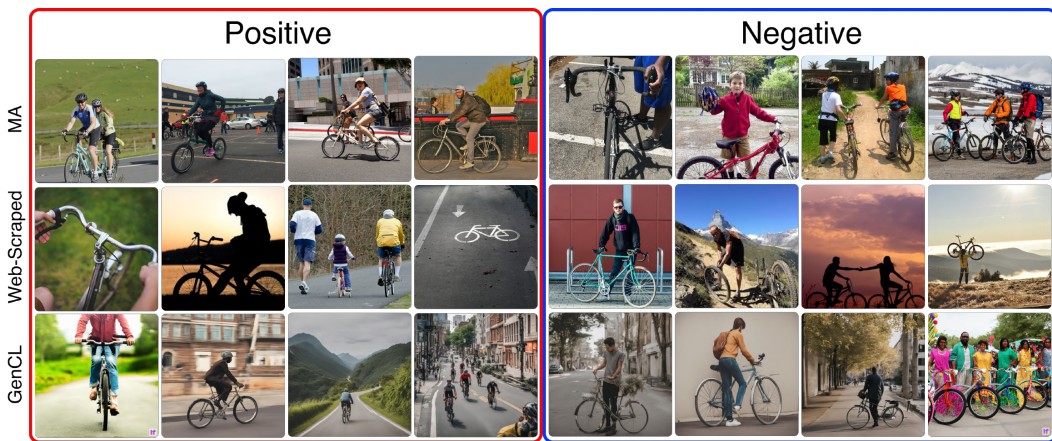

Figure 6: **Samples using different data acquisition methods for the same concept in the MVCIL setup.** The given concept is *'ride a bike'* from the Bongard-HOI benchmark. The left four images represent positive examples that depict the given concept, while the right four images represent negative examples that illustrate different concepts. Here, 'MA' refers to manually annotated data.

### A.7 EXPANDING GENCL TO THE JOINT TRAINING SETUP

We extend our proposed GenCL to the standard learning setup (*i.e.*, joint training setup), where all concepts to be learned are provided at once. In this setting, we compare GenCL not only with training-based methods, such as GLIDE, but also with training-free methods (*i.e.*, CLIP-ZS (Radford et al., 2021), SuS-X-SD (Udandarao et al., 2023), VisDesc (Menon & Vondrick, 2023), SD-Clf (Li et al., 2023a), and CUPL (Pratt et al., 2023)) that leverage pre-trained Vision-Language Models (VLMs), such as CLIP (Radford et al., 2021) or generative models, such as SDXL (Podell et al., 2023). Note that although these methods do not update model weights, they generate images for support sets or create customized prompts using LLMs to classify the target concept. We provide a detailed explanation of training-free baselines in Sec. A.12.

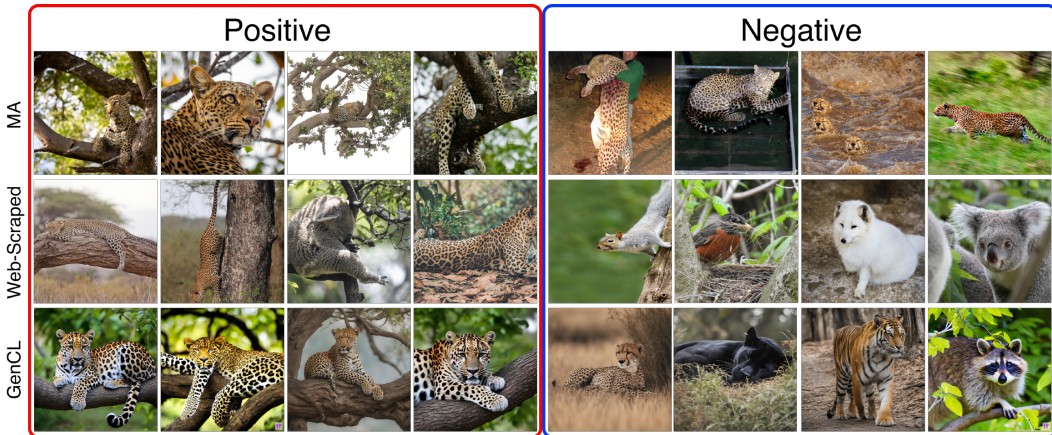

Figure 7: **Samples using different data acquisition methods for the same concept in the MVCIL setup.** The given concept is *'A leopard relaxing on a tree branch'* from the Bongard-OpenWorld benchmark. The left four images represent positive examples that depict the given concept, while the right four images represent negative examples that illustrate different concepts. Here, 'MA' refers to manually annotated data.

We first compare these methods using the same model, *i.e.*, ResNet-50-CLIP, a CLIP model with ResNet-50 as the vision encoder. For this, we utilize the YFCC100M (Thomee et al., 2016) pre-trained CLIP model. We summarize the results in Tab. 9. For training-dependent methods, we train the model for 10 epochs, ensuring the same amount of data is used across all baselines for a fair comparison. As shown in the table, GenCL significantly outperforms existing name-only classification baselines, as well as combinations of baselines with our proposed data ensemble method, *i.e.*, CONAN. Furthermore, compared to diverse prompt generation baselines (LE, CHB, SC, and CCG), our proposed HIRPG outperforms in both setups—with and without CONAN —demonstrating the effectiveness of our proposed components in a joint training setup.

Next, we compare the results with those obtained using only the CLIP-pretrained ResNet-50 for training-dependent methods. While the same model (*i.e.*, CLIP) can be employed for training-free methods, training vision-language models (VLMs) demands substantial computational resources, which impedes real-time adaptation and limits their deployment in real-world applications(Koh et al., 2021; Caccia et al., 2022). Therefore, to improve training efficiency and enable faster adaptation to newly encountered concepts, we also compare the results of training solely on the vision encoder of the CLIP model for training-dependent methods. To assess training efficiency, we train them for 10 epochs, consistent with Tab. 9, and summarize the results in Tab. 10.

As shown in the table, several training-free methods outperform GenCL in the in-domain (ID) scenario. This advantage arises because they utilize off-the-shelf CLIP models, which are pre-trained on large-scale datasets, particularly in the photo domain, which we consider as ID in our experiments. However, despite the benefits of large-scale pre-training, these methods struggle to generalize in out-of-domain (OOD) scenarios, such as the sketch and painting domains.

In contrast, GenCL not only outperforms all baselines but also surpasses a model trained with manually annotated data in the OOD domains of both PACS and DomainNet. This demonstrates that large-scale pre-training alone does not guarantee good generalization across all downstream tasks, highlighting the necessity of few-epoch training for personalization and real-time adaptation in name-only setup.

## A.8 ABLATION STUDY OF GENCL USING THE VIT

In addition to Sec. 5.3, we conduct an ablation study on two components of GenCL, namely HIRPG and CONAN, using the ImageNet-pretrained ViT-base model. We use the same number of images for each baseline to ensure a fair comparison, and summarize the results in Tab. 5.

Similar to the ablation study with ResNet-18, both components significantly enhance performance in both in-distribution (ID) and out-of-distribution (OOD) domains.

## A.9 ABLATION STUDY OF HIRPG

We conduct an ablation study on HIRPG to investigate the benefits of each proposed component, namely hierarchical generation (HIG) and recurrent prompt generation (RPG), in PACS and DomainNet. For a fair comparison, we generate 50 different prompts and use SDXL to generate images for all baselines. For HIG,

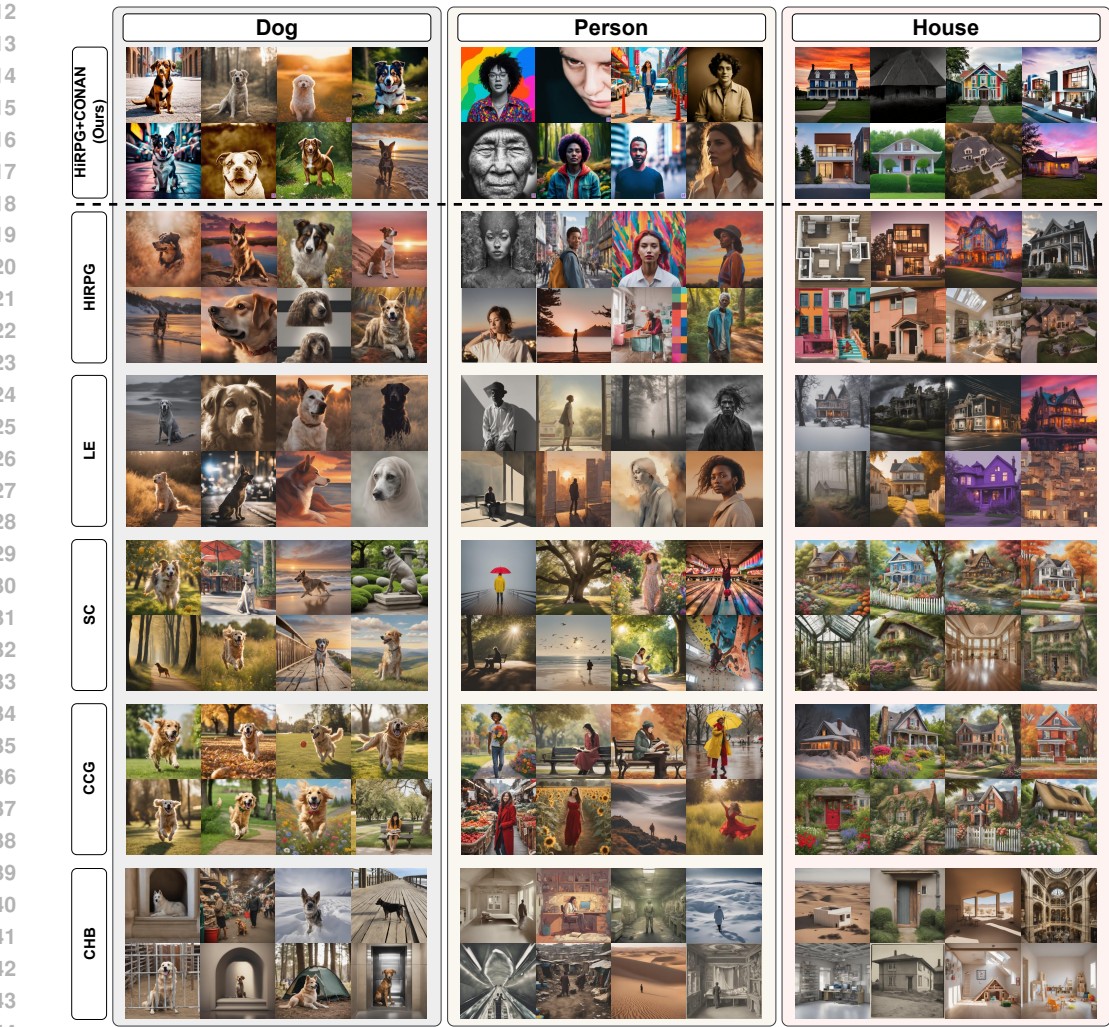

Figure 8: **Samples using different data acquisition methods for the same concept in the CIL setup.** The given concepts are *Dog, Person, House*, which are from PACS.

we use a 7-ary tree with a depth of 2. Since we only need 50 prompts, we sample 50 prompts from the 57 total nodes of a complete 7-ary tree.

The results are summarized in Tab. 11. In the table, vanila prompt generation refers to generating $N$ different prompts using an LLM without applying RPG or HIG. Specifically, we use the following prompts for vanila prompt generation:

> To generate images using a text-to-image generation model, I need to create 50 prompts. Keep the domain photorealistic and use different visual scenes and visual styles or different color profiles/palettes. Please create 50 prompts that does not overlap with each other. Please ensure that each response includes the word '[concept]'. For example, 'A photo of a [concept].', 'A detailed sketch of [concept].', 'A hyper-realistic portrait of [concept].', etc.

As shown in the table, applying RPG alone even degrades the performance compared to vanilla prompt generation. This degradation occurs because, as iterative steps progress, the length of the LLM input increases, making it challenging to utilize the information within the extended context effectively (*i.e.*, *lost-in-the-middle challenge* (Liu et al., 2023c; An et al., 2024)), as discussed in Sec. 4.1. In contrast, combining RPG with HIG addresses the lengthy input problem, leading to improved performance in both in-distribution (ID) and out-of-distribution on PACS and DomainNet.

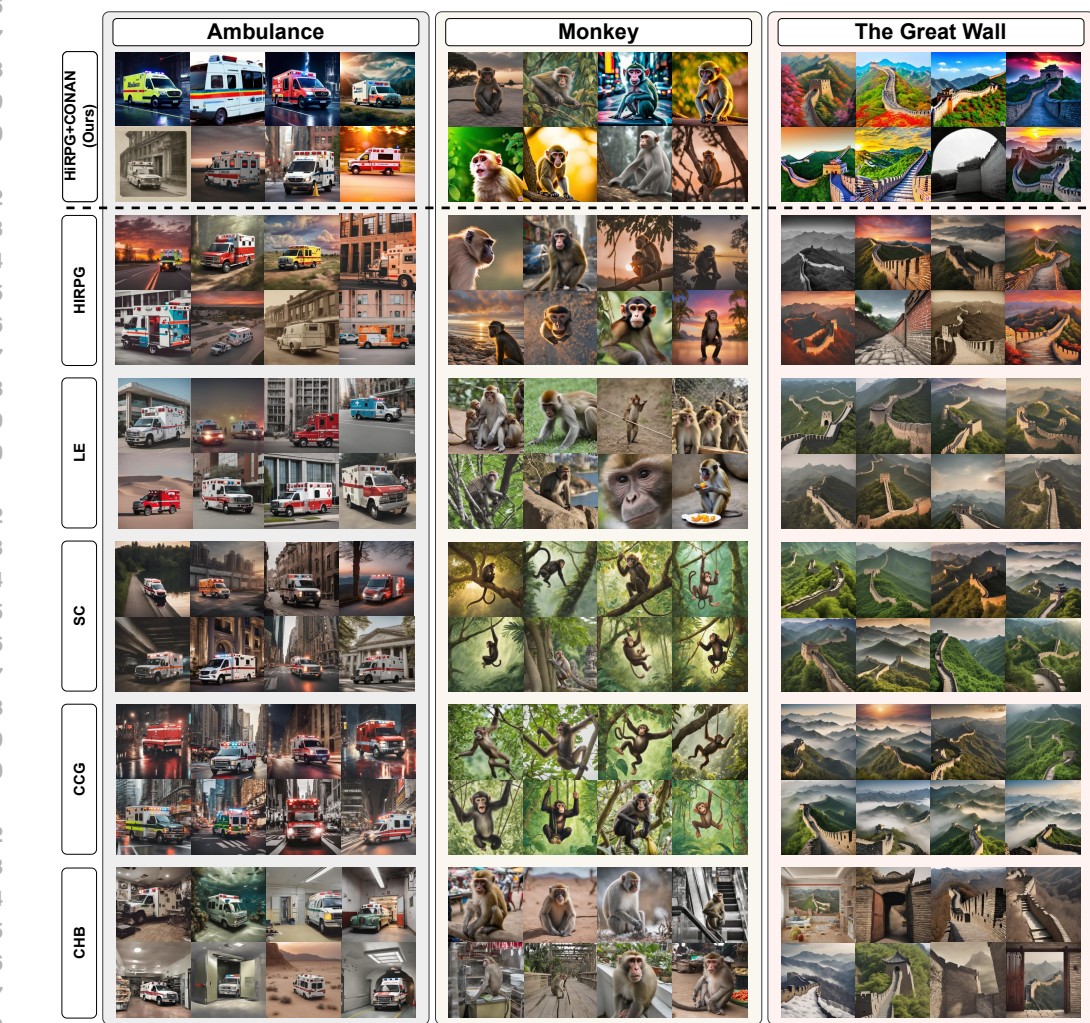

Figure 9: **Samples using different data acquisition methods for the same concept in the CIL setup.** The given concepts are *Ambulance, Monkey, The Great Wall*, which are from DomainNet.

### A.10 PROMPT DIVERSIFICATION AND CONCEPT-SPECIFIC PROMPT GENERATION BASELINES

For a fair comparison, we used the same number of diversified prompts (*i.e.*, 50) for all prompt diversification methods, including our proposed HCFG. Moreover, for all LLM-based prompt generators, we consistently used GPT-4o (Wu et al., 2024c) as LLM.

**LE (He et al., 2023b).** LE leverages an off-the-shelf word-to-sentence T5 model, pre-trained on the "Colossal Clean Crawled Corpus" dataset (Raffel et al., 2020) and fine-tuned on the CommonGen dataset (Lin et al., 2020), to increase the diversity of language prompts and generated images, with the aim of better harnessing the potential of synthesized data. Specifically, the category names are entered into the word-to-sentence model, which generates diversified sentences containing the category names. These diversified sentences are then used as prompts for the text-to-image generation process.

**CCG (Hammoud et al., 2024).** For a given concept set $C$, CCG (Concept-based Captions Generation) uses an LLM generator $G_{\text{LLM}}$ to produce concept-specific prompts for T2I models. The designed prompt $p$ for each concept $c \in C$ is structured as follows:

| Type | Training Data | CIFAR-10-W | DomainNet | |
|---|---|---|---|---|
| | | OOD | ID | OOD |
| Training-free | CLIP-ZS (Radford et al., 2021) | 57.14 | 14.69 | 5.17 |
| | SuS-X-SD (Udandarao et al., 2023) | 53.08 | 20.06 | 7.5 |
| | VisDesc (Menon & Vondrick, 2023) | 51.83 | 16.87 | 6.52 |
| | CuPL (Pratt et al., 2023) | 50.5 | 18.25 | 6.36 |
| | CALIP (Guo et al., 2023) | 51.62 | 16.43 | 6.39 |
| | SD-Clf (Li et al., 2023a) | 52.48 | 12.27 | 11.85 |
| Training-dependent | Glide-syn (He et al., 2023b) | 55.93 | 38.26 | 9.31 |
| | LE (He et al., 2023b) | 73.51 | 47.43 | 14.7 |
| | (+) CONAN | 75.13 | 52.87 | 17.26 |
| | CHB (Sarıyıldız et al., 2023) | 70.61 | 45.28 | 14.62 |
| | (+) CONAN | 75.96 | 52.31 | 17.49 |
| | SC (Tian et al., 2024a) | 71.3 | 40.42 | 12.36 |
| | (+) CONAN | 75.04 | 49.64 | 15.19 |
| | CCG (Hammoud et al., 2024) | 58.25 | 39.32 | 11.57 |
| | (+) CONAN | 63.14 | 42.94 | 14.37 |
| | HIRPG | 74.47 | 52.30 | 20.18 |
| | (+) CONAN (**Ours**) | **77.64** | 54.85 | **22.66** |
| | Manually Annotated | 59.12 | **71.13** | 20.29 |

Table 9: **Quantitative comparison between different name-only baselines on joint training setup.** We employ the YFCC100M pre-trained ResNet50-CLIP, which uses ResNet50 as the vision encoder for the CLIP model, for all methods.

| Type | Training Data | PACS | | DomainNet | |
|---|---|---|---|---|---|
| | | ID | OOD | ID | OOD |
| Training-free | CLIP-ZS (Radford et al., 2021) | **99.11** | 49.12 | 14.69 | 5.17 |
| | SuS-X-SD (Udandarao et al., 2023) | 95.55 | 47.81 | 20.06 | 7.5 |
| | VisDesc (Menon & Vondrick, 2023) | 93.77 | 46.09 | 16.87 | 6.52 |
| | CuPL (Pratt et al., 2023) | 89.32 | 46.51 | 18.25 | 6.36 |
| | CALIP (Guo et al., 2023) | 92.58 | 48.43 | 16.43 | 6.39 |
| | SD-Clf (Li et al., 2023a) | 92.58 | 48.43 | 12.27 | 11.85 |
| Training-dependent | Glide-syn (He et al., 2023b) | 85.16 | 33.2 | 29.02 | 6.73 |
| | LE (He et al., 2023b) | 88.43 | 38.03 | 40.74 | 10.47 |
| | (+) CONAN | 93.47 | 44.54 | 54.60 | 15.62 |
| | CHB (Sarıyıldız et al., 2023) | 83.38 | 30.98 | 35.97 | 9.60 |
| | (+) CONAN | 92.88 | 41.42 | 49.17 | 15.51 |
| | SC (Tian et al., 2024a) | 76.26 | 28.19 | 30.42 | 8.23 |
| | (+) CONAN | 85.46 | 42.05 | 44.66 | 12.01 |
| | CCG (Hammoud et al., 2024) | 81.01 | 31.71 | 26.59 | 6.89 |
| | (+) CONAN | 85.76 | 41.55 | 32.31 | 8.72 |
| | HIRPG | 89.91 | 43.98 | 46.19 | 17.80 |
| | (+) CONAN (**Ours**) | 94.36 | **60.75** | 51.85 | **21.01** |
| | Manually Annotated | 97.03 | 33.80 | **72.54** | 19.09 |

Table 10: **Quantitative comparison between different name-only baselines on joint training setup.** We employ the YFCC100M pre-trained ResNet50-CLIP for training-free methods, while for training-dependent methods, we utilize only the vision encoder of the CLIP model.

| Method | PACS | | | | DomainNet | | | |
|---|---|---|---|---|---|---|---|---|
| | ID | | OOD | | ID | | OOD | |
| | $A_{\text{AUC}}$ ↑ | $A_{last}$ ↑ | $A_{\text{AUC}}$ ↑ | $A_{last}$ ↑ | $A_{\text{AUC}}$ ↑ | $A_{last}$ ↑ | $A_{\text{AUC}}$ ↑ | $A_{last}$ ↑ |
| Vanila Prompt Generation | 47.74±1.52 | 47.30±2.38 | 31.66±1.45 | 25.41±0.66 | 20.82±0.39 | 17.19±0.34 | 7.09±0.21 | 5.55±0.11 |
| (+) RPG | 45.55±1.55 | 47.60±1.90 | 32.07±1.70 | 25.53±1.74 | 17.62±0.35 | 13.96±0.25 | 6.88±0.15 | 5.30±0.11 |
| (+) HIG + RPG (Ours) | **51.36±2.59** | **51.63±2.49** | **34.12±1.27** | **28.18±1.32** | **27.72±0.30** | **23.71±0.39** | **10.70±0.19** | **8.75±0.13** |

Table 11: **Benifits of components of the proposed prompt generation method**. RPG refers to the recurrent prompt generation and HIG refers to the hierarchical generation. Vanilla prompt generation refers to generating 50 different prompts using an LLM without incorporating RPG or HIG.

> Your task is to write me an image caption that includes and visually describes a scene around a concept. Your concept is $c$. Output one single grammatically correct caption that is no longer than 15 words. Do not output any notes, word counts, facts, etc. Output one single sentence only.

Formally, the set of generated captions for concept c can be defined as $T = \{t_{c,n} \sim G_{\text{LLM}}(p,c)\}, \forall c \in C, \forall n \in \{1, 2, ..., N\}$, where $N$ is the number of desired prompts for each concept.

**CHB (Sarıyıldız et al., 2023).** To increase the visual diversity of the output images, CHB (Concept Name + Hypernym + Background) combines background information along with hypernyms, which helps reduce semantic ambiguity. They assume that class $c$ can be seen 'inside' a scene or background. Therefore, to enhance visual diversity, CHB combines the concept name and its hypernym (as defined by the WordNet (Miller, 1995) graph) with scene classes from the Places365 dataset (López-Cifuentes et al., 2020) as background for each concept. Formally, $p_c$ can be defined as "$p_c = c, h_c$inside $b$", where $c$ refers to the concept name, $h_c$ refers to the hypernym of the concept $c$, and $b$ refers to the background.

**SC (Tian et al., 2024a).** SC (Synthesizing captions) consider three types of templates for each concept $c$: $c \to caption$, $(c, bg) \to caption$, and $(c, rel) \to caption$.

- $c \to caption$. They generate a sentence directly from the concept name $c$ using LLM.
- $(c, bg) \to caption$. Similar to CHB (Sarıyıldız et al., 2023), they combine the visual concept $c$ with a background $bg$. However, while CHB randomly selects $bg$ from the Places365 dataset, they generate a list of suitable backgrounds for the chosen concepts using LLM, which helps avoid unlikely combinations, such as a blue whale on a football field.
- $(c, rel) \to caption$. They consider pairing a given visual concept $c$ with a positional relationship word $rel$, such as "in front of." To add variety, $rel$ is randomly selected from a predefined set of 10 relationship words. Using an LLM, they then generate captions that reflect the selected relationship word in relation to the concept.

**Real-Fake (Yuan et al., 2024).** Real-Fake aligns both data and class-conditional distributions through Maximum Mean Discrepancy (MMD)-based loss (Gretton et al., 2006) to minimize the discrepancy between real and synthetic data distributions. For prompt generation, it leverages BLIP2 (Li et al.) to generate captions that incorporate class-relevant information. These captions are combined with class names and intra-class visual features to guide the generation process.

**IE (Li et al., 2023b).** IE (Internet Explorer) dynamically queries search engines by combining concepts from the WordNet hierarchy (Miller, 1995) with descriptors generated by GPT-J (Wang & Komatsuzaki, 2021). For instance, the concept 'duck' can be combined with descriptors like 'baby', or 'red', resulting in queries such as 'baby duck' or 'red duck'. The descriptors are generated using a prompt template like "The {concept} is [descriptor]" and sampled with temperatures to ensure diversity. The retrieved images are filtered based on their similarity to the target dataset using a relevance reward metric.

## A.11 Data Ensemble Baselines

Gradient-based methods (Killamsetty et al., 2021b;a; Shin et al., 2023) minimize the distance between the gradients from the entire dataset $T$ and from the selected coreset $S$ ($S \subset T$) as follows:

$$\min_{\mathbf{w}, S} \left\| \sum_{(x,y) \in T} \frac{\nabla_\theta l(x,y;\theta)}{|T|} - \sum_{(x,y) \in S} \frac{w_x \nabla_\theta l(x,y;\theta)}{\|\mathbf{w}\|_1} \right\|_2, \tag{5}$$

where $\mathbf{w}$ is the vector of learnable weights for the data in selected coreset $S$, $l$ refers to the loss function, $\theta$ denotes the model parameters, and $\| \|_1, \| \|_2$ represent the L1 norm and L2 norm, respectively. To solve Eq. 5, **GradMatch** (Killamsetty et al., 2021a) uses orthogonal matching pursuit algorithm (Elenberg et al., 2016), while **CRAIG** (Mirzasoleiman et al., 2020) uses submodular maximization.

**LCMat (Shin et al., 2023).** While Craig and GradMatch minimize the gradient difference between $T$ and the $S$, LCMat matches the loss curvatures of the $T$ and $S$ over the model parameter space, inspired by the observation that a loss function $L$ quantifies the fitness of the model parameters $\theta$ under a specific dataset. Specifically, they claim that even though optimizing $S$ toward $T$ with respect to $\theta$ would decrease the loss difference between $T$ and $S$ in $\theta$ (*i.e.*, $|L(T;\theta) - L(S;\theta)|$), if the loss difference increases with a small perturbation $\epsilon$ in $\theta$ (*i.e.*, $|L(T;\theta + \epsilon) - L(S;\theta + \epsilon)|$), it indicates a lack of generalization on $\theta + \epsilon$, or an over-fitted reduction of $S$ by $\theta$. Since this generalization failure on the locality of $\theta$ subsequently results in the large difference of loss surfaces between $T$ and $S$, they propose an objective that maximize the robustness of $\theta$ under perturbation $\epsilon$ as follows:

$$\min_S(L_{abs}(T, S; \theta + \epsilon) - L_{abs}(T, S; \theta)), \tag{6}$$

where $L_{abs}$ refers to the loss difference between T and S on $\theta$ (*i.e.*, $L(T;\theta) - L(S;\theta)$).

**Moderate (Xia et al., 2023).** Moderate selects data points with scores close to the score median, using the median as a proxy for the score distribution in statistics.

Specifically, given a well-trained feature extractor $f(\cdot)$ and the full training data $T = \{t_1, t_2, \ldots, t_n\}$, the process begins by computing the hidden representations (or embeddings) of all data points in $T$, *i.e.*, $\{z_1 = f(t_1), z_2 = f(t_2), \ldots, z_n = f(t_n)\}$. Next, the $\ell_2$ distance from the hidden representation of each data point to the class prototype of its corresponding class is calculated. Formally, for a sample $t$ belonging to class $c$, its distance $d(t)$ is given by $d(t) = \|z - z^c\|_2$, where $z = f(t)$ and $z^c$ is the prototype of class $c$. Subsequently, all data points are sorted in ascending order according to their distance, which are denoted by $\{d(\tilde{t}_1), d(\tilde{t}_2), \ldots, d(\tilde{t}_n)\}$. Finally, data points near the distance median are selected as the coreset $S$.

**Uncertainty (Coleman et al., 2020).** Uncertainty suggests that data samples with a lower level of confidence in model predictions will have a greater influence on the formation of the decision boundary. For uncertainty scores, we utilize Entropy, following the approach of Shin et al. (2023), among the methods LeastConfidence, Entropy, and Margin.

**Glister (Killamsetty et al., 2021b).** Glister is a generalization-based data selection method that optimizes generalization error via a bi-level optimization problem to select the coreset $S$, aiming to maximize the log-likelihood on a held-out validation set.

## A.12 DESCRIPTION OF NAME-ONLY CLASSIFICATION BASELINES.

**Glide-Syn (He et al., 2023b).** This approach takes *category name* as input and employs the word-to-sentence T5 model (pre-trained on 'Colossal Clean Crawled Corpus' dataset (Raffel et al., 2020) and finetuned on 'CommonGen' dataset (Lin et al., 2019)), to generate diverse concept-specific sentences. After generating diverse sentences using the word-to-sentence T5 model, they generate corresponding images using prompts and the Glide (Nichol et al., 2021) text-to-image generative model. Finally, they introduce a clip filter to reduce noise and enhance robustness.

**CLIP-ZS (Radford et al., 2021).** CLIP-ZS refers to zero-shot classification using a pre-trained CLIP model, where the model classifies images without any additional training, leveraging its knowledge from large-scale pre-training. Since CLIP is pre-trained on large-scale web dataset, it demonstrates impressive zero-shot performance (Qian et al., 2024).

**SuS-X-SD (Udandarao et al., 2023).** This approach uses generated SuS (Support Sets) to ensure accurate predictions for target categories by taking only categories as input. Specifically, SuS-X-SD generates support sets using Stable Diffusion (Podell et al., 2023) and uses them as a combination with the pre-trained vision language model and an adapter module named TiP-X for inference.

**CuPL (Pratt et al., 2023).** While the standard zero-shot open vocabulary image classification model, *e.g.*, CLIP (Radford et al., 2021), uses only the set of base prompts, *i.e.*, 'A photo of {*category name*}', and target images for classification, CuPL proposes to use customized prompts using LLM. Specifically, they propose using GPT-3 (Brown et al., 2020), but we replace it with GPT-4o (Wu et al., 2024c), which is a stronger LLM and the one used in our proposed GenCL, for a fair comparison.

**CALIP (Guo et al., 2023).** CALIP enhances the zero-shot performance of CLIP (Radford et al., 2021) by introducing a parameter-free attention module. This module enables visual and textual representations to interact and explore cross-modal informative features via attention. As a result, image representations are enriched with textual-aware signals, and text representations are guided by visual features, leading to better adaptive zero-shot alignment.

**SD-Clf (Li et al., 2023a).** SD-Clf leverages large-scale text-to-image diffusion models, such as Stable Diffusion (Podell et al., 2023), for classification tasks. Given an input $x$ and a finite set of classes $C$, the model computes the class-conditional likelihoods $p_\theta(x|c)$. By selecting an appropriate prior distribution $p(c)$ and applying Bayes' theorem, SD-Clf predicts class probabilities $p(c|x)$, effectively classifying the input based on the computed likelihoods.

## A.13 JUSTIFICATION FOR THE USE OF RMD SCORE

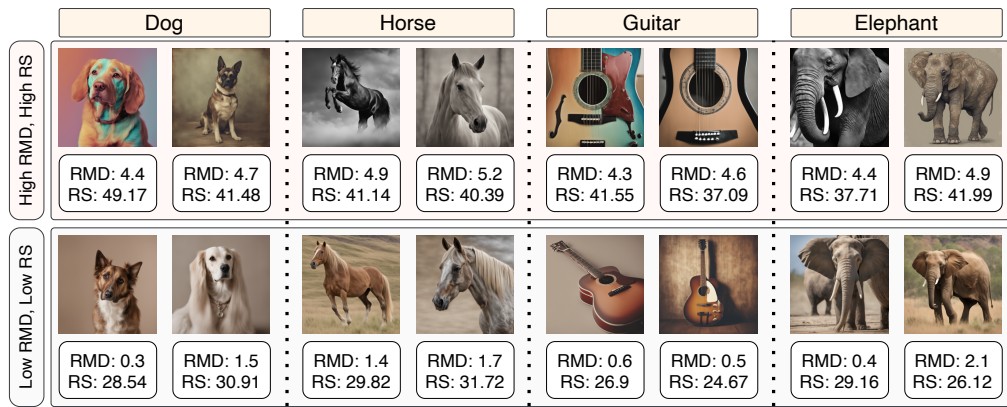

Figure 10: Examples of samples with high RMD & high Rarity scores, as well as samples with low RMD & low Rarity scores. The average RMD scores for Dog, Horse, Guitar, and Elephant are 2.91, 3.03, 2.43, and 3.25, respectively, while the corresponding average Rarity scores are 33.59, 33.58, 33.57, and 33.18.

Many recent works endeavor to assess the diversity (Naeem et al., 2020; Han et al., 2022; Kim et al., 2024b), complexity (Hwang et al., 2023), aesthetics (Somepalli et al., 2024; Khajehabdollahi et al., 2019), and realism (Chen et al., 2023a; 2024; 2023b) of the generated images. In our work, we use the relative Mahalanobis distance (RMD) score (Cui et al., 2023), to evaluate the complexity of the generated samples. The reason for selecting RMD is its independence from the need for real samples in its calculation, while other diversity metric, such as the Rarity Score (Han et al., 2022) and the TopP&R (Kim et al., 2024b) requires *real* samples, *i.e.*, data that have not been generated. Note that our proposed framework, GenCL, operates exclusively with *concept* inputs rather than *real* data, thus we cannot access *real* data.

As we can see in Fig. 10, the Rarity score and the RMD score exhibit similar trends, showing the ability of the RMD score to effectively measure complexity even in the absence of real samples.

## A.14 COMPARISON BETWEEN CONAN AND VARIOUS RMD-BASED ENSEMBLE

We compare CONAN with various RMD-based ensemble approaches in PACS, and summarize the result in Tab. 12. The table reveals that CONAN significantly outperforms others in both In-Distribution (ID) and Out-of-Distribution (OOD) evaluations. Furthermore, with the exception of **CONAN**, all ensemble methods even exhibit a decrease in performance compared to the scenario where no ensembling[1] is applied. The $k$-highest RMD ensemble, which excludes easy samples, leads to insufficient learning in the class-representative region, while the $k$-lowest RMD concentrates solely on easy samples, resulting in limited diversity. Inverse CONAN employs the inverse of the probabilities utilized in CONAN. Similar to the $k$-lowest RMD ensemble, it tends to prioritize easy samples, resulting in limited diversity and accuracy loss.

## A.15 DETAILS ABOUT DETERMINING ID AND OOD DOMAINS

To compare with the model trained with GenCL and the model trained with manually annotated (MA) data, we select one domain as MA data from each domain generalization benchmark.

---

[1]*No ensembling* denotes the usage of images generated exclusively through SDXL (Podell et al., 2023).

| Ensemble Method $\Delta$ | ID | | OOD | |
|---|---|---|---|---|
| | $A_{\text{AUC}}$ ↑ | $A_{last}$ ↑ | $A_{\text{AUC}}$ ↑ | $A_{last}$ ↑ |
| No ensembling | 51.36±2.59 | 51.63±2.49 | 34.12±1.27 | 27.18±1.32 |
| Equal weight ensemble | 50.56±2.32 | 50.03±2.13 | 35.49±1.41 | 27.13±3.44 |
| $k$-highest RMD ensemble | 52.80±2.82 | 50.09±3.06 | 36.24±1.52 | 30.09±1.35 |
| $k$-lowest RMD ensemble | 41.72±2.88 | 37.98±2.19 | 31.17±2.34 | 24.58±2.49 |
| Inverse Prob | 45.01±3.03 | 38.70±4.12 | 32.94±1.62 | 27.61±1.97 |
| CONAN (Ours) | **55.89±3.06** | **55.43±2.49** | **38.53±1.15** | **33.73±1.82** |

Table 12: Comparison of ensemble methods in PACS (Zhou et al., 2020), using ER (Rolnick et al., 2019) for all ensemble methods. The proposed ensemble method outperforms other ensemble methods. We used Photo domain as the ID domain, whereas we used Art, Cartoon, and Sketch domain as OOD domains. For OOD domains, $A_{\text{AUC}}$ and $A_{last}$ refer to the average value across all OOD domains.

## A.16 DETAILS ABOUT METRICS

**Area Under the Curve of Accuracy ($A_{\text{AUC}}$).** In online CL setup, the model is trained using the stream data in real time, thus the model is used for inference at every moment rather than the predefined time point (*e.g.*, end of the task) (Koh et al., 2021; Caccia et al., 2022; Banerjee et al., 2023; Pellegrini et al., 2020). Therefore, to measure inference performance at any time, we evaluated the model at regular intervals during a specified evaluation period and then calculated the area under the accuracy curve, denoted $A_{\text{AUC}}$ (Koh et al., 2021; Caccia et al., 2022; Koh et al., 2023), which is defined as follows:

$$A_{\text{AUC}} = \sum_{i=1}^{k} f(i\Delta n)\Delta n, \qquad (7)$$

where the step size $\Delta n$ is the number of samples encountered between inference queries and $f(\cdot)$ is the accuracy in the curve of the # of samples-to-accuracy plot. High $A_{\text{AUC}}$ indicates that the model maintains good inference performance throughout the entire training process.

**Recognizability.** Following Boutin et al. (2022); Fan et al. (2024), we evaluate whether the images accurately represent the intended concepts by computing the F1 score for each class. As previous work utilized a pre-trained classifier (ViT-Base), we initialize the feature extractor with an ImageNet pre-trained ViT-Base. We then perform linear probing (Alain, 2016) on this model with a downstream dataset to train a classification head for the dataset. Recognizability is then calculated by averaging the F1 scores across all classes.

**Diversity.** To assess the diversity of generated images, Naeem et al. (2020) measures coverage, defined as the ratio of real samples encompassed by the generated samples. Specifically, they calculate the fraction of real samples whose $k$-nearest neighborhoods contain at least one generated sample. Formally, given the embedded real and generated data, represented by $\{X_i\}$ and $\{Y_j\}$ from an ImageNet pre-trained feature extractor, coverage is defined as:

$$\text{coverage} := \frac{1}{N} \sum_{i=1}^{N} \mathbb{1}_{\exists j \text{ s.t. } Y_j \in B(X_i, \text{NND}_k(X_i))}, \qquad (8)$$

where $\text{NND}_k(X_i)$ denotes the distance from $X_i$ to the $k^{th}$ nearest neighboring among $\{X_i\}$ excluding itself and $B(x, r)$ denotes the sphere in $\mathbb{R}^D$ around $x$ with radius $r$.

**Consensus-based Image Description Evaluation (CIDEr).** CIDEr (Vedantam et al., 2015) aims to automatically evaluate how well a predicted sentence, $s_p$, matches the consensus of a set of ground-truth sentences, $S = \{s_{gt,1}, \ldots, s_{gt,N}\}$. The intuition is that the measure of consensus should encode how often n-grams from the candidate sentence appear in the reference sentences. In contrast, $n$-grams that are absent from the reference sentences should not appear in the candidate sentence. To encode this intuition, they calculate the TF-IDF (Robertson, 2004) vectors for the $n$-gram elements within the candidate and reference sentences by computing the cosine similarity between the two vectors. Formally, CIDEr for $n$-grams is calculated as follows:

$$\text{CIDEr}_n(s_p, s_{gt}) = \frac{g^n(s_p) \cdot g^n(s_{gt})}{\|g^n(s_p)\|\|g^n(s_{gt})\|}, \qquad (9)$$

where $g(s)$ represents the vectorized form of a sentence $s$, obtained by calculating the TF-IDF values for its $n$-gram elements. Finally, they combine the scores from $n$-grams of varying lengths as follows:

$$\text{CIDEr} = \frac{1}{N} \sum_{i=1}^{N} \text{CIDEr}_n. \qquad (10)$$

Following Vedantam et al. (2015), we use $N = 4$ and define the set of ground truth sentences in the positive set as $S$.

### A.17 DETAILS ABOUT GENERATORS $\mathcal{G}$

For the set of generators $\mathcal{G}$, we use five text-to-image generative models: SDXL (Podell et al., 2023), DeepFloyd IF[2], SD3 (Esser et al., 2024), CogView2 (Ding et al., 2022), and Auraflow[3]. As illustrated in Figure 11, different generators produce varied samples when prompted with identical prompts conditioned on the same concept.

Details of each generator are as follows:

**SDXL.** SDXL is an enhanced latent diffusion model for text-to-image synthesis, building upon the previous versions of Stable Diffusion. Specifically, SDXL introduces three key improvements: (1) a UNet (Ronneberger et al., 2015) backbone that is 3× larger than in previous Stable Diffusion models, (2) an additional conditioning technique, and (3) a diffusion-based refinement model to further improve the visual quality of generated images.

**DeepFloyd IF.** DeepFloyd IF utilizes a frozen text encoder alongside three cascaded pixel diffusion stages. Initially, the base model produces a 64x64 image from a text prompt, which is then progressively enhanced by two super-resolution models to reach 256x256 and ultimately 1024x1024 pixels. At every stage, the model uses a frozen T5 transformer-based text encoder to derive text embeddings, which are then passed into a UNet.

**CogView2.** CogView2 pretrain a 6B-parameter transformer using a straightforward and adaptable self-supervised task, resulting in a cross-modal general language model (CogLM). This model is then fine-tuned for efficient super-resolution tasks. The hierarchical generation process is composed of three steps: (1) A batch of low-resolution images ($20 \times 20$ tokens) is first generated using the pretrained CogLM. Optionally, poor-quality samples can be filtered out based on the perplexity of CogLM image captioning, following the post-selection method introduced in CogView (Ding et al., 2021). (2) These generated images are then upscaled to $60 \times 60$-token images via a direct super-resolution module fine-tuned from the pretrained CogLM. (3) Finally, these high-resolution images are refined through another iterative super-resolution module fine-tuned from CogLM.

**SD3.** SD3 enhances current noise sampling methods for training rectified flow models (Liu et al., 2023d) by steering them toward perceptually significant scales. In addition, SD3 introduces a new transformer-based architecture for text-to-image generation, employing distinct weights for the two modalities. This design facilitates a bidirectional flow of information between image and text tokens, leading to improved typography, text comprehension, and higher human preference ratings.

**AuraFlow.** AuraFlow, inspired by SD3, is currently the largest text-to-image generation model. It introduces several modifications to SD3, including the removal of most MMDiT blocks (Esser et al., 2024) and their replacement with larger DiT encoder blocks (Peebles & Xie, 2023).

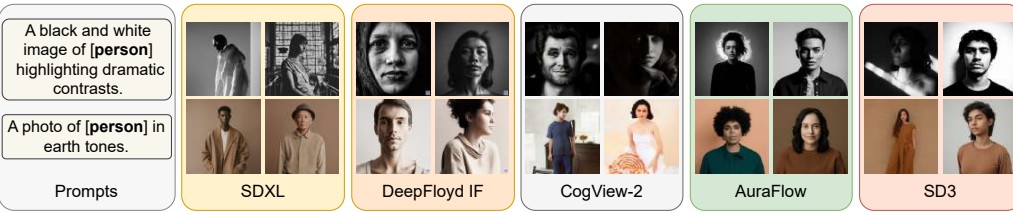

Figure 11: Examples of PACS (Zhou et al., 2020) generated samples from various generators using two of the prompt rewrites. Illustrations from the concept "Person" are showcased.

### A.18 EXTENDED RELATED WORK

**Methods for Continual Learning.** Replay-based method, which stores data from previous tasks in episodic memory for replay, is one of the most widely used approaches, due to its effectiveness in preventing catastrophic forgetting (Zhang et al., 2023b; Yoo et al., 2024; Kozal et al., 2024). However, despite their effectiveness in preventing forgetting, they raise data privacy concerns due to the storage of real data from previous tasks in episodic memory. To address these privacy concerns, pseudo-replay approaches have been proposed (Graffieti et al., 2023; Thandiackal et al., 2021; Van de Ven et al., 2020; Shin et al., 2017; Van de Ven & Tolias, 2018),

---

[2] https://github.com/deep-floyd/IF
[3] https://huggingface.co/fal/AuraFlow

which leverage generative models to generate images of previous tasks instead of storing actual data in episodic memory. While these approaches utilize generative models similar to our GenCL framework, they still require manually annotated data to train the generative model (Shin et al., 2017; Van de Ven et al., 2020; Van de Ven & Tolias, 2018). In contrast, our GenCL framework eliminates the need for any manually annotated data, relying solely on the category names the model aims to learn.

### A.19    MANUAL ANNOTATION VS. WEB-SCRAPING VS. GENERATIVE DATA

In modern deep learning, the trajectory of advancement is heavily influenced by the exponential growth of training data and the corresponding models trained on these vast datasets. Foundation models are typically exposed to datasets in the order of billions during training, obtained predominantly through web scraping (Schuhmann et al., 2022; Xue et al., 2020; Zhu et al., 2024; Gao et al., 2020; Kocetkov et al., 2022; Bain et al., 2021). Although web scraping is a cost-effective method to produce high-quality datasets, studies underscore issues such as potential biases (Foerderer, 2023; Packer et al., 2018; Caliskan et al., 2017), copyright, privacy, and license concerns (Quang, 2021; Solon, 2019), and the risks of data contamination (Dekoninck et al., 2024; Li, 2023) or data leakage from evaluation (Balloccu et al., 2024).

As demonstrated in Fig. 1, we highlight the key differences between Manually Annotated (MA), Web scraped, and Generated data on six different axes: (a) Controllability, (b) Storage issues, (c) Usage restrictions, (d) Privacy issues, (e) Acquisition cost, and (f) Noise. In this section, we aim to provide the definition of each of these axes and their corresponding implications on each type of data source.

**Controllability.**    encompasses the ability to generate or acquire images with various contexts, backgrounds, settings, and themes as desired. It pertains to the ability to obtain images depicting different concepts in compositions not commonly found in natural environments, as well as in domains relevant to the task at hand. Under this definition, we assert that the MA data exhibit low controllability. This limitation arises from its reliance on data captured from a finite set of scenarios or sensors, which inherently restricts the breadth of diverse settings where the concept can be observed. Web-scrapped data also suffer from low controllability for the same reasons. In contrast, the generated data have high controllability due to the ability of foundation text-to-image (T2I) generators to produce diverse images for each concept through varied prompting.

**Storage Issues.**    Storing extensive data, locally or in the cloud, imposes additional costs, which can become impractical in environments constrained by limited total storage capacity. In addition, transmitting large, substantial data samples in a federated setup can face challenges arising from bandwidth and latency bottlenecks. In such contexts, depending on a large corpus of MA data becomes counterintuitive. On the other hand, both web-scraped and generated data present themselves as cost-effective alternatives for accessing substantial data volumes without necessitating explicit storage expenditures.

**Usage Restrictions.**    encompass limitations imposed on the use of images for training machine/deep learning models, typically due to copyright or licensing protections. These restrictions arise from various legal frameworks across different demographics, regulating, and sometimes prohibiting, the training of models on protected data for commercial deployment. This challenge is particularly prevalent in web-scraped data, where the abundance of protected data may not be adequately filtered (Khan & Hanna, 2020). In contrast, MA data bypass this issue, as it is presumed that the data are filtered or obtained from a proprietary source with appropriate permissions during annotation. Notably, generated data offer a more advantageous position, as they do not necessitate such filtering and encounter limited or no usage restrictions, thereby providing a readily available solution to issues arising from data protection concerns.

**Privacy Issues.**    may arise when data samples inadvertently leak or explicitly contain sensitive, confidential, or private user information. Examples of such images could include those featuring people's faces (O'Sullivan, 2020; Murgia & Harlow, 2019) or personal objects that disclose identity-related details, such as addresses or financial assets. Once again, web data emerge as the primary source vulnerable to issues stemming from the use of private data, an issue extensively present in web-scraped data (Subramani et al., 2023; Wenger et al., 2022; Solon, 2019; Lukas et al., 2023). MA data are expected to be protected from privacy concerns due to prior filtering or explicit agreement on data usage prior to annotation. Using generative model for continual learning make it avoid storing real data in episodic memory, which can cause privacy concerns (Shin et al., 2017; Liu et al., 2024). However, generat

**Acquisition Cost.**    refers to the total expenses incurred in obtaining a specific number of data samples necessary to train or evaluate the learner $f_\theta$ for a particular task. As emphasized in 1, MA data entail a substantial acquisition cost, primarily due to the expenses associated with densely annotating the data through human workers. This, coupled with the rigorous filtering process, makes MA data prohibitively expensive to acquire at scale. Although web data do not require such significant financial outlay for annotation, they do require intensive filtering, which contributes to an elevated cost and poses a barrier to constructing large datasets solely from web sources. In contrast, due to the advantages in controllability, generated data boast a notably low acquisition cost for generating large and diverse datasets.

**Noise.**  pertains to instances where data that are not related to a concept are erroneously labeled as belonging to that concept. It may also mean discrepancies between the context of the data and the associated concept. As highlighted in Sec. A.24, web data often exhibit a high degree of noise, necessitating extensive filtering or label correction processes. In contrast, both MA data and generated data are less susceptible to such noise. In the case of MA data, the presumption of prior filtering serves as a primary solution to mitigate noisy data. Meanwhile, for generated data, the advantages of controllability enable the mitigation of noise resulting from inconsistencies in concept-image alignment. Despite the drawback of requiring GPU usage, T2I model inference incurs lower costs compared to MA due to its ability to generate pure images, making it a more cost-effective option.

## A.20 DETAILS OF RMD SCORE CALCULATION

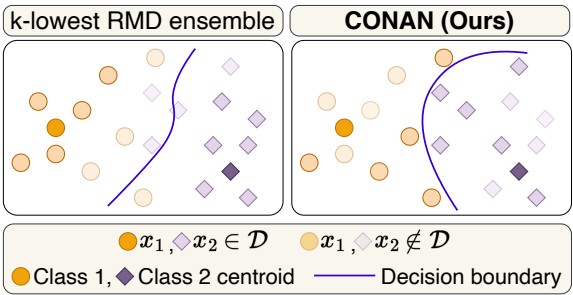

Figure 12: **CONAN** helps in finding a tighter decision boundary due to having a higher probability of including high RMD scored samples in the ensemble of generated data $\mathcal{D}$. Intuitively, high RMD scored samples are farther away from their class prototype but closer to other class samples in comparison to low RMD scored samples which are concentrated closer around the class prototype. Note that CONAN includes not only high RMD samples but also some low RMD (*i.e.*, class-representative) samples.

The RMD (Cui et al., 2023) score of a sample $(x_i, y_i)$ is defined as follows:

$$\mathcal{RMD}(x_i, y_i) = \mathcal{M}_{cls}(x_i, y_i) - \mathcal{M}_{agn}(x_i),$$

$$\mathcal{M}_{cls}(x_i, y_i) = -\left(G(x_i) - \mu_{y_i}\right)^T \Sigma^{-1} \left(G(x_i) - \mu_{y_i}\right), \tag{11}$$

$$\mathcal{M}_{agn}(x_i) = -\left(G(x_i) - \mu_{agn}\right)^T \Sigma_{agn}^{-1} \left(G(x_i) - \mu_{agn}\right),$$

where $G$ represents the feature extractor, $\mathcal{M}_{cls}(x_i, y_i)$ denotes the Mahalanobis distance from $G(x_i)$ to the corresponding class mean vector $\mu_{y_i} = \frac{1}{N_i} \sum_{y_j = y_i} G(x_j)$, with $N_i$ being the count of samples labeled as $y_i$, $\Sigma^{-1}$ denotes the inverse of the averaged covariance matrix across classes. Furthermore, $\mathcal{M}_{agn}(x_i)$ represents the class-agnostic Mahalanobis distance, where $\mu_{agn}$ denotes the overall sample mean, and $\Sigma_{agn}^{-1}$ denotes the inverse covariance for the class-agnostic case.

In the online CL setup, where data arrive in a continuous stream, it is not feasible to calculate $\mu$ and $\Sigma$ of the entire dataset. Instead, a necessity arises to continuously update these statistical parameters to accommodate the dynamic nature of the incoming data stream.

Starting with the initially computed mean vector $\mu_{y_i}$ and the covariance matrix $\Sigma$ from $N$ samples, the arrival of a new sample $x_{N+1}$ triggers an update. The updated mean vector $\mu_{\text{new}}$ is computed incrementally using a simple moving average (SMA), as follows:

$$\mu_{\text{new}} = \frac{N\mu_{\text{old}} + x_{N+1}}{N+1}. \tag{12}$$

Similarly, we calculate $\Sigma$ using a simple moving variance. Specifically, the update for the new covariance matrix $\Sigma_{\text{new}}$ is calculated using the deviation of the new sample from the old mean $\Delta = x_{N+1} - \mu_{\text{old}}$, and its deviation from the new mean $\Delta_{\text{new}} = x_{N+1} - \mu_{\text{new}}$.

Formally, we formulate the update process as follows:

$$\Sigma_{\text{new}} = \frac{1}{N+1} \left(N\Sigma_{\text{old}} + \Delta\Delta_{\text{new}}^T\right). \tag{13}$$

The update process for the class-agnostic mean vector $\mu_{\text{agn}}$ and covariance $\Sigma_{\text{agn}}$ follows the same incremental approach as described for the class-specific components.

CONAN includes a significant number of samples with high RMD scores in the ensemble dataset. Not only does it include samples with the highest RMD scores, but it also probabilistically incorporates samples with low RMD scores. This approach ensures a core-set ensemble, and we illustrate the effect of CONAN in Fig. 12

## A.21 SCALING BEHAVIOR

Recent scaling law studies (Hernandez et al., 2021; Hoffmann et al., 2022) offer predictive insight into model performance by scaling computation, data, and model capacity. Despite the limited exploration of scaling in continual learning settings (Ramasesh et al., 2022), and particularly with synthetic data (Fan et al., 2024) being confined to static frameworks, our empirical analysis in Fig. 13 delves into scaling dynamics with varying proportions of generated data for online continual learning setup.

For ResNet18 (He et al., 2016) and ViT (Dosovitskiy & Brox, 2016), we observe a consistent linear improvement trend in both ID and OOD $A_{AUC}$ as the volume of generated data increases, across the PACS (Zhou et al., 2020) dataset. This scaling behavior underscores the positive correlation between performance improvement and larger generated data ensembles in online continual learning, reinforcing the rationale for the use of generators in the absence of annotated data.

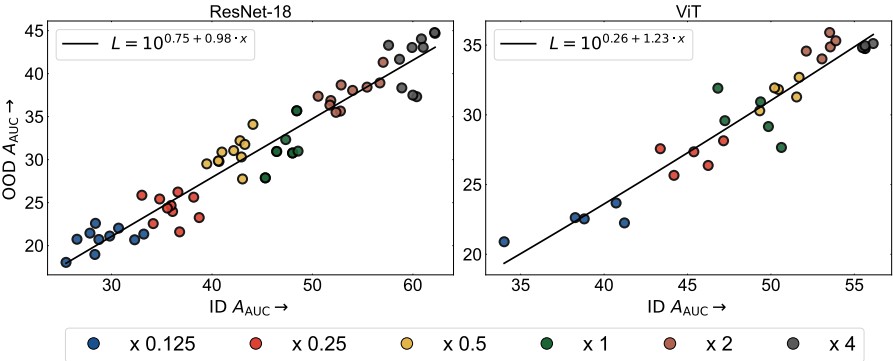

Figure 13: Ensemble scaling behavior of (a) ResNet18 (He et al., 2016) and (b) ViT (Dosovitskiy & Brox, 2016) for ID $A_{AUC}$ vs. OOD $A_{AUC}$ on the PACS dataset (Zhou et al., 2020) using ER (Rolnick et al., 2019). (x 1) denotes the ensemble volume in primary experiments, the default data budget.

## A.22 EXPERIMENTAL RESULTS ON CIFAR-10-W

We compared manually annotated (MA) data and generated data using CONAN on CIFAR-10-W (Sun et al., 2024). As mentioned in Sec. 5.1, since CIFAR-10-W only contains data on the OOD domains of CIFAR-10, we evaluated only the performance of the out-of-distribution (OOD) domain.

Additionally, since CIFAR-10-W is a web-scraped dataset, the domain of CIFAR-10-W and the web-scraped data are the same. Therefore, we excluded web-scraped data in experiments on CIFAR-10-W and only evaluated OOD performance. We summarize the results in Tab. 13.

| Method | CIFAR-10-W | |
|---|---|---|
| | $A_{AUC} \uparrow$ | $A_{last} \uparrow$ |
| Glide-Syn (ICLR 2023) | 47.14±0.80 | 34.13±0.54 |
| LE (ICLR 2023) | 47.20±0.67 | 34.03±0.60 |
| (+) CONAN | 51.69±0.70 | 41.32±1.38 |
| CHB (CVPR 2023) | 45.30±0.62 | 31.20±0.54 |
| (+) CONAN | 51.29±0.72 | 37.06±1.77 |
| SC (CVPR 2024) | 44.75±0.62 | 30.41±0.84 |
| (+) CONAN | 48.60±0.63 | 36.39±0.88 |
| CCG (arXiv 2024) | 38.96±0.94 | 24.71±0.70 |
| (+) CONAN | 41.32±0.99 | 28.94±0.77 |
| HIRPG | 52.52±0.37 | 41.04±1.26 |
| (+) CONAN (**Ours**) | **55.53±0.41** | **43.51±1.13** |

Table 13: **Qantitative comparison between different diverse prompt generation baselines on CIFAR-10-W.**

---

**Algorithm 1** GenCL

---

1: **Input** Model $f_\theta$, Prompt generation module $\psi$, Set of Generators $\mathcal{G}$, Ensembler $\Delta$, Concept stream $\mathcal{C}$, Learning rate $\mu$, Episodic memory $\mathcal{M}$
2: **for** $y \in \mathcal{Y}$ **do**          ▷ New concept arrives from concept stream $\mathcal{Y}$
3:     **Generate** $\mathcal{P}_c \leftarrow \psi(c)$       ▷ Generate prompt set $\mathcal{P}_c$ for a given concept $c$ using $\psi$
4:     **Generate** $\{\mathcal{X}_c^{(i)}\}_{i=1}^{|G|} \leftarrow \mathcal{G}(\mathcal{P}_c)$       ▷ Generate image set $\mathcal{X}_c$ using $\mathcal{G}$ and $\mathcal{P}_c$
5:     $(\mathcal{X}_c, c) \leftarrow \Delta(\{\mathcal{X}_c^{(i)}\}_{i=1}^{|G|})$       ▷ Ensemble generated image set using ensembler $\Delta$
6:     $\mathcal{L} = \mathcal{L}_{\text{CE}}(f_\theta(\mathcal{X}_c), c)$       ▷ Calculate cross entropy loss
7:     **Update** $\theta \leftarrow \theta - \mu \cdot \nabla_\theta \mathcal{L}$       ▷ Update model
8:     **Update** $\mathcal{M} \leftarrow \text{ReservoirSampler}(\mathcal{M}, (X_c, c))$       ▷ Update episodic memory
9: **end for**
10: **Output** $f_\theta$

---

**Algorithm 2** RPG

---

1: **Input** Maximum number of leaf nodes of $K$-ary Tree $K$, System prompt $P_s$, Large language model $LLM$, Prompt of parent node $P_{d,k}$
2: $\mathcal{P} \leftarrow \emptyset$       ▷ Initialize the generated prompt set $\mathcal{P}$
3: $k' \leftarrow 1$       ▷ Initialize the number of child node of $P_{d,k}$
4: **while** $k' \leq K$ **do**       ▷ Generate $k'_{th}$ child node of $P_{d,k}$
5:     **if** $k' = 1$ **then**
6:        $P_{d+1,k'} \leftarrow LLM(P_s, P_{d,k})$
7:     **else**
8:        $P_{d+1,k'} \leftarrow LLM(P_s, P_{d,k} \cup \mathcal{P})$    ▷ Recurrently forward the previously generated prompts $\mathcal{P}$
9:     **end if**
10:    $\mathcal{P} \leftarrow \mathcal{P} \cup \{P_{d+1,k'}\}$       ▷ Add the currently generated prompts to $\mathcal{P}$
11:    $k' \leftarrow k' + 1$
12: **end while**
13: **Output** $\mathcal{P}$

---

**Algorithm 3** HIRPG

---

1: **Input** Newly encountered concept $y$, Maximum number of leaf nodes of $K$-ary Tree $K$, Prompt of parent node $P_{d,k}$
2: $\mathcal{P} \leftarrow \text{RPG}(P_{d,k})$       ▷ Generate $K$ number of prompts using RPG
3: $\mathcal{P}_{ch} \leftarrow \emptyset$       ▷ Initialize the prompt set generated from the child nodes
4: **if** $d < D$ **then**
5:     **for** $P_{k'} \in \mathcal{P}$ **do**
6:        $\mathcal{P}_{ch} \leftarrow \mathcal{P}_{ch} \cup \text{HIRPG}(P_{k'}, d+1)$       ▷ Merge prompt generated in child noes
7:     **end for**
8: **end if**
9: **Output** $\mathcal{P} \cup \mathcal{P}_{ch}$    ▷ Merge the prompts generated in the child nodes and the current node, then return

---

## A.23 PSEUDOCODE FOR THE GENCL

Algorithm 1 provides a detailed pseudocode for GenCL. When a new concept is encountered, the prompt generation module $\psi$ generates concept-specific prompts. These prompts are then used by a set of T2I generators $G$ to create concept-specific images. Subsequently, the ensembler $\Delta$ selects a coreset from these generated images for efficiency, instead of training on the entire dataset. The continual learner is then trained using this selected ensemble set. During training, GenCL also stores a small portion of previously generated samples in episodic memory $M$. Although GenCL can generate images related to previously encountered concepts, retaining these samples helps to reduce computational overhead.

Additionally, Algorithm 3, Algorithm 4, Algorithm 5 provide a detailed pseudo code for prompt generation module $\psi$, a set of generators $G$, ensembler $\Delta$, respectively, which are components of GenCL.

---

**Algorithm 4** Set of Generators $\mathcal{G}$

---

1: **Input** Rewritten prompt set $\mathbf{P}$, Generative models $\mathcal{G} = \{g_1, g_2, ..., g_{|\mathcal{G}|}\}$
2: $U_1, ..., U_{|\mathcal{G}|} \leftarrow \emptyset, ..., \emptyset$ ▷ Initialize the sets of generated images for each model
3: **for** $p \in \mathbf{P}$ **do**
4:     **for** $g_i \in \mathcal{G}$ **do**
5:         **Generate** $x_y^{(i)} \leftarrow g_i(p)$ ▷ Generate image $x_y^{(i)}$ using prompt $p$ and generative model $g_i$
6:         $U_i \leftarrow U_i \cup \{x_y^{(i)}\}$ ▷ Append $x_y^{(i)}$ to the set $U_i$
7:     **end for**
8: **end for**
9: **Output** $\{U_1, U_2, ..., U_{|\mathcal{G}|}\}$ ▷ Return the generated image sets for each model

---

**Algorithm 5** Ensembler $\Delta$

---

1: **Input** Generated image sets $\{U_1, U_2, ..., U_{|\mathcal{G}|}\}$, Coreset size $|V|$, Temperature parameter $\tau$
2: $U \leftarrow \bigcup_{i=1}^{|\mathcal{G}|} U_i$ ▷ Combine all generated image sets into a single set $U$
3: **for** each sample $(x_i, y_i) \in U$ **do**
4:     **Compute** $\mathcal{RMD}(x_i, y_i) \leftarrow \mathcal{M}(x_i, y_i) - \mathcal{M}_{\text{agn}}(x_i)$ ▷ Compute RMD scores for each sample
5: **end for**
6: **Truncate** $\mathcal{RMD}(x_i, y_i)$ ▷ Remove outliers from RMD scores
7: **Normalize** $\mathcal{R\bar{M}D}(x_i, y_i) \leftarrow \mathcal{RMD}(x_i, y_i)$ ▷ Apply Z-score normalization
8: **Compute selection probability** $p_{x|y} = \frac{e^{\mathcal{R\bar{M}D}_{x|y}/\tau}}{\sum_{x' \in \mathcal{U}} e^{\mathcal{R\bar{M}D}_{x'|y}/\tau}}$ for each $x \in U$ ▷ Compute the selection probability using softmax function
9: **Select** $\mathbf{V} \leftarrow$ Sample $|V|$ images from $U$ based on probabilities $p_{x|y}$
10: **Output** Coreset $\mathbf{V}$

---

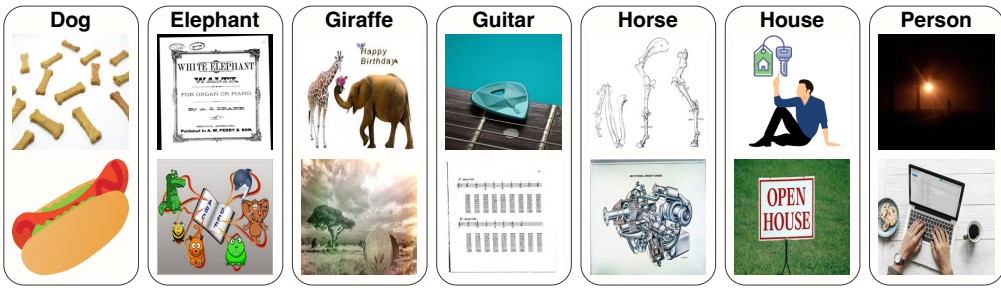

Figure 14: Examples of noisy raw data obtained via web-scraping for the classes in the PACS dataset.

## A.24 DETAILS ABOUT WEB-SCRAPPING

For web-scrapping, we follow C2C (Prabhu et al., 2024), which proposes scraping data from the web using category names. C2C (Prabhu et al., 2024) uses four search engines, including Flickr, Google, Bing, and DuckDuckGo, using the publicly available querying tool[4] to collect URLs. While C2C uses four search engines for scraping, we only use three search engines, *i.e.*, Flickr, Google, and Bing, since ICrawler did not support web data scraping from DuckDuckGo at the time of our attempt on February 20, 2024. After collecting the URLs from each search engine, we use a multi-threaded downloader[5] to quickly download the images, following (Prabhu et al., 2024). For Flickr, we are able to download approximately 500 images per minute due to the rapid URL collection facilitated by the official API. Meanwhile, for Google and Bing, the download rate is slower, at approximately 100 images per minute on a single CPU machine. However, the download rate depends on the network conditions and the status of the API and the search engine. In C2C, the ensemble of web-scrapped data from search engines is weighted differently for each benchmark. For example, in the Stanford Cars benchmark, the weights are Google: Bing: Flickr = 5:4:2, while in the Flowers benchmark, they are 1:1:2, respectively. Since we use different benchmarks compared to C2C, we select equal weight selection to ensemble web-scrapped data, *i.e.*, Google: Bing: Flickr = 1:1:1, which is one of the most straightforward and widely used ensemble techniques (Shahhosseini et al., 2022; Ju et al., 2018).

---

[4] https://github.com/hellock/icrawler
[5] https://github.com/rom1504/img2dataset

Datasets scraped from search engines such as Flickr, Google, and Bing contain uncurated (*i.e.*, noisy) samples. To clean these datasets, following (Schuhmann et al., 2022), we use a pre-trained CLIP (Radford et al., 2021) model to measure the similarity between the images and corresponding promts. Specifically, we scraped 10% more data than the required dataset size (*i.e.*, the number of manually annotated data) and removed samples with a low CLIP similarity score for each experiment. Although prior work (Prabhu et al., 2024) addressed the ambiguity of queries through manual query design, such as adding an auxiliary suffix to refine queries, in an online CL scenario, where new concepts stream in real-time, such hand-crafted query designs for each concept are limited.

In summary, data noise, network dependency, and the need for manual query design specific to each concept restrict the use of web-scrapped data in real-world scenarios where new concepts are encountered in real-time.

### A.25 ANALYSIS OF BIAS

We analyze gender bias, race bias, and geographical bias in web-scraped data (*i.e.*, C2C) and generated data from GenCL and other generative baselines (*i.e.*, LE, CHB, SC, and CCG), as well as MA for comparison.

| Bias Type | Category | Attribution Keywords |
|---|---|---|
| Gender | Female | [*'she', 'her', 'hers', 'woman', 'female', 'girl'*] |
| | Male | [*'he', 'him', 'his', 'man', 'male', 'boy'*] |
| Race | Black | [*'Black person', 'Black man', 'Black woman', 'Black boy', 'Black girl'*] |
| | White | [*'White person', 'White man', 'White woman', 'White boy', 'White girl'*] |
| | Asian | [*'Asian person', 'Asian man', 'Asian woman', 'Asian boy', 'Asian girl'*] |

Table 14: **Attention keywords for Gender/Race bias.** We categorize gender bias into female and male and race bias into Black, White, and Asian, respectively.

**Gender/Race Bias.** To measure gender/race bias, we evaluate how closely each image aligns with a specific gender or race. Specifically, we calculate the similarity between the text embeddings of gender/race attribution keywords and the image embeddings. We follow the attribution keywords used in Mandal et al. (2023) for assessing gender/race bias and summarize them in Tab. 14. Based on this alignment, we assign each image to its closest gender or race category and compare the number of images across these categories, following Wan et al. (2024); He et al. (2024).

We first compare gender and race biases among GenCL, web-scraped data (*i.e.*, C2C), and manually annotated (MA) data, summarizing the results in Fig. 15. As shown, GenCL exhibits less bias in both gender and race compared to C2C and MA, except for the race bias observed in C2C. Next, we evaluate gender and race biases in generative methods, including GenCL, and we summarize the results in Fig. 16. As depicted, GenCL demonstrates the least bias in both categories compared to other generative baselines. We believe these results stem from our proposed prompt diversification method (*i.e.*, HIRPG), which increase the diversity of generated data, as shown in Tab. 8 in Sec. A.5, thereby mitigating bias issues. To this end, we believe that training with data generated by GenCL is unlikely to introduce significant biases toward specific genders or races, as its bias levels are minimal, even when compared to manually annotated data.

To analyze the impact of biased data in continual learning, we continually fine-tune a CLIP model on the PACS dataset acquired by baselines and GenCL, which includes the *person* class. We then evaluate whether the predictions for the *person* class exhibit biases toward specific genders. Specifically, we fine-tune a pre-trained CLIP model on the person class using the prompt '*A photo of a person*' and its corresponding image pairs. During evaluation, we measure accuracy, which evaluates whether the model correctly predicts the ground truth gender of the test data. We summarize the results in Tab. 15. As shown in the table, CLIP models trained on data generated by CHB and CCG, which exhibit significant biases toward specific genders, as illustrated in Fig. 16, tend to reflect these biases in their predictions. In contrast, CLIP models trained using data generated by GenCL correctly predict gender with higher accuracy, highlighting the impact of the unbiased data generated by GenCL.

| Method | Accuracy |
|---|---|
| C2C | 71.26 |
| CHB | 64.37 |
| CCG | 64.72 |
| SC | 72.16 |
| LE | 74.71 |
| GenCL (Ours) | **77.01** |

Table 15: **Accuracy of the fine-tuned CLIP model on the 'person' class in the PACS dataset.**

**Geographical bias.** In addition to gender/race bias, we measure the geographical bias of objects, evaluating whether the generated content reflects artifacts and surroundings from across the globe, rather than disproportionately representing certain regions, as proposed by Basu et al. (2023). Specifically, similar to the method for

measuring gender and race bias, we calculate the similarity between the text embedding of *'a high-definition image of a typical {concept} in {nation}'* and the image embeddings for all concepts in the dataset. We summarize the results in Fig. 17.

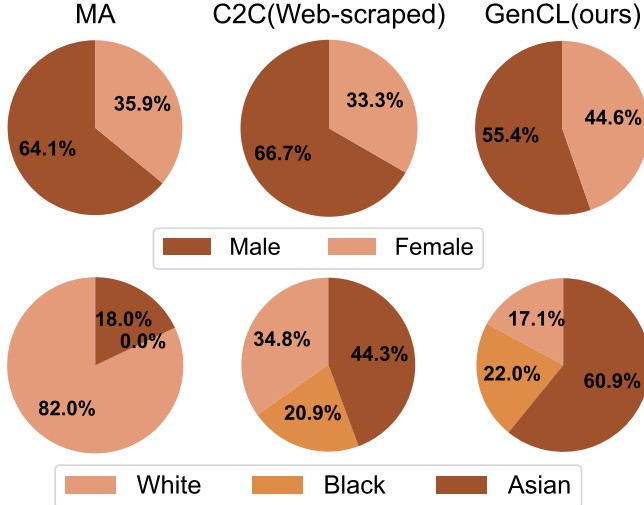

Figure 15: **Comparison of gender/race bias between GenCL, web-scraped data (*i.e.*, C2C), and manually annotated (MA) data.**

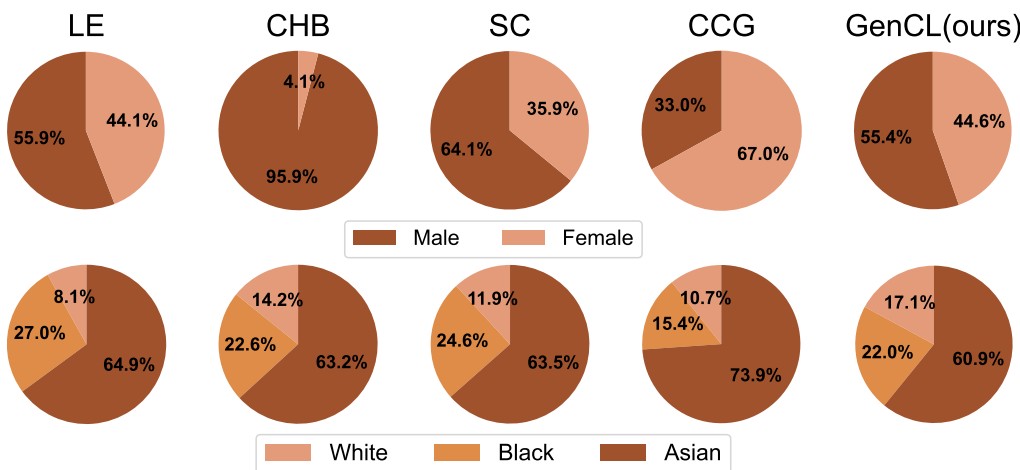

Figure 16: **Comparison of gender/race bias between generative baselines with GenCL**

### A.26 COMPARISON OF CONAN WITH HARD NEGATIVE SAMPLING METHODS

In addition to comparing CONAN with coreset selection methods, we also evaluate it against hard negative sampling methods, which prioritize hard samples based on their own selection criteria, including HCL (Robinson et al., 2021) and H-SCL (Jiang et al., 2024). We summarize the results in Tab. 16.

As shown in Tab. 16, CONAN outperforms hard negative sampling baselines in both PACS and DomainNet. We believe that the lower performance of hard negative sampling methods stems from the lack of class-representative samples, which are crucial in coreset, similar to the lower performance with $k$-highest RMD selection strategy in Sec. A.14.

### A.27 COMPARISON OF CLASS-SPECIFIC PROMPTS AND CLASS-AGNOSTIC PROMPTS

We can categorize prompt diversification strategies, which aim to generate diverse samples through diversified prompt, into two categories: class-agnostic diversification and class-aware diversification. Class-agnostic

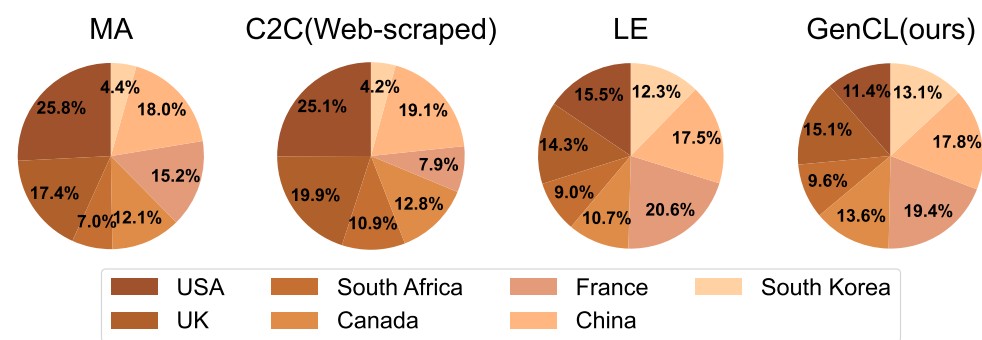

Figure 17: **Comparison of geographical bias between GenCL, web-scraped data (*i.e.*, C2C), and manually annotated (MA) data.**

| Method | PACS | | | | DomainNet | | | |
|---|---|---|---|---|---|---|---|---|
| | ID | | OOD | | ID | | OOD | |
| | $A_{\text{AUC}} \uparrow$ | $A_{last} \uparrow$ | $A_{\text{AUC}} \uparrow$ | $A_{last} \uparrow$ | $A_{\text{AUC}} \uparrow$ | $A_{last} \uparrow$ | $A_{\text{AUC}} \uparrow$ | $A_{last} \uparrow$ |
| HCL | 51.18±2.76 | 46.10±3.76 | 36.52±1.66 | 29.25±2.27 | 30.34±0.78 | 24.76±0.82 | 13.26±0.56 | 11.56±0.43 |
| H-SCL | 51.55±1.69 | 47.41±2.91 | 35.99±1.20 | 27.57±1.70 | 32.22±0.34 | 26.47±0.25 | 13.88±0.13 | 11.36±0.20 |
| CONAN (**Ours**) | **55.89±3.06** | **55.43±2.49** | **38.53±1.15** | **33.73±1.82** | **34.60±0.31** | **30.09±0.11** | **14.53±0.22** | **12.65±0.09** |

Table 16: **Quantitative comparison between hard negative sampling methods on CIL setup.**

diversification strategy first generate a set of diverse prompts, *e.g.*, {A vibrant photo of {concept} during sunrise with a warm color palette, A cinematic wide-angle shot of {concept} at dusk, ...}, and then inserts a given concept into the concept placeholder. As a result, prompts for all classes follow the same sample templates. In contrast, class-aware prompts generate unique prompts for each class, which can differ in aspects like background and color schemes.

LE, SC and CCG generate class-aware prompts—causing prompt generation time to scale with the total number of classes—GenCL employs the same template across all classes and benchmarks. While revising prompts using LLMs, such as GPT-3.5, can enhance their naturalness by incorporating suitable backgrounds and styles for each class, applying fixed templates uniformly across all classes may result in unnatural prompts. However, we draw inspiration from the renowned psychologist Dr. K. Anders Ericsson, who argued that *high-end performance of human results from extensive practice beyond one's comfort zone* Ericsson et al. (1993); Huang et al. (2022). Following this perspective, we believe that samples generated from such unnatural prompts can facilitate the learning of concepts from more diverse viewpoints. To this end, by applying fixed templates across all classes rather than generating class-specific prompts, we can significantly reduce computational costs—*i.e.*, prompt generation time becomes independent of the total number of classes—while also providing opportunities to train with more challenging samples.

### A.28 DISCUSSION OF COMPUTATIONAL AND MEMORY COST OF GENCL AND BASELINES

We compare computation cost, memory cost, and extra resources, and summarize results in Tab. 17. Since measuring FLOPs is not feasible when using APIs like GPT-4o, we compare the wall time of the methods instead. Manually annotated (MA) data requires 50000 human working hours (*i.e.*, 3,000,000 minutes) to filter out outliers. C2C does not require any GPU resources; instead, it relies on web browsers for crawling. Note that the wall time of C2C can vary significantly depending on network conditions, the status of the API, and the search engine's performance. Generative baselines, such as Glide-Syn, LE, CHB, SC, CCG, and our proposed GenCL, utilize 32 RTX 4090 GPUs for image generation. Additionally, these methods require 12GB of memory to store the weights of generative models.

The major reasons for the lower wall time of GenCL compared to baselines is the use of class-agnostic prompts, which make its prompt generation time independent of the total number of concepts, as mentioned in Sec. A.27. Real-Fake (Yuan et al., 2024) requires more storage than other baselines due to the necessity of fine-tuning diffusion models, specifically through the use of LoRA (Hu et al., 2021) for training.

### A.29 EFFECT OF HYPERPARAMETERS

Hyperparameters, including temperature $\tau$, truncate ratio $L$, depth $D$ and width $K$ of $K$-ary tree, are selected through a hyperparameter search on DomainNet and are consistently applied to the other benchmarks.

| Method | Wall time (min) | Storage usage (GB) | Extra resources |
|---|---|---|---|
| MA | 3,000,000 | 0 | Human annotation |
| IE | 330 | 0 | Web browsers |
| C2C | 300 | 0 | Web browsers |
| Glide-Syn | 254 | 4.5 | $32 \times$ RTX 4090 GPUs |
| Real-Fake | 480 | 32 | |
| LE | 240 | | |
| CHB | 222 | | |
| SC | 952 | 12 | $32 \times$ RTX 4090 GPUs |
| CCG | 952 | | |
| GenCL | 182 | | |

Table 17: **Comparison of computational cost, memory budget, and extra resources for acquiring data of concepts in DomainNet.**

**Effect of Temperature** $\tau$. A lower temperature causes the ensemble method to focus more on samples with high RMD scores (*i.e.*, difficult samples). However, setting the temperature too low can hinder performance by excluding low RMD samples (*i.e.*, easy samples). Conversely, a higher temperature increases the likelihood of including samples with low RMD scores (*i.e.*, easy samples), but setting it too high results in an ensemble containing too many easy samples. Therefore, we select an appropriate temperature via a hyperparameter search. The results of this search are presented in Fig. 18. While both excessively high and low temperatures lead to diminished performance, there is a wide range of temperatures that maintain stable performance.

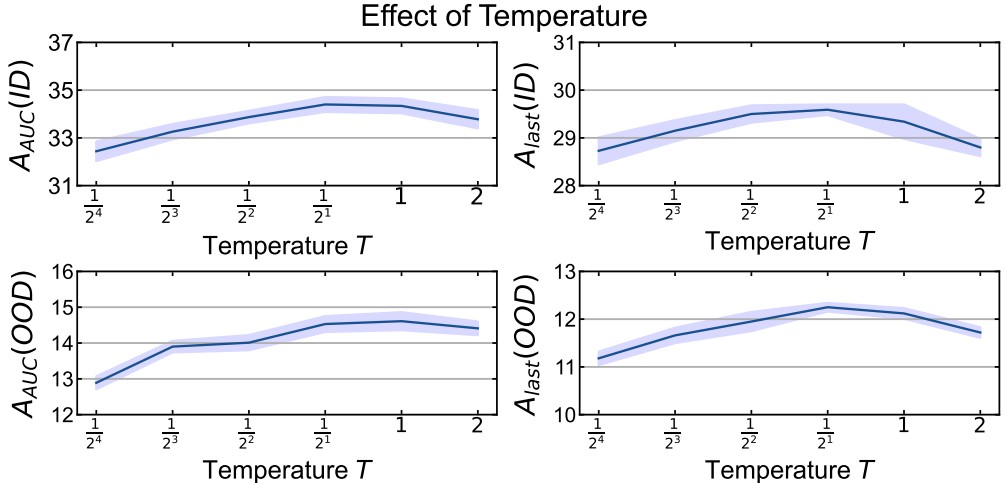

Figure 18: **Effect of temperature** $T$. We measure $A_{\text{AUC}}$ and $A_{last}$ on both ID and OOD domains in DomainNet.

**Effect of Truncate Ratio** $L$. In CONAN, we truncate the samples with RMD scores in the upper and lower $L\%$ to minimize the impact of outliers on the probability distribution. Truncating a very small portion of the candidate set may cause the ensemble set to include outliers and generated images with artifacts. In contrast, truncating a large portion of the candidate set can discard difficult samples that are not outliers, as well as easy samples (*i.e.*, concept-representative samples) that are crucial for coreset construction (Bang et al., 2021). Therefore, as shown in Fig. 19, both very high and low truncation ratios cause performance degradation. To this end, we select an appropriate truncation ratio by balancing the advantages and drawbacks of high and low truncate ratios.

**Effect of Depth** $D$ **and Width** $K$ Our proposed prompt diversification method (*i.e.*, HIRPG) utilizes a hierarchical tree structure. Specifically, we construct a complete $K$-ary tree with a depth of $D$, allowing us to generate $\frac{K^{d+1}-1}{K-1}$ diverse prompts. To generate a desired number of diverse prompts, we can adjust the parameters by either increasing $K$ and decreasing $D$, or decreasing $K$ and increasing $D$. Below, we demonstrate the effects of the tree's depth ($D$) and width ($K$), and explain how we selected the hyperparameters used in GenCL.

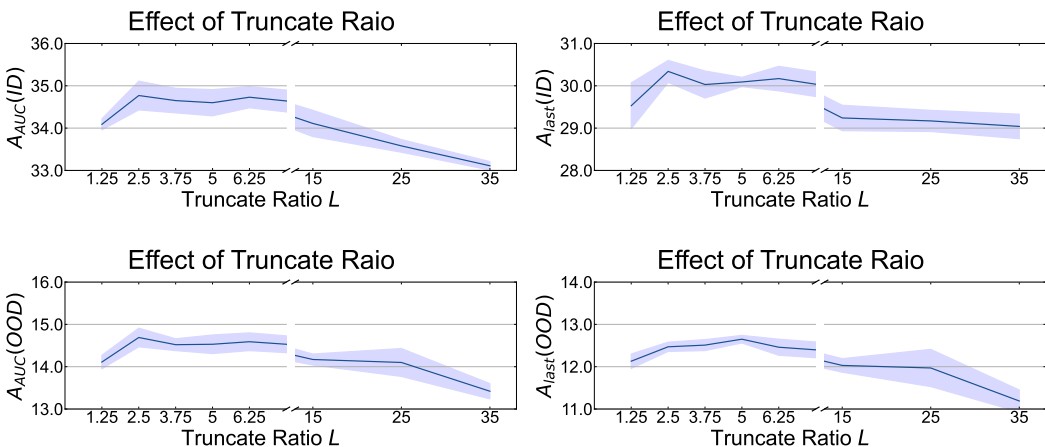

Figure 19: **Effect of Truncate Ratio** $L$. We adjust the truncation ratio over a wide range in DomainNet and measure $A_{AUC}$ and $A_{last}$ on both in-distribution and out-of-distribution domains.

In Fig.20 and Fig.21, the case of $D = 1$ demonstrates low diversity and recognizability. This occurs because significantly increasing $K$ and decreasing $D$ can lead to difficulties in fully utilizing information within the long context, a challenge referred to as the 'lost-in-the-middle' problem (Liu et al., 2023c; An et al., 2024). Specifically, to generate $N$ distinct prompts, we iteratively generate prompts at each RPG step. In the final step, $N - 1$ previously generated prompts are used as negative examples. Providing such a long context to the LLM can hinder its ability to fully comprehend and effectively reference the negative examples (*i.e.*, previously generated prompts). As a result, the LLM may produce unrecognizable prompts or generate prompts that duplicate previously created ones.

Conversely, the cases of $D = 3$ and $D = 4$, which correspond to low $K$ values, exhibit low recognizability. This is because providing only a few negative examples offers insufficient context. Specifically, recent studies (Agarwal et al., 2024) have observed signifi-

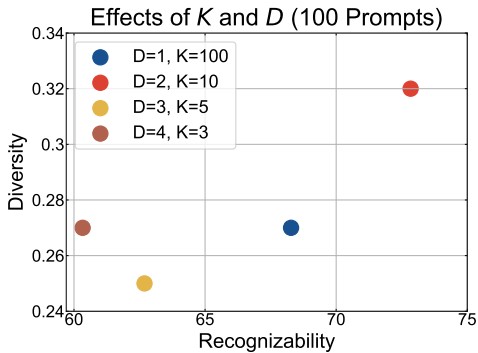

Figure 20: **Effect of** $K$ **and** $D$ **of K-ary tree in HIRPG.** To generate 100 different prompts on PACS, we generate prompts using various combinations of $K$ and $D$ of the tree. We then use these prompts to generate images and measure the Recognizability and Diversity of the generated images.

cant performance improvements across a variety of generative and discriminative tasks when using many-shot examples, compared to few-shot examples, in in-context learning.

By balancing the trade-offs of both scenarios, we recommend selecting appropriate values for $K$ that avoid being excessively high or low, along with the corresponding $D$.

### A.30 QUANTITATE ANALYSIS ON FINE-GRAINED BENCHMARKS

We evaluate GenCL on fine-grained benchmarks. Specifically, as our focus extends beyond in-distribution accuracy to include out-of-distribution accuracy, we conduct experiments on Birds-31 (Yu et al., 2024), a fine-grained domain generalization benchmark comprising CUB-200-2011 (He & Peng, 2019), NABirds (Horn et al., 2015), and iNaturalist2017 (Van Horn et al., 2018). We summarize the results in Tab. 18.

As shown in Tab. 18, GenCL outperforms the baselines and achieves performance comparable to MA. We attribute this success to GenCL's ability to effectively generate diverse images of fine-grained species through our proposed prompt diversification method (*i.e.*, HIRPG) and data ensembling method (*i.e.*, CONAN), as shown in Tab. 8 in Sec. A.5

### A.31 ANALYSIS OF SIMILARITY BETWEEN PROMPTS GENERATED AT EACH NODE IN HIRPG

We empirically demonstrate that the overlap between generated prompts from different nodes is rare by measuring the similarity of prompts generated at each node. To measure similarity, we first construct a K-ary Tree with

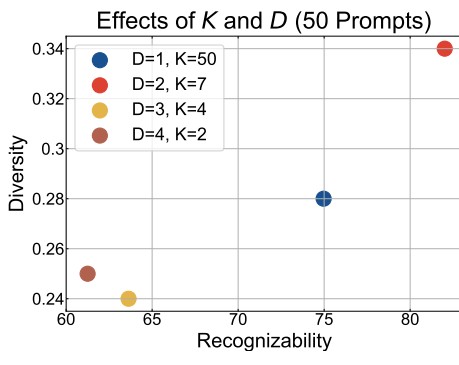 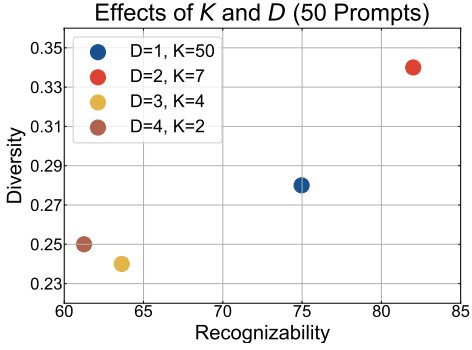

(a) Generating 50 prompts in PACS

(b) Generating 50 prompts in DomainNet

Figure 21: **Effect of $K$ and $D$ of K-ary tree in HIRPG.** To generate 50 different prompts on PACS and DomainNet, we generate prompts using various combinations of $K$ and $D$ of the tree. We then use these prompts to generate images and measure the Recognizability and Diversity of the generated images.

| Method | ID | | OOD | |
|---|---|---|---|---|
| | $A_{\text{AUC}}$ ↑ | $A_{last}$ ↑ | $A_{\text{AUC}}$ ↑ | $A_{last}$ ↑ |
| MA | 26.19±1.12 | 16.09±1.97 | 21.29±0.76 | 12.98±0.74 |
| GenCL (Ours) | 24.25±0.41 | 20.06±0.93 | 18.62±0.33 | 13.49±1.01 |

Table 18: **Qualitative comparison of baselines on Birds-31.** MA refers to the manually annotated data.

$K = 7$ and $D = 2$, following hyperparameters determined in Sec. A.29. Next, we divide the generated prompts into 7 groups, where $i$-th group refers to the set of prompts generated by RPG using the prompt of $i$-th node at depth 1 is used as the initial negative prompt. We then measure the group-wise similarity by calculating the average of all pairwise similarities between prompts generated from each node, where the similarity between prompts is measured by the cosine similarity of their respective text embeddings extracted by Sentence-BERT (Reimers, 2019). We summarize the group-wise similarity of generated prompts in a similarity matrix, as shown in Fig. 22.

As shown in the similarity matrix, The similarity between the prompt sets generated by RPG from two different nodes is generally low. In many cases, it is even lower than the intra-similarity (*i.e.*, diagonal elements) observed within the prompt sets generated from the same node. We believe this is attributed to a characteristic of LLM, where different examples provided during in-context learning lead to varied outputs (Su et al., 2022; Agarwal et al., 2024). Specifically, since RPG at each node begins with distinct initial negative examples passed from the previous depth, it generates different prompts, even though the model cannot directly reference prompts generated by other nodes as negative examples.

### A.32 QUALITATIVE RESULTS FOR PROMPT GENERATION METHODS

We also qualitatively evaluate the performance of our proposed prompt generation method, HIRPG, against existing prompt diversification baselines, including LE, CHB, SC, and CCG. The comparison is illustrated across multiple concepts from the PACS and DomainNet datasets, as shown in Table 19, Table 20, Table 21, and Table 22. We observe that most methods are not able to generate diverse prompts as well as maintain coherence and logic across generated instances. Common issues across baseline methods include irrelevant content, repetitions, and overused phrases.

LE generates repetitive phrases across difference concepts that, while slightly different in wording, essentially convey the same meaning. In Table 19 and Table 20, despite that phrases differ in their choice of words, they describe the same visual concept: a subject illuminated by soft, warm light, typically seen at sunrise or sunset. CHB, on the other hand, generate prompts with nonsensical combinations of objects and environments, such as "horse inside a bakery" or "house inside an aquarium". While diverse, the prompts are not grounded in reality, which limits their practical use in downstream tasks.

SC and CCG methods produce more coherent and consistent prompts. However, they show a tendency toward redundancy, particularly in descriptors like "majestic" and "gallops" for the *horse* concept, or "charming" and "rustic" for the *house* concept, reducing the overall uniqueness of the generated prompts. Compared to these

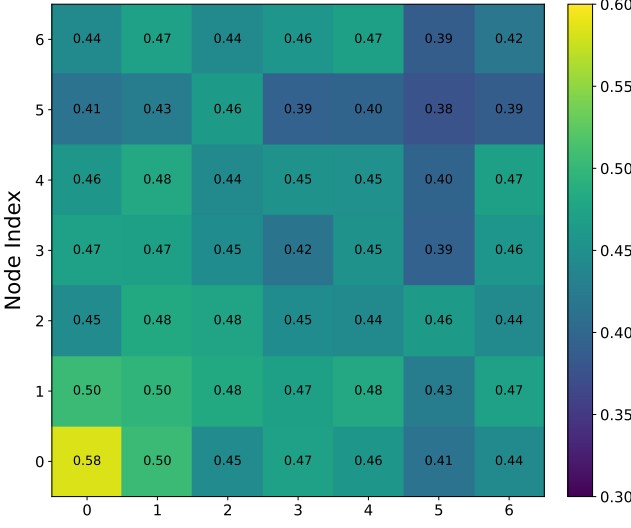

Figure 22: **Similarity matrix of prompts generated at each node in HIRPG in a $K$-ary structure with $D = 2$ and $K = 7$ on DomainNet.** Node index $i$ refers to the set of prompts generated by RPG, where the prompt of the $i$-th node at depth 1 is used as the initial negative example. We measure the similarity between two nodes by calculating the average of all pairwise similarities between prompts generated from each node. The similarity between prompts is measured by the cosine similarity of their respective text embeddings.

existing prompt diversification baselines, our proposed prompt generation method, HIRPG, successfully captures not only the diversity but also the originality and coherence within its generated prompts.

## A.33 Extended Quantitative Analysis

We perform additional comparisons with various combinations of diverse prompt generation baselines and data ensemble methods on DomainNet, and summarize the results in Tab. 23. For all data ensemble methods, including our proposed CONAN, we select an equal number of samples for the coreset to ensure a fair comparison. As shown in the table, CONAN is not only effective when combined with HIRPG, but also shows strong performance when paired with other prompt generation baselines, demonstrating its plug-and-play applicability across various methods.

## A.34 Limitations and Future Work

While GenCL replaces manual annotation by generating diverse images that only require *concepts* using HIRPG and CONAN, we acknowledge the limitations arising from the use of generative models. Here, we outline these limitations and propose future directions to address them.

**Privacy and Copyright Concerns.** Using generative models addresses privacy concerns associated with storing real data in episodic memory in replay-based CL methods by preventing data leakage from real data (Shin et al., 2017; Liu et al., 2024). However, generative models may also introduce privacy issues, such as the potential memorization of training data (Wang et al., 2023; Carlini et al., 2023). We believe that advancing privacy-preserving generative models (Xu et al., 2023; Chen & Yan, 2024) would effectively mitigate these concerns.

**Capability of Generation.** GenCL, along with other generative baselines, depends on the capability and coverage of the generative models, thus there are limitations when it comes to completely new concepts that generative models have not been trained on. To generate unseen concepts, we can fine-tune generative models using real data from these unseen concepts. We believe that developing efficient continual training methods for generative models (Uehara et al., 2024), which require less real data and computational resources, will accelerate the expansion of GenCL to accommodate unseen classes.

| Concept | Method | Examples |
|---|---|---|
| Horse | LE | • A calm horse illuminated by the first light of morning.
• A sunrise horse bathed in warm orange hues.
• A dew-drenched horse in vibrant sunset colors.
• A serene horse under a pastel sunrise.
• A peaceful horse bathed in the soft glow of dawn. |
| | CHB | • horse equine inside aquarium
• horse equine inside bakery
• horse equine inside music studio
• horse, equine inside wave
• horse, equine inside pizzeria |
| | SC | • A majestic horse stands gracefully in the middle of a sun-dappled apple orchard, surrounded by rows of apple-laden trees and the sweet fragrance of ripe fruit filling the air.
• A majestic horse stands in a lush green pasture, its mane flowing gracefully in the gentle breeze
• A majestic brown horse with a glossy coat stands gracefully in a sunlit meadow, surrounded by lush green grass and blooming wildflowers.
• A majestic horse stands gracefully in a sunlit meadow, its coat shimmering in the golden rays of the setting sun.
• A majestic horse stands gracefully in a lush, green meadow, its mane gently blowing in the breeze. |
| | CCG | • A majestic brown horse gallops freely through a sunlit meadow of wildflowers.
• A majestic horse gallops through a sunlit meadow with wildflowers swaying in the breeze.
• A majestic brown horse gallops freely across a sunlit, grassy meadow surrounded by blooming wildflowers.
• A majestic horse gallops across a sunlit field, mane flowing in the breeze.
• A majestic horse gallops through a sunlit meadow, its mane flowing in the breeze. |
| | HIRPG | • A high-contrast black and white photograph of horse.
• A serene watercolor painting of horse.
• A vibrant photo of horse during sunrise with a warm color palette.
• A vivid painting of horse using vibrant colors.
• A cinematic wide-angle shot of horse at dusk. |

Table 19: Prompt samples using different prompt generation methods for the concept *Horse* from PACS dataset. Irrelvant content, repetitions, and overused phrases are marked in red, brown, and turquoise respectively.

**Scalability.** Recent studies (Yuan et al., 2024; Hammoud et al., 2024) have shown that training a model with samples generated by text-to-image models—producing 10 to 100 times more samples than real data—can achieve performance comparable to models trained on real data. However, such scaling methods may not be suitable for the name-only continual learning setup, as they require substantial time and computational cost, whereas continual learning demands fast adaptation to new concepts (Koh et al., 2021; Ghunaim et al., 2023).

In real-world scenarios, there are various applications where a few manually annotated samples are provided along with concept names. Therefore, a small number of real data samples can be used alongside data generated by GenCL. Recent works (Seib et al., 2020; Yuan et al., 2024) have shown that jointly using generated and real

| Concept | Method | Examples |
|---------|--------|----------|
| House | LE | • A calm house illuminated by the first light of morning.
• A sunrise house bathed in warm orange hues.
• A dew-drenched house in vibrant sunset colors.
• A serene house under a pastel sunrise.
• A peaceful house bathed in the soft glow of dawn. |
| | CHB | • house, building inside aquarium
• house, building inside bakery
• house, building inside music studio
• house, building inside wave
• house, building inside pizzeria |
| | SC | • house => A charming, rustic house with ivy-covered walls and a thatched roof, nestled amidst a flourishing garden of vibrant, blooming flowers, creating a serene and picturesque scene.
• A charming countryside house with a thatched roof and ivy-covered walls, surrounded by a lush, colorful garden blooming with flowers on a sunny day
• A charming rustic house stands besides a tranquil pond, surrounded by lush greenery and vibrant blooming plants, reflecting its serene image in the still water.
• A charming rustic house, nestled in a vibrant, colorful garden filled with blooming flowers and tall trees, creating a picturesque and cozy scene.
• A charming rustic house stands besides a tranquil pond, surrounded by lush greenery and vibrant blooming plants, reflecting its serene image in the still water. |
| | CCG | • A cozy, red-brick house with a white picket fence stands amidst colorful autumn trees.
• A cozy, rustic house stands peacefully surrounded by vibrant autumn trees and a cobblestone path..
• A cozy, rustic house nestled in a forest, surrounded by vibrant autumn leaves and tall trees.
• A cozy countryside house stands surrounded by vibrant autumn foliage under a clear, blue sky.
• A cozy wooden house surrounded by colorful autumn trees under a clear blue sky. |
| | HIRPG | • A high-contrast black and white photograph of house.
• A serene watercolor painting of house.
• A vibrant photo of house during sunrise with a warm color palette.
• A vivid painting of house using vibrant colors.
• A cinematic wide-angle shot of house at dusk. |

Table 20: Prompt samples using different prompt generation methods for the concept *House* from PACS dataset. Irrelevant content, repetitions, and overused phrases are marked in red, brown, and turquoise respectively.

data further enhances neural network performance. Combining these not only improves scalability and reduces computational costs but also boosts overall performance.

**Domain Gap Between the Training Domain of Generative Models and Target Domains.** There are limitations in applying GenCL to entirely new domains that differ from the training domains of the generative models, such as medical domains. To address this, it would be beneficial to replace the generative models in

| Concept | Method | Examples |
|---|---|---|
| Diving Board | LE | • Scuba diver lands on the dry seas with his bare feet on a
• The man with the green board doing darts is really nice.
• Man in the snow at the beach... the guy on the scuba boat...
• Man on the seaboard while diving
• A man is caught in a sea of dolphins scuba diving on a |
| | CHB | • diving board, springboard inside home office
• diving board, springboard inside bakery
• diving board, springboard inside music studio
• diving board, springboard inside subway station
• diving board, springboard inside pizzeria |
| | SC | • A sleek, modern diving board extends over a pristine pool at an indoor diving facility, brightly lit with natural light streaming through large windows, casting reflections on the water's surface below.
• A sleek diving board extends over a shimmering blue swimming pool, surrounded by lush greenery and bustling with activity under the bright summer sun.
• A sleek diving board extends over an oceanview pool, with the shimmering blue waters of the ocean and a clear sky stretching out in the background.
• On a serene summer day, a sleek diving board extends over a sparkling zero-entry pool, inviting swimmers to take the plunge into its crystal-clear, gradually deepening waters.
• A sleek diving board extends over a crystal-clear synchronized swimming pool, surrounded by vibrant, choreographed swimmers creating mesmerizing patterns in the water. |
| | CCG | • A young girl prepares to leap off a wooden diving board into a sparkling pool below.
• A young girl jumps joyously off a colorful diving board into the sparkling blue pool below.
• A young girl poised on the diving board, ready to leap into the sparkling pool below.
• A young girl leaps joyfully off a high diving board into a sparkling blue pool.
• A young girl stands poised on a diving board, ready to leap into the sparkling pool. |
| | HIRPG | • A high-contrast black and white photograph of diving board.
• A serene watercolor painting of diving board.
• A vibrant photo of diving board during sunrise with a warm color palette.
• A vivid painting of diving board using vibrant colors.
• A cinematic wide-angle shot of diving board at dusk. |

Table 21: Prompt samples using different prompt generation methods for the concept *Diving Board* from DomainNet dataset. Irrelevant content, repetitions, and overused phrases are marked in red, brown, and turquoise respectively.

GenCL with those specifically trained on medical domain data, such as Medical Diffusion (Khader et al., 2022) or MedM2G (Zhan et al., 2024). Since the generative models in GenCL can be easily swapped with other generative models, we believe that choosing a generative model that is suitable for the downstream task can maximize the performance in the name-only CL setup.

| Concept | Method | Examples |
|---|---|---|
| The Great Wall of China | LE | • The great wall of china was completed
• The great wall of china
• The Great Wall of the China
• The great wall of the province of the city of Beijing
• The Wall of China... a building in the countryside |
| | CHB | • The Great Wall of China, wall inside home office
• The Great Wall of China, wall inside bakery
• The Great Wall of China, wall inside music studio
• The Great Wall of China, wall inside subway station
• The Great Wall of China wall inside pizzeria |
| | SC | • The Great Wall of China winds majestically through the dense, vibrant greenery of a bamboo forest, creating a striking contrast between ancient architecture and natural beauty.
• The Great Wall of China winds majestically through the landscape, surrounded by ancient historical courtyards that echo with the rich history of past dynasties.
• The Great Wall of China winds majestically through the dense, vibrant greenery of a bamboo forest, creating a striking contrast between ancient architecture and natural beauty.
• Majestic Great Wall of China winding through lush green hills, while sailboats gently glide across a serene lake in the foreground.
• The Great Wall of China majestically winds its way through a vibrant, sunny meadow, with lush green grass and colorful wildflowers stretching out in the foreground. |
| | CCG | • The ancient Great Wall of China winds through lush green hills under a bright blue sky.
• Tourists wander along the ancient, winding Great Wall of China amidst lush mountains.
• Tourists hike along the ancient stone path of the Great Wall winding through green hills.
• Visitors hike along the winding, ancient Great Wall of China amidst lush, rolling green hills.
• A majestic stretch of ancient stone wall winds over lush, rolling hills under a bright sky. |
| | HIRPG | • A high-contrast black and white photograph of The Great Wall of China.
• A serene watercolor painting of The Great Wall of China.
• A vibrant photo of The Great Wall of China during sunrise with a warm color palette.
• A vivid painting of The Great Wall of China using vibrant colors.
• A cinematic wide-angle shot of The Great Wall of China at dusk. |

Table 22: Prompt samples using different prompt generation methods for the concept *The Great Wall of China* from DomainNet dataset. Irrelevant content, repetitions, and overused phrases are marked in red, brown, and turquoise respectively.

| Method | ID | | OOD | |
|---|---|---|---|---|
| | $A_{\text{AUC}}\uparrow$ | $A_{last}\uparrow$ | $A_{\text{AUC}}\uparrow$ | $A_{last}\uparrow$ |
| LE | 20.01±0.27 | 15.38±0.31 | 6.40±0.13 | 4.59±0.09 |
| (+) Uncertainty | 14.66±0.30 | 9.40±0.14 | 4.85±0.08 | 3.08±0.06 |
| (+) CRAIG | 28.64±0.55 | 23.91±0.27 | 8.53±0.27 | 6.83±0.07 |
| (+) Glister | 17.53±0.44 | 11.57±0.25 | 5.67±0.15 | 3.61±0.06 |
| (+) GradMatch | 27.68±0.68 | 22.89±0.14 | 8.57±0.31 | 6.86±0.07 |
| (+) AdaCore | 24.73±0.56 | 18.95±0.25 | 7.62±0.17 | 5.53±0.12 |
| (+) LCMat | 27.72±0.49 | 23.10±0.23 | 8.50±0.22 | 6.84±0.02 |
| (+) Moderate | 21.33±0.54 | 15.91±0.27 | 6.47±0.20 | 4.56±0.11 |
| (+) CONAN | **30.80±0.63** | **25.33±0.20** | **9.54±0.25** | **7.59±0.17** |
| CHB | 16.69±0.16 | 13.45±0.19 | 5.61±0.11 | 4.18±0.05 |
| (+) Uncertainty | 11.15±0.35 | 7.06±0.15 | 3.97±0.09 | 2.41±0.05 |
| (+) CRAIG | 26.42±0.35 | 22.49±0.33 | 8.11±0.09 | 6.61±0.20 |
| (+) Glister | 14.07±0.13 | 9.38±0.06 | 4.68±0.05 | 2.98±0.04 |
| (+) GradMatch | 25.20±0.36 | 21.58±0.27 | 7.97±0.15 | 6.51±0.14 |
| (+) AdaCore | 22.29±0.31 | 17.27±0.16 | 7.23±0.09 | 5.27±0.08 |
| (+) LCMat | 24.99±0.37 | 21.46±0.24 | 7.99±0.12 | 6.68±0.10 |
| (+) Moderate | 18.64±0.24 | 13.96±0.10 | 5.92±0.07 | 4.08±0.06 |
| (+) CONAN | **29.06±0.37** | **24.52±0.17** | **9.28±0.14** | **7.56±0.14** |
| SC | 11.89±0.17 | 8.66±0.20 | 3.90±0.07 | 2.68±0.04 |
| (+) Uncertainty | 10.32±0.26 | 6.41±0.20 | 3.17±0.05 | 1.87±0.05 |
| (+) CRAIG | 20.05±0.25 | 17.13±0.16 | 6.02±0.12 | 4.83±0.08 |
| (+) Glister | 11.30±0.24 | 7.24±0.08 | 3.42±0.07 | 2.10±0.04 |
| (+) GradMatch | 19.83±0.38 | 16.82±0.19 | 5.94±0.10 | 4.78±0.08 |
| (+) AdaCore | 17.67±0.37 | 13.29±0.38 | 5.19±0.13 | 3.69±0.06 |
| (+) LCMat | 19.86±0.32 | 16.79±0.29 | 5.98±0.11 | 4.77±0.07 |
| (+) Moderate | 14.17±0.24 | 10.34±0.06 | 4.03±0.07 | 2.72±0.05 |
| (+) CONAN | **22.36±0.34** | **19.13±0.32** | **6.71±0.15** | **5.48±0.13** |
| CCG | 12.55±0.22 | 10.21±0.26 | 4.03±0.10 | 2.91±0.10 |
| (+) Uncertainty | 14.73±0.41 | 10.65±0.22 | 4.48±0.14 | 3.13±0.07 |
| (+) CRAIG | 16.72±0.27 | 14.23±0.30 | 5.16±0.08 | 4.11±0.05 |
| (+) Glister | 14.51±0.38 | 10.54±0.15 | 4.46±0.12 | 3.08±0.03 |
| (+) GradMatch | 16.75±0.26 | 14.31±0.21 | 5.15±0.11 | 4.10±0.07 |
| (+) AdaCore | 17.11±0.36 | 13.87±0.18 | 5.20±0.14 | 3.93±0.11 |
| (+) LCMat | 16.71±0.23 | 14.08±0.26 | 5.15±0.09 | 4.05±0.05 |
| (+) Moderate | 14.52±0.29 | 11.01±0.14 | 4.40±0.10 | 3.15±0.07 |
| (+) CONAN | **18.32±0.42** | **15.83±0.34** | **5.78±0.17** | **4.70±0.14** |
| HIRPG (**Ours**) | 27.72±0.30 | 23.71±0.39 | 10.70±0.19 | 8.75±0.13 |
| (+) Uncertainty | 21.90±0.37 | 15.70±0.08 | 10.01±0.23 | 7.19±0.11 |
| (+) CRAIG | 32.53±0.20 | 28.44±0.23 | 13.25±0.15 | 11.53±0.06 |
| (+) Glister | 23.16±0.37 | 16.98±0.35 | 10.56±0.26 | 7.60±0.18 |
| (+) GradMatch | 32.53±0.43 | 28.36±0.41 | 13.48±0.31 | 11.74±0.18 |
| (+) AdaCore | 32.15±0.55 | 26.83±0.18 | 13.62±0.27 | 11.37±0.04 |
| (+) LCMat | 32.38±0.44 | 28.36±0.32 | 13.42±0.26 | 11.76±0.17 |
| (+) Moderate | 25.57±0.42 | 20.38±0.16 | 10.53±0.29 | 8.17±0.13 |
| (+) CONAN (**Ours**) | **34.60±0.31** | **30.09±0.11** | **14.53±0.22** | **12.65±0.09** |

Table 23: **Quantitative comparison between data selection methods with different diverse prompt generation baselines on DomainNet.** Uncertainty, CRAIG, Glister, GradMatch, Adacore, and LCMat require fine-tuning on the full dataset to compute gradient calculations for the fine-tuned model, despite using a pretrained model for initialization. In contrast, Moderate and CONAN do not require any fine-tuning.