# OpenReview forum: "Rainbow Generator: Generating Diverse Data for Name Only Continual Learning"
_ICLR.cc/2025/Conference — Submitted to ICLR 2025_

### Official Review · Reviewer_Z3yn · 2024-10-29

**Soundness:** 3
**Presentation:** 3
**Contribution:** 2
**Rating:** 3
**Confidence:** 4

**Summary:**

This paper presents a synthetic data augmentation approach for name-only continual learning, eliminating the need for web scraping. The authors propose diverse prompt generation and a post-generation sample selection method to create a more informative coreset for new concepts. The resulting dataset achieves state-of-the-art performance in both the class continual learning setting and multimodal visual recognition tasks.

**Strengths:**

The paper is well-written and easy to follow. The authors have done an excellent job in presenting the figure and explaining the proposed method. Additionally, the experimental ablation study (both in the main paper and in the appendix) is extensive, providing a thorough analysis.

**Weaknesses:**

1. The claim (figure 1 and line 188) that synthetic data generation does not have privacy issues is debatable. Text-to-Image generative models can suffer from memorization issues, potentially reproducing its own training data, which also violates privacy. Can you address this potential limitation in the paper and discuss any measures you could take to mitigate privacy risks associated with generative models.

2. The baselines discussed in the paper are not sufficient. Numerous synthetic data augmentation methods for image classification, like [1] and [2], share similar techniques such as prompting language models, text-to-image generation, and ensemble/filtering approaches. These methods, which can be easily adapted to the name-only continual learning setting, should be compared to evaluate if the proposed method offers any unique advantages for the name-only continual learning setting. I request that the authors either include comparisons with these methods or provide a detailed explanation of why such comparisons would not be appropriate for their specific setting. It's unclear whether this name-only continual learning setting is truly distinct from normal image classification with an expanding label space, and whether it requires a dedicated solution.

3. The MVCIL setup is a little strange, as the LLM pretraining would likely have exposed it to the new concepts and do not fully align with the continual learning setting (where the introduced concepts need to be disjoint from the previous training data). It is perhaps a different setting rather than continual learning.

I'll be willing to raise the score if authors provide satisfactory response.

[1] https://arxiv.org/abs/2310.10402
[2] https://arxiv.org/pdf/2302.14051

**Questions:**

1. Can you discuss what is the sample efficiency for your proposed method? Can you report the number of synthetic data samples generated and the GPU time required for generation and training? Without these details, it is hard to assess the method's sample efficiency and computational requirements.

2. Instead of CIL, an alternative approach to address name-only continual learning could be continual open-world representation learning [3]. This involves training a generalizable embedding model in a continual learning fashion. During testing, labels can be generated by retrieving them from a gallery that associates images with their corresponding labels, leveraging the learned representations. Would the proposed method still work in this setting?

[3] https://arxiv.org/pdf/2409.05312

---

> ### Author Response · Authors · 2024-11-24
> **Answers to the questions of Reviewer Z3yn (1/2)**
>
> > The claim (figure 1 and line 188) that synthetic data generation does not have privacy issues is debatable. Text-to-Image generative models can suffer from memorization issues, potentially reproducing its own training data, which also violates privacy. Can you address this potential limitation in the paper and discuss any measures you could take to mitigate privacy risks associated with generative models.
>
> $\to$ Great point. It is indeed arguable and we revise the argument in L182 in Sec.4 of the revision. As the reviewer pointed out, the generative models may introduce privacy issues due to the potential memorization of training data. We believe that incorporating privacy-preserving generative models [4, 5] in GenCL would address data privacy associated with generated data. Considering the limitation, we revise Figure 1 to include privacy concerns associated with the use of generative models. Additionally, we add these potential limitations and discussions in Sec.A.34 of the revision.
>
> Note that to minimize privacy risks in generative models, we use off-the-shelf generative models. While fine-tuning generative models with a small amount of real data can help generate data in a desired style, it also significantly increases the risk of privacy risks [1, 2]. Therefore, we generate images exclusively using off-the-shelf generative models to mitigate these privacy concerns.
>
>
>
>
> > The baselines discussed in the paper are not sufficient. Numerous synthetic data augmentation methods for image classification, like [6] and [7], share similar techniques such as prompting language models, text-to-image generation, and ensemble/filtering approaches. These methods, which can be easily adapted to the name-only continual learning setting, should be compared to evaluate if the proposed method offers any unique advantages for the name-only continual learning setting. I request that the authors either include comparisons with these methods or provide a detailed explanation of why such comparisons would not be appropriate for their specific setting.
>
> $\to$ Thank you for the detailed suggestions. We compare the effects of the suggested methods (i.e., Real-Fake [6] and IE [7]) for the GenCL, and summarize the results in Table A and Table B below (Table 1 in the revision). Additionally, we include them in the baselines (L383) and provide a brief explanation of them in Sec.A.10 of the revision. As shown in the table, GenCL outperforms them in both the ID domain and the OOD domain.
>
> We excluded Real-Fake from the baselines in the original version because it requires real data to align the diffusion model with the real data distribution. Since Real-fake relies on both names and real data, its inclusion would not allow for a fair comparison with other name-only baselines. Additionally, we excluded IE because it lacks a publicly available code implementation, which can hinder reproducibility.
>
>
> **Table A. Comparison of GenCL with Real-Fake and IE in PACS**
> | Method | ID $A_\text{AUC}\uparrow$ | ID $A_{last}\uparrow$ | OOD $A_\text{AUC}\uparrow$ | OOD $A_{last}\uparrow$ |
> |----------------------|------------|------------|------------|------------|
> | IE | 47.29$\pm$3.29 | 38.99$\pm$2.94 | 25.74$\pm$2.11 | 18.23$\pm$1.87 |
> | Real-Fake | 55.60$\pm$2.36 | 53.00$\pm$2.26 | 28.66$\pm$1.47 | 21.22$\pm$1.33 |
> | GenCL (Ours) |55.89$\pm$3.06 |55.43$\pm$2.49 |38.53$\pm$1.15 |33.73$\pm$1.82 |
>
>
> **Table B. Comparison of GenCL with Real-Fake and IE in DomainNet**
> | Method | ID $A_\text{AUC}\uparrow$ | ID $A_{last}\uparrow$ | OOD $A_\text{AUC}\uparrow$ | OOD $A_{last}\uparrow$ |
> |----------------------|------------|------------|------------|------------|
> | IE | 34.76$\pm$0.52 | 27.55$\pm$0.24 | 11.92$\pm$0.26 | 8.50$\pm$0.14 |
> | Real-Fake | 24.43$\pm$0.26 | 18.89$\pm$0.30 | 6.33$\pm$0.11 | 4.50$\pm$0.05 |
> | GenCL (Ours) | 35.60$\pm$0.31 | 29.99$\pm$0.11 | 14.53$\pm$0.22 | 12.65$\pm$0.09 |

---

> ### Author Response · Authors · 2024-11-24
> **Answers to the questions of Reviewer Z3yn (2/2)**
>
> > It's unclear whether this name-only continual learning setting is truly distinct from normal image classification with an expanding label space, and whether it requires a dedicated solution.
>
> $\to$ The name-only continual learning setup [1] shares overlapping areas with existing methods for normal image classification with expanding label space. Consequently, our proposed GenCL can be seamlessly extended to the standard learning setup (Tab. 9), and existing methods (e.g., CHB, LE, SC, and CCG) can also be adapted for the name-only CL setup, as shown by their use as baselines in this setup (Tab. 1).
>
> However, the main difference lies in the need for real-time adaptation. Specifically, in the standard learning setup, the time spent on data collection and preprocessing does not impact performance, as these steps are completed before model training begins (Line 34). In contrast, in the name-only CL setup, ongoing data preparation is required throughout training because new concepts are continuously encountered. Delays in preparing data for these newly encountered concepts can hinder the model’s ability to quickly adapt, which is a critical requirement for a continual learner (Line 38). For example, as shown in Tab. 16 of the revision, existing baselines like SC and CCG take more than five times longer than GenCL, hindering the real-world CL application.
>
>
> > The MVCIL setup is a little strange, as the LLM pretraining would likely have exposed it to the new concepts and do not fully align with the continual learning setting (where the introduced concepts need to be disjoint from the previous training data). It is perhaps a different setting rather than continual learning.
>
> $\to$ We respectively argue that our proposed MVCIL setup is a kind of continual learning setup for the following reasons.
>
> First, although concepts such as 'ride a bike' and their corresponding images are included in pretraining, MLLMs are typically pre-trained on single-image scenarios [8], and thus struggle in multi-image scenarios such as distinguishing whether the query set belongs to the positive or negative support set. This limitation is why we use Bongard benchmarks, which consist of multiple images (e.g., three positive and three negative images). Specifically, for the pre-trained LLaVA-1.5-7B model used in our experiments, the zero-shot performance on Bongard-HOI and Bongard-OpenWorld is only 7.8% and 9.2%, respectively, while the continually trained LLaVA-1.5-7B achieves approximately 70% accuracy. As a result, we can consider the MVCIL setup a form of capability-incremental learning [9], as it continuously learns the ability to distinguish various concepts using multi-image inputs.
>
> Second, we respectfully argue that concepts learned in earlier stages can reappear in subsequent stages in continual learning setups. For instance, [10] introduces a periodic continual learning setup, where concepts previously encountered recur periodically by providing web search data as an example: the search volume for air conditioners increases during the summer, while the search volume for heaters rises in the winter, and this pattern repeats annually.

---

> ### Author Response · Authors · 2024-11-24
> **References**
>
> [1] Li et al., Shake to leak: Fine-tuning diffusion models can amplify the generative privacy risk., IEEE SaTML 2024
>
> [2] Chen et al., The Janus Interface: How Fine-Tuning in Large Language Models Amplifies the Privacy Risks, CCS 2024
>
> [3] Hayes et al., Replay in deep learning: Current approaches and missing biological elements, Neural Computation 2021
>
> [4] Chen et al., Privacy-Preserving Diffusion Model Using Homomorphic Encryption, arXiv 2024
>
> [5] Xu et al., Ffpdg: Fast, fair and private data generation., arXiv 2024
>
> [6] Yuan et al., Real-Fake: Effective Training Data Synthesis Through Distribution Matching, ICLR 2024
>
> [7] Li et al., Internet Explorer: Targeted Representation Learning on the Open Web, ICML 2023
>
> [8] Li et al., LLaVA-NeXT-Interleave: Tackling Multi-image, Video, and 3D in Large Multimodal Models, arXiv 2024
>
> [9] Cao et al., Continual LLaVA: Continual Instruction Tuning in Large Vision-Language Models, arXiv 2024
>
> [10] Koh et al., Online Boundary-Free Continual Learning by Scheduled Data Prior, ICLR 2023

---

> ### Comment · Reviewer_Z3yn · 2024-11-24
>
> Thank you for providing detailed ablation studies and explanations! After reviewing them, I am even more convinced that my initial impression was correct—this setting appears to be incremental and the contribution may be trivial. Thus I'm lowering the score to 3. However, I do appreciate the hard work. Thanks again!

---

### Official Review · Reviewer_cBvu · 2024-11-02

**Soundness:** 4
**Presentation:** 4
**Contribution:** 3
**Rating:** 5
**Confidence:** 5

**Summary:**

This paper introduces GenCL, a generative approach to address the challenge of continual learning with minimal reliance on labeled data. The authors propose a hierarchical recursive prompt generation (HIRPG) method to create diverse prompts, enabling generative models to produce varied and representative samples that cover a broad range of visual styles and contexts. Additionally, they introduce a complexity navigation method (CONAN) that selects the most representative samples from the generated data based on sample complexity. These samples are then used to train a continual learning model on class-incremental learning (CIL) and multi-modal visual-concept incremental learning (MVCIL) tasks. The authors evaluate GenCL on standard domain generalization datasets like PACS and DomainNet, demonstrating superior performance in both in-domain (ID) and out-of-domain (OOD) scenarios compared to other methods. The results highlight GenCL’s ability to generalize to new tasks while mitigating catastrophic forgetting.

**Strengths:**

1. The paper introduces a novel approach to continual learning that minimizes reliance on labeled data through effective generative data augmentation.

2. HIRPG and CONAN methods enhance data diversity and representative sample selection, contributing to improved model generalization in incremental learning.

3. Experimental results demonstrate GenCL’s superior performance in both in-domain and out-of-domain tasks, highlighting its robustness and practical applicability.

**Weaknesses:**

1. The choice of the K-ary tree structure in the HIRPG method appears arbitrary, lacking theoretical or empirical justification. In the paper, K is set to 7 and the depth D to 2, yet there is no discussion on how these hyperparameters affect the quality of generated prompts and, consequently, the diversity of generated data. Without exploring the impact of different values for K and D, it’s unclear if these settings are optimal or if they generalize well across different tasks. This lack of analysis raises concerns about the adaptability and robustness of the proposed method in other continual learning scenarios where different prompt structures may be required.

2. The paper says that "overlap between generated prompts from different nodes is rare," but it doesn’t give any theoretical reasoning or experimental proof for this.

3. The paper doesn’t provide a hyperparameter analysis for the parameters in Equation 4, which would help clarify how these settings impact experimental results and generalization. A sensitivity analysis on these hyperparameters would strengthen the claims of robustness and adaptability across tasks. Additionally, visualizing real sample features to demonstrate the effectiveness of Relative Mahalanobis Distance (RMD) could provide clearer evidence of RMD’s role in selecting complex samples.

**Questions:**

For detailed questions and suggestions, please refer to the Weaknesses section

---

> ### Author Response · Authors · 2024-11-24
> **Answers to the questions of Reviewer cBvu**
>
> > The paper doesn’t provide a hyperparameter analysis for the parameters in Equation 4, which would help clarify how these settings impact experimental results and generalization. A sensitivity analysis on these hyperparameters would strengthen the claims of robustness and adaptability across tasks. Additionally, visualizing real sample features to demonstrate the effectiveness of Relative Mahalanobis Distance (RMD) could provide clearer evidence of RMD’s role in selecting complex samples.
>
> $\to$ Great suggestions. We add hyperparameter analysis of the temperature $\tau$ and the truncation ratio $L$, which discards the upper and lower $L$% of values to minimize the impact of outliers in Sec.A.29 in the revision. There is a wide range of temperature values that maintain stable performance (i.e., insensitive to changes), while excessively high or low temperatures result in performance degradation, as shown in Fig.18. Similarly, variations in the truncation ratio have a limited impact on performance, as illustrated in Fig.19. The hyperparameters are selected through a hyperparameter search on DomainNet and are consistently applied to the other benchmarks.
>
>
>
> > The choice of the K-ary tree structure in the HIRPG method appears arbitrary, lacking theoretical or empirical justification.
>
> $\to$ Great point. We use the K-ary tree structure to generate diverse prompts, leveraging the LLM’s characteristic that providing different examples in in-context learning often leads to varied outputs [3, 4]. Specifically, by using each of the prompts generated at the current depth as initial negative examples for the next depth, we can generate prompts on an exponential scale as the depth increases, harnessing the variability in LLM-generated outputs.
>
> >> In the paper, K is set to 7 and the depth D to 2, yet there is no discussion on how these hyperparameters affect the quality of generated prompts and, consequently, the diversity of generated data. Without exploring the impact of different values for K and D, it’s unclear if these settings are optimal or if they generalize well across different tasks. This lack of analysis raises concerns about the adaptability and robustness of the proposed method in other continual learning scenarios where different prompt structures may be required.
>
> $\to$ An excessively high or low value of $K$ degrades the recognizability and diversity of the generated data. We add discussion on the effect of $K$ and $D$ in the K-ary tree in Sec.A.29 of the revision. To obtain a specific number of diverse prompts, one can either increase $K$ and decrease $D$, or decrease $K$ and increase $D$. However, excessively high values of $K$ result in a large number of negative examples being provided to the LLM at once, increasing the length of the LLM input for in-context learning (ICL). This can lead to difficulties in fully utilizing information within the long context, a challenge known as the “lost-in-the-middle” problem [1, 2].
> Excessively low values of $K$ also cause a problem, as providing only a few negative examples offers insufficient context in in-context learning [3]. To this end, by balancing the trade-offs of both scenarios, we select appropriate values for $K$ that avoid being excessively high or low, along with the corresponding $D$.
>
>
> > The paper says that "overlap between generated prompts from different nodes is rare," but it doesn’t give any theoretical reasoning or experimental proof for this.
>
> $\to$ Great suggestion, again. We provide experimental evidence to support our claims by measuring the similarity of prompts generated at each node in the $K$-ary tree structure, as detailed in Sec.A.31 of the revision.

---

> ### Author Response · Authors · 2024-11-24
> **References**
>
> [1] Liu et al., Lost in the middle: How language models use long contexts, TACL 2023
>
> [2] An et al., Make Your LLM Fully Utilize the Context, arXiv 2024
>
> [3] Agarwal et al., Many-Shot In-Context Learning, NeurIPS 2024
>
> [4] Su et al., Selective annotation makes language models better few-shot learners, arXiv 2022

---

> ### Author Response · Authors · 2024-11-27
> **Discussion Reminder**
>
> We sincerely appreciate your effort in reviewing our submission. We gently remind the reviewer that we tried our best to address your concerns via our replies and manuscript revision. As the discussion period is nearing its end, we would be glad to hear more from you if there are further concerns.

---

> ### Author Response · Authors · 2024-11-30
> **Discussion Reminder (Closing in a few days)**
>
> We sincerely appreciate your effort in reviewing our submission. We gently remind the reviewer that we tried our best to address your concerns via our replies and manuscript revision. As the discussion period is nearing its end, we would be glad to hear more from you if there are further concerns.

---

> ### Comment · Reviewer_cBvu · 2024-11-30
>
> Thank you for the detailed ablation studies and explanations. After reviewing the revision, I agree with Reviewer Z3yn’s concerns that the proposed setting appears incremental. While I appreciate the authors’ efforts, I am lowering my score to 5.

---

### Official Review · Reviewer_QykQ · 2024-11-03

**Soundness:** 2
**Presentation:** 3
**Contribution:** 2
**Rating:** 3
**Confidence:** 5

**Summary:**

This paper tackles the continual learning with the idea of data generation. Two additional techniques are proposed. The first idea is to generate diverse samples with a hierarchical prompt generation approach. The second idea is to select samples from multiple generative models with minimal overlap. The experiments are conducted on domain generalization and multi-modal visual reasoning benchmarks.

**Strengths:**

The idea of using generative models for generating useful samples for downstream tasks is interesting. It is important to explore the way of generating diverse samples with diverse prompts and generative models. Generally, it is quite easy to follow the main idea of the paper.

**Weaknesses:**

(1) The evaluation of CIL setup is very limited to a few classes (7 classes for PACS). The conventional CIL setup considered more classes (e.g. ImageNet with 1000 classes), It seems unclear how the proposed method behave when it the data scales up, which can be evaluated on CIFAR-100 and ImageNet-1K.

(2) There is no discussion of the computational cost or extra resources needed for the proposed method, although the performance is better than the competing baselines. It would be good to include a section to discuss the computational cost, such as training time, storage usage, extra resources for generating images.

(3) What about the classes of fine-grained species, such of different breeds of dogs or different kinds of birds. Does it still work with the proposed framework? It can be evaluated on fine-grained benchmarks, such as CUB, Flowers and Cars.

(4) I agree that the data generation is crucial for CL but it is difficult to scale up in its current form with text-to-image generative models, it would be good to discuss the limitations of proposed method and possible solutions.

**Questions:**

(1) The evaluation is based on pre-trained ImageNet-1k ViT-Base models, can you explain why this is the case because there are other settings for CIL?

(2) Why for the results on Tab. 3, a CLIP-pretrained ResNet-50 is used, which is different from other results?

(3) Regarding copyright and privacy concerns, I am wondering if it exists as well for generative models used in the paper?

(4) Why difficult samples are preferred for section 4.3?

**Details Of Ethics Concerns:**

The paper uses generated data from several text-to-images generative models.

---

> ### Author Response · Authors · 2024-11-24
> **Answers to the questions of Reviewer QykQ (1/2)**
>
> > The evaluation of CIL setup is very limited to a few classes (7 classes for PACS). The conventional CIL setup considered more classes (e.g. ImageNet with 1000 classes). It seems unclear how the proposed method behave when it the data scales up, which can be evaluated on CIFAR-100 and ImageNet-1K.
>
> $\to$ Interesting point! Even when the dataset scales up in number of classes, GenCL still outperforms baselines and achieves performance comparable to MA in out-of-distribution (OOD) domains. As the reviewer suggested, we compare GenCL with baselines on ImageNet-1K and summarize the results in Table A below. For evaluating in-distribution (ID) domain performance, we use the test set of manually annotated ImageNet-1K. For OOD evaluation, we use ImageNet-C [12], which introduces algorithmically generated corruptions, such as blur and noise, to the ImageNet test set.
>
> Note that we perform experiments not only on PACS, which includes 7 classes, but also on DomainNet, which includes 345 classes. We perform quantitative analysis (Section 5.2) and ablation studies (Section 5.3) on both datasets. Our proposed framework demonstrates superior performance on both relatively small and large datasets.
>
>
>
> **Table A. Comparison of GenCL with Baselines in ImageNet-1K**
> | Model | ID $A_\text{AUC}\uparrow$ | ID $A_{last}\uparrow$ | OOD $A_\text{AUC}\uparrow$ | OOD $A_{last}\uparrow$ |
> |----------------------|------------|------------|------------|------------|
> | MA | 28.36 | 20.33 | 7.52 | 5.49 |
> | Glide-syn |11.42 | 8.20 | 5.24 |3.36 |
> | CHB |12.85 | 9.35 | 6.18 | 4.05 |
> | CCG |12.96 | 9.41| 5.98 | 4.20 |
> | SC |12.06 |9.22 | 6.05 | 4.16 |
> | LE |11.25 | 8.75 | 5.56 | 3.76 |
> | GenCL (Ours) |15.24 |11.32 | 7.73 | 5.91 |
>
>
> > What about the classes of fine-grained species, such of different breeds of dogs or different kinds of birds. Does it still work with the proposed framework? It can be evaluated on fine-grained benchmarks.
>
> $\to$ Great question! Yes, the proposed GenCL still demonstrates strong performance even on fine-grained benchmarks. As the reviewer suggested, we compare GenCL with baselines on fine-grained benchmarks. Specifically, our focus extends beyond in-distribution domain performance to include out-of-domain performance, so, we conduct experiments on Birds-31 [13], a fine-grained domain generalization benchmark. We summarize the results in Tab.18 in Sec.A.30 in the revision. As shown in the table, GenCL outperforms the baselines and achieves performance comparable to MA. We attribute this success to GenCL's ability to effectively generate diverse images of fine-grained species through our proposed prompt diversification method and data ensembling method.
>
>
>
> > There is no discussion of the computational cost or extra resources needed for the proposed method, although the performance is better than the competing baselines. It would be good to include a section to discuss the computational cost, such as training time, storage usage, extra resources for generating images.
>
> $\to$ Thank you for the suggestion. GenCL requires additional resources (*e.g.*, GPUs) and storage capacity (*e.g.*, weights for generative models), similar to other generative baselines, but it consumes less wall time compared to other data acquisition methods. We add these discussions of computational cost, including training time, storage usage, and extra resources in Sec. A.28 of the revision. Since measuring FLOPs is not feasible when using proprietary APIs such as GPT-3.5, we compare the wall time of the methods instead. As shown in Tab.16 in Sec.A.28 of the revision, we consistently use 32 X RTX4090 GPUs for GenCL and other generative baselines for fair comparison. This efficiency of GenCL is attributed to its use of class-agnostic prompts, which makes it independent of the total number of concepts, as mentioned in Sec.A.27 in the revision. As a result, GenCL significantly reduces the time required for data acquisition, effectively replacing the labor-intensive process of manual annotation.

---

> ### Author Response · Authors · 2024-11-24
> **Answers to the questions of Reviewer QykQ (2/2)**
>
> > I agree that the data generation is crucial for CL but it is difficult to scale up in its current form with text-to-image generative models, it would be good to discuss the limitations of proposed method and possible solutions.
>
> $\to$ Great suggestion, again. Learning generative models demands significant computational resources to scale up for large-scale applications, despite recent studies demonstrating their feasibility in such setups. Specifically, [14, 15] have shown that training a model with samples generated by text-to-image models—producing 10 to 100 times more samples than the real data—can achieve performance comparable to models trained on real data. However, we agree with the reviewer that such scaling methods are not suitable for the name-only continual learning setup, as they require a substantial amount of time, whereas CL demands fast adaptation to new concepts. Therefore, we propose potential solutions for future research as follows.
>
> First, a small number of real data samples can be used alongside data generated by GenCL. As noted by reviewer **sA3C**, there are various applications where a few manually annotated samples are provided along with concept names. Since many recent works [14, 16] have shown that jointly using generated data and real data further enhances neural network performance, combining them not only improves scalability but also boosts overall performance.
>
> Next, efficient generative models [17, 18], capable of generating a large number of images in a short time and with minimal computational costs, can expedite the applicability of GenCL in large-scale setups. Additionally, lightweight generative models [19], which occupy a small amount of memory, enable GenCL to be applied on edge devices.
>
> We add the limitations of GenCL and possible solutions in Sec. A.34 of the revision.
>
>
> > The evaluation is based on pre-trained ImageNet-1k ViT-Base models, can you explain why this is the case because there are other settings for CIL?
>
> $\to$ It is because the pre-trained ViT models help avoid underfitting caused by insufficient training in continual learning [1]. Specifically, due to the large capacity of the ViT model and the limited training in continual learning, training a ViT model from scratch can lead to severe underfitting [1]. Therefore, following prior work [2, 3, 4], we initialize the ViT model with ImageNet-1K pre-trained weights. Furthermore, to demonstrate the effectiveness of GenCL in various architectures, we also perform experiments using from-scratch ResNet.
>
>
> > Why for the results on Tab. 3, a CLIP-pretrained ResNet-50 is used, which is different from other results?
>
> $\to$ To clarify, as a continual learning model, we did not use the CLIP-pretrained ResNet-50 but used a ResNet-18 model (L484 in the revision), consistent with other experiments, while employing a CLIP-pretrained ResNet-50 solely for the data ensembling process.  To improve clarity, we have revised the paragraph “Comparison of CONAN with Data Ensemble” in the revision. Sorry for the confusion.
>
> Since training-free data ensembling methods require a well-trained feature extractor, and training-dependent ensembling methods can also significantly reduce training time for candidate sets using a pre-trained model, we exploit CLIP-pretrained ResNet-50 for data ensembling. The resulting ensemble dataset is then used to train the ResNet-18 model as the continual learner.
>
>
> > Regarding copyright and privacy concerns, I am wondering if it exists as well for generative models used in the paper?
>
> $\to$ Yes, the generative models we used still pose privacy risks, such as the potential memorization of training data. To mitigate the concern,  privacy-preserving generative models [9, 10, 11] could be used. Considering this limitation, we modified Fig. 1 and the corresponding explanations to include concerns related to copyright and privacy issues associated with generative models. In addition, we add privacy concerns to the limitations and proposed future work in Sec. A.34 of the revision.
>
>
> > Why difficult samples are preferred for section 4.3?
>
> $\to$ It is inspired by a recent study that reveals training with difficult samples improves out-of-domain generalization performance [6]. Meanwhile, several studies [7, 8] emphasize the importance of concept-representative samples for the ‘coreset’ construction. By considering both perspectives, we prioritize selecting difficult samples over easy ones to enhance learning capability, while including a few easy samples in the ensemble set.

---

> ### Author Response · Authors · 2024-11-24
> **References**
>
> [1] Seo et al., Budgeted Online Continual Learning by Adaptive Layer Freezing and Frequency-based Sampling, arXiv 2024
>
> [2] Smith et al., A Closer Look at Rehearsal-Free Continual Learning, CVPR 2023 Workshop
>
> [3]  Li et al., Towards Continual Learning Desiderata via HSIC-Bottleneck Orthogonalization and Equiangular Embedding, AAAI 2024
>
> [4] Jeon et al., REP: Resource-Efficient Prompting for On-device Continual Learning, arXiv 2024
>
> [5] K Anders et al., The role of deliberate practice in the acquisition of expert performance, American Psychological Association 1993
>
> [6] Huang et al., The two dimensions of worst-case training and the integrated effect for out-of-domain generalization, CVPR 2022
>
> [7] Bang et al., Rainbow memory: Continual learning with a memory of diverse samples, CVPR 2021
>
> [8] Harun et al., GRASP: A Rehearsal Policy for Efficient Online Continual Learning, arXiv 2023
>
> [9] Chen et al., Privacy-Preserving Diffusion Model Using Homomorphic Encryption, arXiv 2024
>
> [10] Xu et al., Ffpdg: Fast, fair and private data generation., arXiv 2024
>
> [11] Tillman et al., Privacy-Preserving Energy-Based Generative Models for Marginal Distribution Protection, TMLR 2023
>
> [12] Hooker et al., What do compressed deep neural networks forget?, arXiv 2019
> [13] Yu et al., Fine-Grained Domain Generalization with Feature Structuralization, arXiv 2024
>
> [14] Yuan et al., Real-Fake: Effective Training Data Synthesis Through Distribution Matching, ICLR 2024
>
> [15] Hammoud et al., SynthCLIP: Are We Ready for a Fully Synthetic CLIP Training?, arXiv 2024
>
> [16] Seib et al., Mixing Real and Synthetic Data to Enhance Neural Network Training - A Review of Current Approaches, arxiv 2020
>
> [17] Ma et al., Efficient Diffusion Models: A Comprehensive Survey from Principles to Practices, arXiv 2024
>
> [18] Zhu et al., DiG: Scalable and Efficient Diffusion Models with Gated Linear Attention, arXiv 2024
>
> [19] Zhu et al., SlimFlow: Training Smaller One-Step Diffusion Models with Rectified Flow, ECCV 2024

---

> ### Author Response · Authors · 2024-11-27
> **Discussion Reminder**
>
> We sincerely appreciate your effort in reviewing our submission. We gently remind the reviewer that we tried our best to address your concerns via our replies and manuscript revision. As the discussion period is nearing its end, we would be glad to hear more from you if there are further concerns.

---

> ### Author Response · Authors · 2024-11-30
> **Discussion Reminder (Closing in a few days)**
>
> We sincerely appreciate your effort in reviewing our submission. We gently remind the reviewer that we tried our best to address your concerns via our replies and manuscript revision. As the discussion period is nearing its end, we would be glad to hear more from you if there are further concerns.

---

### Official Review · Reviewer_sA3C · 2024-11-04

**Soundness:** 4
**Presentation:** 3
**Contribution:** 3
**Rating:** 6
**Confidence:** 4

**Summary:**

This paper introduces the Generative name-only Continual Learning (GenCL) framework, designed to enhance CL but without requiring manually annotated data. The GenCL framework take care of generating diverse prompts and managing diverse data generated from multiple models. GenCL addresses privacy concerns, costs, and limitations associated with manually curated and web-scraped data. Empirical results demonstrate its effectiveness across several image recognition and multi-modal visual reasoning tasks.

**Strengths:**

The main strength of this work lies in the following presepectives:
1) It introduces an interesting and innovative approach to use generative models in name-only continual learning as a novel solution to limitations associated with labeled data which typically quite expansive in practice.
2) The introduce two techniques, HIRPG and CONAN, are important components for enhancing intra- and inter-diversity, which are essential for effective continual learning, I believe it is could also be useful in other type of learning problems.
3) The experiments are extensive and comprehensive, covering in-distribution and out-of-distribution evaluations, with comparisons against various baselines, shows the effectiveness and robustness of the proposed method.

**Weaknesses:**

1) Though the proposed method achieved promising results, it still have a large gap compared to manually annotated data. A more practical scenario worth exploring is how GenCL would perform if supplemented with a limited number (1–2) of manually annotated examples.
2) The proposed method still relies heavily on generative models that were trained on extensive datasets, potentially including names and concepts from the test data used in this paper. It remains unclear how the model would perform in domains entirely disjoint from any related training data, such as in the medical domain.

**Questions:**

1) Are there any potential biases in the generated data? How might these biases impact continual learning?
2) Have you used any mechanism to select hard negative examples and how this selection impacts model performance?

---

> ### Author Response · Authors · 2024-11-24
> **Answers to the questions of Reviewer sA3C (1/2)**
>
> > Are there any potential biases in the generated data? How might these biases impact continual learning?
>
> $\to$ Good point. Yes, generated data can potentially exhibit gender and race biases, and these biases can lead to biased predictions by models trained on them. We analyze the biases present in data generated by GenCL and other generative baselines (*i.e.*, LE, CHB, SC, and CCG), as well as MA data and web-scraped data (*i.e.*, C2C), for comparison. We summarize the result and details in Fig.15, Fig.16, and Fig.17 in Sec. A.25 of the revision. As shown in the figures, there are biases in the aspect of gender and race, data generated by GenCL exhibit less bias compared to MA and web-scraped data, as well as other generative baselines. We believe this improvement is attributed to our proposed prompt diversification methods (*i.e.*, HIRPG) and complexity-enhancing ensemble techniques (*i.e.*, CONAN), which increase the diversity of generated data, as shown in Tab.8 in Sec.A.5, thereby mitigating bias issues.
>
> To analyze the impact of biased data in continual learning, we continually fine-tune a CLIP model on the PACS dataset acquired by baselines and \frameworkname, which includes the 'person' class. We then evaluate whether the predictions for the 'person’ class exhibit biases toward specific genders. Specifically, we fine-tune a pre-trained CLIP model on the person class using the prompt `A photo of a person’' and its corresponding image pairs. During evaluation, we measure accuracy, which evaluates whether the model correctly predicts the ground truth gender of the test data. We summarize the results in Tab.15 in Sec.A.25 of the revision. As shown in the table, CLIP models trained on data generated by CHB and CCG, which exhibit significant biases toward specific genders, as illustrated in Fig.16, tend to reflect these biases in their predictions. In contrast, CLIP models trained using data generated by GenCL correctly predict gender with higher accuracy, highlighting the impact of the unbiased data generated by GenCL.
>
> > Have you used any mechanism to select hard negative examples and how this selection impact model performance?
>
> $\to$ Great suggestion. Yes, we have used $k$-highest RMD selection strategy (Tab.12 in Sec.A.14) to select the hard negative examples, and it performs poorly due to the lack of class-representative samples, which are crucial for coreset construction [9]. Additionally, as the reviewer suggested, we compare CONAN with hard negative selection strategies, including HCL [1] and H-SCL [2], on PACS and DomainNet in Tab.15 in Sec. A.26 of the revision. As shown in the table, our proposed CONAN outperforms these strategies in both ID and OOD domains. We believe that the disappointing performance of hard negative sampling stems from the lack of class-representative samples—a similar issue observed with the $k$-highest RMD selection strategy, which prioritizes only hard samples (L1880 in Sec.A.14).

---

> ### Author Response · Authors · 2024-11-24
> **Answers to the questions of Reviewer sA3C (2/2)**
>
> > Though the proposed method achieved promising results, it still have a large gap compared to manually annotated data. A more practical scenario worth exploring is how GenCL would perform if supplemented with a limited number (1–2) of manually annotated examples.
>
> $\to$ Thank you for the suggestion. For the suggested experiments, we perform experiments where a small number of manually annotated data are given, and we summarize the results in Table A and Table B below. Specifically, we assume 5 samples per class are given on both PACS and DomainNet, and fine-tune generative models using the given real data, following DreamBooth [16]. As shown in the tables, fine-tuning generative models significantly enhances performance, narrowing the gap between training with generated data and training with fully manually annotated data. Notably, it leads to substantial improvements in in-distribution (ID) domain accuracy, while out-of-distribution (OOD) domain accuracy shows minimal improvement. This indicates that fine-tuning with real data is particularly effective for improving model performance within the training domain.
>
> Despite the advantages, the reason for using off-the-shelf generative models for GenCL and assuming that real data is not provided is fine-tuning generative models with a limited number of real data can lead to overfitting or a loss of their original generalization ability [3, 4, 5], as well as catastrophic forgetting [6]. In addition, fine-tuning generative models can even amplify privacy issues [10]. Moreover, the significant computational costs of training diffusion models [14] can impede fast adaptation to novel concepts, which is a critical objective in continual learning [15].
>
> **Table A. Effect of combining GenCL with a few real samples in PACS**
> | Method | ID $A_\text{AUC}\uparrow$ | ID $A_{last}\uparrow$ | OOD $A_\text{AUC}\uparrow$ | OOD $A_{last}\uparrow$ |
> |----------------------|------------|------------|------------|------------|
> | GenCL wo real data | 51.36$\pm$2.59 | 51.63$\pm$2.49 | 34.12$\pm$1.27 | 27.18$\pm$1.32 |
> | GenCL with 3 real samples/class  | 53.86$\pm$2.54 | 51.93$\pm$2.34 | 34.20$\pm$1.49 | 27.38$\pm$2.29 |
> | GenCL with 5 real samples/class  | 54.32$\pm$1.92 | 56.67$\pm$1.02 | 34.09$\pm$1.21 | 27.35$\pm$1.29 |
>
> **Table B. Effect of combining GenCL with a few real samples in DomainNet**
> | Method | ID $A_\text{AUC}\uparrow$ | ID $A_{last}\uparrow$ | OOD $A_\text{AUC}\uparrow$ | OOD $A_{last}\uparrow$ |
> |----------------------|------------|------------|------------|------------|
> | GenCL wo real data | 27.72$\pm$0.30 | 23.71$\pm$0.39 | 10.70$\pm$0.19 | 8.75$\pm$0.13 |
> | GenCL with 5 real samples/class  | 30.27$\pm$0.31 | 25.72$\pm$0.17 | 11.36$\pm$0.21 | 9.58$\pm$0.14 |
>
> > The proposed method still relies heavily on generative models that were trained on extensive datasets, potentially including names and concepts from the test data used in this paper. It remains unclear how the model would perform in domains entirely disjoint from any related training data, such as in the medical domain.
>
> $\to$ Interesting point. If the model performs in a distant domain, *e.g.*, medical, the performance would be degraded since it is the limitation as most of the generative model-based methods face. But if we are allowed to fine-tune the generative models with a few real examples, it may fill a performance gap, as shown in Table A and Table B.

---

> ### Author Response · Authors · 2024-11-24
> **References**
>
> [1] Robinson et al., Contrastive Learning with Hard Negative Samples, ICLR 2021
>
> [2] Jiang et al., Supervised Contrastive Learning with Hard Negative Samples, IJCNN 2024
>
> [3] Moon et al., Fine-tuning Diffusion Models with Limited Data, NeurIPS 2022 Workshop
>
> [4] Han et al., SVDiff : Compact Parameter Space for Diffusion Fine-Tuning, ICCV 2023
>
> [5] Zeng et al., Infusion: Preventing Customized Text-to-Image Diffusion from Overfitting, ACM Multimedia 2024
>
> [6] Staniszewski et al., Low-Rank Continual Personalization of Diffusion Models, arXiv 2024
>
> [7] Khader et al., Medical diffusion: denoising diffusion probabilistic models for 3D medical image generation, arXiv 2022
>
> [8] Zhan et al., MedM2G: Unifying Medical Multi-Modal Generation via Cross-Guided Diffusion with Visual Invariant, CVPR 2024
>
> [9] Bang et al., Rainbow Memory: Continual Learning with a Memory of Diverse Samples, CVPR 2021
>
> [10] Li et al., Shake to leak: Fine-tuning diffusion models can amplify the generative privacy risk., IEEE SaTML 2024
>
> [11] Masip et al., Continual Learning of Diffusion Models with Generative Distillation, CoLLAs 2024
>
> [12] Chen et al., Privacy-Preserving Diffusion Model Using Homomorphic Encryption, arXiv 2024
>
> [13] Xu et al., Ffpdg: Fast, fair and private data generation., arXiv 2024
>
> [14] Xu et al., Towards Faster Training of Diffusion Models: An Inspiration of A Consistency Phenomenon, arXiv 2024
>
> [15] Caccia et al., On Anytime Learning at Macroscale, CoLLAs 2022
>
> [16] Ruiz et al., DreamBooth: Fine Tuning Text-to-Image Diffusion Models for Subject-Driven Generation, CVPR 2023

---

> ### Author Response · Authors · 2024-11-27
> **Discussion Reminder**
>
> We sincerely appreciate your effort in reviewing our submission. We gently remind the reviewer that we tried our best to address your concerns via our replies and manuscript revision. As the discussion period is nearing its end, we would be glad to hear more from you if there are further concerns.

---

> ### Author Response · Authors · 2024-11-30
> **Discussion Reminder (Closing in a few days)**
>
> We sincerely appreciate your effort in reviewing our submission. We gently remind the reviewer that we tried our best to address your concerns via our replies and manuscript revision. As the discussion period is nearing its end, we would be glad to hear more from you if there are further concerns.

---

> > ### Comment · Reviewer_sA3C · 2024-12-02
> >
> > Thank you to the authors for the detailed responses. Most of my concerns have been addressed. After carefully reviewing other reviews and responses, I believe the manuscript requires further enhancements, particularly in addressing privacy issues and ensuring fair comparisons. Therefore, I am keeping my original rating.

---

### Author Response · Authors · 2024-11-24
**General response**

We sincerely thank the reviewers for their valuable feedback and encouraging comments including  interesting and innovative approach (**sA3C, cBvu**), applicability (**sA3C**), useful approach (**QykQ**), extensive and comprehensive experiments (**sA3C, cBvu, Z3yn**), and clear presentation (**QykQ, Z3yn**).

We have uploaded the first revision of the manuscript (changes are highlighted by red color).

---

### Meta-Review · Area_Chair_9qX4 · 2024-12-22

**Metareview:**

This paper introduces the generative name-only continual learning setting, where generative models are used to create data for training based on label names. The key methods introduced are how to generate appropriate prompts such that they are diverse, as well as selection of the data from multiple prompts/models. While the reviewers appreciated the paper's overall clarity of the idea and experimental results, a number of strong concerns were raised. First, the usage of such generative models that have been trained on a large amount of prior data raise many concerns including potential biases (sA3C), inability to address truly novel categories that are not in-distribution in the sense of new categories or fine-grained (or heavily covered) (sA3C, QykQ, Z3yn), existence of biases (sA3C), and unclear advantages in terms of privacy/copyright (sA3C, Z3yn, QykQ). There were other concerns about the method itself including computational cost, ad-hoc nature of the method, and importantly lack of thorough experimental results covering in-distribution, out-of-distribution, fine-grained, and truly novel categories as well as comparison to well-known state-of-art baselines such as Internet Explorer (IE), which the paper is situated against in the main figure. While the rebuttal provided a number of results including ImageNet-1K, fine-grained datasets, and comparisons to methods such as IE, three of the reviewers overall recommended not accepting this paper due to these limitations.

  After reviewing all of the materials, including the paper, reviews, rebuttals, and discussion, I agree with the reviewers that significant effort is needed to improve the paper and do not recommend acceptance at this time. While the idea is certainly interesting, the large amount of both practical and conceptual issues that were raised cannot be addressed without significant revision. For example, methods such as IE should have been compared to across all datasets (including ImageNet, etc.) and this is not possible to thoroughly do in the rebuttal period. Further, the underlying idea of using generative models raises a number of concerns including whether the categories are truly "new categories"; this does not in and of itself mean that such a method cannot be accepted, but there should be thorough experiments demonstrating exactly where such methods work and don't work (e.g. medical imagery as mentioned by the authors) to really understand the level of generalization involved. I encourage the authors to significantly address all of the reviewers concerns for a future resubmission.

**Additional Comments On Reviewer Discussion:**

The reviewers raised a large number of concerns across the different reviewers, and while the authors provided some responses and new experiments, reviewers were not overall satisfied and many of the concerns just cannot be well-addressed in a rebuttal period. For example, comprehensive experiments comparing appropriately to state-of-art baselines across a range of distribution shifts.

---

### Decision · Program_Chairs · 2025-01-22

Reject